# Representational Similarity and Model Behavior in Multi-Agent Interaction

Yujin Potter [† 1]   Seun Eisape [* 1]   Shiyang Lai [* 2]   Alexander Huth [1]   James Evans [2 3 4]   Been Kim [5]
Jacob Eisenstein [5]   Dawn Song [1]   Alane Suhr [1]

## Abstract

Researchers have shown that neural similarity among humans predicts social closeness and cooperative success, whereas innovation often emerges from interactions among dissimilar individuals. We investigate whether these principles extend to artificial intelligence by examining interactions between large language models. In our experiments, 276 model pairs interact across eight games spanning both cooperation and novelty. We find that pairs with more similar representation spaces achieve significantly higher cooperation but exhibit reduced novelty and creativity. The effects of representational similarity on cooperation and novelty remain robust even after controlling for other factors such as performance disparity and model size. We also find that similarity in the early layers consistently shows the strongest association with cooperation and novelty, compared to the middle and later layers. This suggests that a central factor underlying these patterns could be the extent to which the two models share lexical and semantic grounding. Overall, representational similarity can be an important consideration in multi-agent system design.

## 1. Introduction

The deployment of multiple large language models (LLMs) in multi-turn, multi-agent interactions has progressed rapidly from concept to practice, with recent investigations in applications to social simulations (Park et al., 2023; Xie et al., 2024; Zhou et al., 2023), coding (Wu et al., 2024a; Ishibashi & Nishimura, 2024), and a range of creative tasks

[†]Lead author; [*]Core contributor; Code and dataset are available at https://github.com/sunblaze-ucb/similarity_interaction. [1]University of California, Berkeley [2]The University of Chicago, Knowledge Lab [3]Santa Fe Institute [4]Google [5]Google DeepMind. Correspondence to: Yujin Potter <yujinyujin9393@berkeley.edu>.

*Proceedings of the 43rd International Conference on Machine Learning*, Seoul, South Korea. PMLR 306, 2026. Copyright 2026 by the author(s).

such as writing, brainstorming, and scientific idea generation (Chen et al., 2023; Fukumura & Ito, 2025; Su et al., 2025). In many collaborative tasks, prior work has found that interaction between multiple agents facilitates stronger performance than single-agent systems (Talebirad & Nadiri, 2023; Zhuge et al., 2023). Beyond treating multi-agent systems as tools, some have even proposed evolving LLMs through multi-agent interaction (Lai et al., 2024; Eisenstein et al., 2025; Wu et al., 2025).

On the other hand, by their very nature, multi-agent systems are more complex than single-agent systems, increasing the potential for unexpected behaviors (Piatti et al., 2024; Hammond et al., 2025; de Witt, 2025). One central concern is whether agents can reliably cooperate with one another, which is critical for the safe and reliable deployment of multi-agent systems (Piedrahita et al., 2025). Being able to understand and predict the dynamics of multi-agent systems is therefore essential. Yet, most efforts to date have focused on single-agent cases, while studies of multi-agent systems have primarily focused on output-level behaviors rather than internal mechanisms.

This work provides an initial exploration of multi-agent interaction through the lens of representational alignment. Specifically, we ask:

*What is the relationship between representational similarity and interactive behavior of models?*

Evidence from neuroscience and social sciences suggests that similar neural responses among humans are significantly associated with their social closeness and cooperative performance (Parkinson et al., 2018; Thornton & Mitchell, 2017; Shen et al., 2025b; Reinero et al., 2021), while interaction between dissimilar individuals often sparks innovation (Hewlett et al., 2013; Østergaard et al., 2011). Analogously, we hypothesize that *models with higher representational similarity are more likely to cooperate and predict one another, but exhibit reduced collective novelty and creativity.*

To test this, we conduct experiments involving 276 model pairs spanning 23 open-weight LLMs from eight model families. Specifically, we examine cooperation through four games: word guessing, public good, divide-a-dollar, and the Keynesian Beauty Contest (KBC); and assess creativity

and novelty through four generative tasks: story writing, fictional biography, haiku composition, and vacation benefit brainstorming.

Our experiments reveal that representational similarity is a strong predictor of interactive outcomes. Figure 1 illustrates how these outcomes vary with increasing internal similarity across scenarios: cooperation performance rises significantly as representational similarity increases. For example, in the word-guessing game where one player attempts to identify their partner's secret word, correct guesses increase by roughly 66.2% (relative change) as representational similarity rises from the minimum to the maximum observed values. By contrast, novelty declines consistently across the four creative tasks, though the magnitude and statistical significance vary. Even when accounting for other factors such as performance disparity, representational similarity shows a strong effect on cooperation and novelty. These findings suggest a likely tradeoff: model pairs with higher representational similarity tend to cooperate better, but also manifest reduced collective novelty. These results provide new insights into the design of multi-agent systems, where single-model deployment is currently the dominant paradigm.

## 2. Related Work

**Multi-Agent Systems and Their Behaviors.** A growing body of literature examines how LLMs interact and cooperate within multi-agent systems (Piatti et al., 2024; Lai et al., 2024; Li & Shirado, 2025; Piedrahita et al., 2025; Wu et al., 2024b; Zhu et al., 2025; Kim, 2025). To study cooperative behaviors, these works commonly employ economic games such as the public good game (Hauert et al., 2006) and the tragedy of the commons (Hardin, 1968), both of which have long served as canonical paradigms for analyzing cooperation. Lai et al. (2024) found that agents become substantially more cooperative in the public good game, after engaging in multi-round, free-form interactions with peers. Wu et al. (2024b) further showed that LLM agents can develop emergent cooperative strategies across various environments, even in settings that are competitive or only partially cooperative. In contrast, several studies focusing on reasoning models have reported reduced cooperation, where these models often act as free riders in economic games (Li & Shirado, 2025; Piedrahita et al., 2025).

Another line of research compares the performance of multi-agent systems with that of single-agent systems (Chen et al., 2023; Lai et al., 2024; Li et al., 2024; Du et al., 2023; Su et al., 2025). For example, AutoAgents (Chen et al., 2023) demonstrated that sharing diverse perspectives among agents can yield more creative novel writing. Lai et al. (2024) similarly found that agents that undergo free-form interaction exhibit enhanced creativity in sentence-generation

tasks. Building on these prior studies, we investigate cooperative behaviors and creativity of multi-agent systems.

**Neural Similarity as a Predictor of Interaction in Humans and Models.** In neuroscience, similar neural responses between humans significantly predict social dynamics: Parkinson et al. (2018) found that friendship formation is predicted by similarity of neural response patterns to videos, as measured by fMRI. Shen et al. (2025b) extended this analysis to the similarity of neural activations during story-listening. Others have also shown that consistent neural activity patterns appear among personally familiar individuals (Thornton & Mitchell, 2017; Hyon et al., 2020). Such consistent findings suggest that greater neural similarity may facilitate stronger social bonds. Moreover, a related body of research has investigated the relationship between neural similarity and cooperative performance (Cui et al., 2012; Hu et al., 2017; Reinero et al., 2021; Réveillé et al., 2024), where it has been consistently found that higher interbrain synchrony is positively associated with cooperation.

As AI models scale in size and improve in performance, their internal representations increasingly align with human neural activity patterns (Goldstein et al., 2022; Schrimpf et al., 2021; Caucheteux & King, 2022; Shen et al., 2025a; Gurnee et al., 2023; Zhou et al., 2025). For example, Shen et al. (2025a) reported a strong correlation between brain-model similarity scores and model performance across both language models and vision models. This growing evidence of brain–model alignment motivates our central hypothesis. It raises the possibility that principles observed in human cognition and social behavior may also extend to advanced artificial systems.

**Representational Similarity in Neural Networks.** Researchers have long sought to understand the behavior of neural networks by comparing their internal representations. A variety of metrics for representational similarity in artificial neural networks have been proposed (Kornblith et al., 2019; Hotelling, 1992; Morcos et al., 2018; Raghu et al., 2017; Kriegeskorte et al., 2008). One popular method is Centered Kernel Alignment (CKA; Kornblith et al., 2019), which enables comparison of representations between models regardless of their architecture or layer count. Moschella et al. (2022) demonstrated that representational similarity can serve as a strong predictor of model performance, in tasks such as classification with vision models.

Beyond standard similarity metrics, new approaches have been proposed for comparing representation spaces. For example, *model stitching*—connecting two neural networks—has been argued to capture aspects of representational structure that metrics like CKA can not (Lenc & Vedaldi, 2015; Bansal et al., 2021). In this view, models with greater similarity are expected to achieve higher stitching success.However, model stitching is substantially more expen-

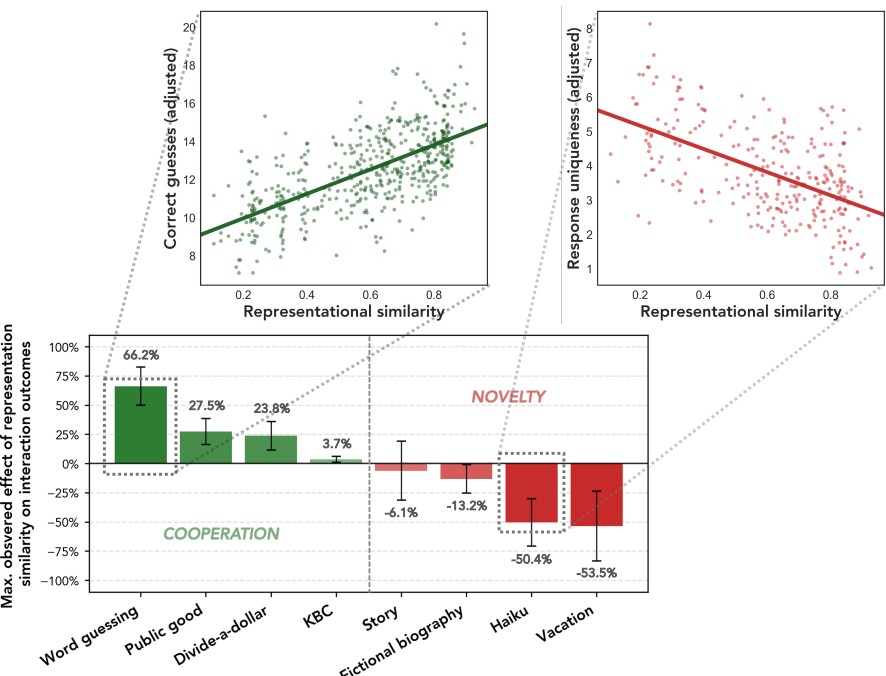

*Figure 1.* The effect of representational similarity on each game outcome. Representational similarity is quantified using linear Centered Kernel Alignment (CKA) (Kornblith et al., 2019) with WikiText (Merity et al., 2016). In the bar graph, the effect size reflects the relative change (%) in predicted outcomes between the lowest and the highest observed value of representational similarity, with error bars indicating 95% confidence intervals. In the scatter plots, each point represents a model pair, and the $y$ values are adjusted via mixed-effects regression to control for model-specific tendencies, thereby isolating the effect of representational similarity on interaction outcomes. Overall, greater similarity corresponds to higher cooperation but lower novelty.

sive and less scalable than other representational similarity metrics, as it requires training a connector layer between two models. Hacohen et al. (2020) proposed comparing the similarity of classification predictions in vision models as an alternative perspective on model comparison.

**Diversity, Creativity, and Collective Intelligence.** Behavioral research on innovation finds that higher diversity within a group of collaborators leads to increased novelty in their creations. For example, Uzzi et al. (2013) analyzed millions of scientific papers and found that the highest-impact science often arises from groups that combined existing research in novel ways. Similarly, Paulus (2000) showed that the effectiveness of brainstorming depends on cognitive diversity—that is, differences in how individuals perceive and think. Our results empirically explore this human-inspired principle: Does representational diversity within sets of LLM agents predict greater novelty in multi-agent creative tasks?

## 3. Representational Similarity of LLMs

We present related work in Appendix 2. To test our hypothesis—whether there is a relationship between representational similarity and interactive behavior of models—we first need a way to measure representational similarity of LLMs. In this section, we describe how we compute this similarity. *It is important to note that CKA computation is*

*conducted independently of model interaction.*

### 3.1. Similarity metrics

Representational similarity quantifies how similarly two neural models embed the same inputs. Measuring similarity involves two steps: **1)** extracting representational vectors from each model using a probe dataset (i.e., a set of prompts) and **2)** computing a similarity score between the extracted representations using a metric.

**Step 1. Extracting representations.** The first step can be formalized as follows. The probe dataset $\mathcal{D} \subset \mathcal{X}$ contains $m$ texts $x \in \mathcal{X}$, where $\mathcal{X}$ is the set of all possible texts. Thus, $\mathcal{D} = \{x_i\}_{i=1}^m$. For a neural model with parameters $\theta$, we define $f_\theta^k : \mathcal{X} \to \mathbb{R}^n$ as the mapping from a text $x \in \mathcal{X}$ to an $n$-dimensional activation at the $k$-th layer, where $1 \le k \le l$ and the model has $l$ layers. Stacking the embeddings for all $x \in \mathcal{D}$ yields a matrix $R_\theta^k \in \mathbb{R}^{m \times n}$, with the $i$-th row equal to $f_\theta^k(x_i)$.

**Step 2. Computing similarity.** The next step is to compute similarity between the representational spaces $\{R_{\theta_1}^i\}_{1 \le i \le l_1}$ and $\{R_{\theta_2}^j\}_{1 \le j \le l_2}$, for two models with parameters $\theta_1, \theta_2$, depths $l_1, l_2$, and hidden dimensions $n_1, n_2$. A variety of similarity metrics ($M$) have been proposed, including Centered Kernel Alignment (CKA; Kornblith et al., 2019), Canonical Correlation Analysis (CCA; Hotelling, 1992;

Morcos et al., 2018), Singular Vector Canonical Correlation Analysis (SVCCA; Raghu et al., 2017), and Representational Similarity Analysis (RSA; Kriegeskorte et al., 2008). Each defines a function $M : \mathbb{R}^{m \times n_1} \times \mathbb{R}^{m \times n_2} \to \mathbb{R}$ that takes two matrices (i.e., $R_{\theta_1}^i$ and $R_{\theta_2}^j$) as input.

We use CKA (Kornblith et al., 2019) given its popularity of use in prior work (Ciernik et al., 2024; Shen et al., 2025a; Liu et al., 2025). CKA enables the comparison between two models with different architectures and different numbers of layers. There are four common CKA variants: linear CKA, RBF CKA, unbiased linear CKA, and unbiased RBF CKA. These CKA values range in $[0, 1]$, with higher values indicating greater similarity. Following prior work (Liu et al., 2025; Shen et al., 2025a; Zou et al., 2023; Raffel et al., 2020), we obtain $f_{\theta_1}^i(x)$ and $f_{\theta_2}^j(x)$ from the activation of the last token of each input $x$ at their respective layers. CKA scores are then calculated for every layer pair of the two models. That is, $\text{CKA}(R_{\theta_1}^i, R_{\theta_2}^j)$ for all $1 \leq i \leq l_1$ and $1 \leq j \leq l_2$, producing an $l_1 \times l_2$ grid of scores.

To summarize similarity with a single score per model pair, there are multiple approaches. The first approach is to average the CKA scores (i.e., global average): $\frac{\sum_{i,j} \text{CKA}(R_{\theta_1}^i, R_{\theta_2}^j)}{l_1 \cdot l_2}$. This captures overall similarity between all layers of the two models. Please note that identical model pairs can score below 1, since off-diagonal layer pairs ($i \neq j$) yield values less than 1. An alternative summary measure of CKA is the layer-wise maximum-aligned average, which captures how well each layer aligns with its best-matching layer in the other model. That is,

$$\frac{1}{2} \left( \frac{\sum_i \max_j \text{CKA}(R_{\theta_1}^i, R_{\theta_2}^j)}{l_1} + \frac{\sum_j \max_i \text{CKA}(R_{\theta_1}^i, R_{\theta_2}^j)}{l_2} \right) \quad (1)$$

With this measure, identical model pairs achieve 1, since each layer's best match is itself and $\text{CKA}(R_\theta^i, R_\theta^i) = 1$.

We observe consistent trends between representational similarity and interactive behavior across both aggregation methods and all four CKA variants. Unless otherwise noted, CKA refers to the global averages of linear CKA. Results for other variants appear in Appendix E.

### 3.2. Probe dataset

A recent study (Ciernik et al., 2024) shows that representational similarity can depend on the choice of probe dataset. To examine whether the relationship between representational similarity and model interactions depends on probe dataset, we use four probe datasets spanning different domains, from which we compute a CKA score for each: WikiText (Merity et al., 2016) for general language, GSM8K (Cobbe et al., 2021) and MATH (Hendrycks et al., 2021) for mathematics, and TruthfulQA (Lin et al., 2021) for truthfulness. From WikiText and MATH, we randomly

sample 1000 prompts each. For GSM8K, we use the entire test set (1319 prompts), and for TruthfulQA, we use the full dataset (817 prompts).

### 3.3. Representational similarity range

We consider 23 open-weight LLMs spanning eight families and sizes ranging from 1B to 72B parameters, yielding 276 model pairs. The full model list is provided in Table 1 in Appendix. These models exhibit a wide range of representational similarity. For example, using the global average score with WikiText as the probe dataset, values range from 0.106 (gemma-3-4b-it vs. gemma-3-12b-it) to 0.92 (phi-4 vs. phi-4). Using the average of maximum-aligned scores, values range from 0.288 (gemma-3-4b-it vs. gemma-3-12b-it) to 1 (for all identical model pairs). The complete set of CKA scores for all 276 pairs is shown in Figures 6 and 7. We find that the Gemma family (Kamath et al., 2025) generally exhibits lower similarity to other models, while pairs within the same family tend to show higher similarity. Figure 8 also reports correlations across different CKA variants, where similarities computed with GSM8K and WikiText display relatively lower agreement.

## 4. Cooperation increases when similar models meet

Here, we define cooperation broadly to refer to behaviors in which agents align their actions or expectations to achieve mutually beneficial outcomes (Bowles & Gintis, 2003; Grice, 1975). Building on evidence that greater interbrain synchrony among humans is linked to increased cooperation, we test the hypothesis that model pairs with higher representational similarity will demonstrate increased cooperative behavior.

### 4.1. Game Settings & Analysis

We use four game settings that involve cooperation: word guessing (Clark & Wilkes-Gibbs, 1986), public goods (Hauert et al., 2006), divide-a-dollar (Kalai, 1977), and the Keynesian Beauty Contest (KBC) (Duffy & Nagel, 1997). The word-guessing game is a form of referential communication, in which a speaker provides a clue referring to a target word and a listener attempts to identify the target based on that clue. Referential communication has long been used to examine how players coordinate and collaborate (Clark & Wilkes-Gibbs, 1986). The latter three games have been widely adopted in economics and social science to study cooperative and coordination dynamics among humans. Moreover, these four games have commonly been employed to study the behaviors of AI models (Tang et al., 2024; Lai et al., 2024; Wu et al., 2024b; Piedrahita et al., 2025; Li & Shirado, 2025; Huang et al., 2024; Kim, 2025). Based on the prior literature, we adopt these four games in our study. The following presents a description of each

game rule along with associated outcome metrics, which capture the extent to which the two agents cooperated with one another during a game. Please refer to Appendix C for the prompts used in each game, along with examples.

- **Word Guessing:** In the game, one player chooses their own target word that begins with a given letter ("a" to "z") and provides a one-word hint to the other player. Here, the player is instructed to make the hint different from the target word. The other player should guess that secret word based on the hint and the given starting letter. Each round is one-shot and independent. We use the number of correct guesses over 26 rounds, one for each letter in the alphabet, as the outcome metric. Please note that it is not trivial to expect more similar models to perform better in this setting, because the game itself is asymmetric. The roles of the clue giver and the guesser are fundamentally distinct, and prior work in linguistics and cognitive science has shown that producing an informative cue and interpreting one are not mirror tasks (Hendriks & Koster, 2010; Hendriks, 2014; Mayol, 2018; Hendriks, 2016). This implies that even for a single individual, generating a hint and inferring the correct target from a self-produced hint rely on different underlying mechanisms. Consequently, success in the word-guessing game reflects cooperative behavior that goes beyond simple alignment or representational homogeneity.

- **Public Good:** The game repeats for five rounds. At the beginning of the game, each agent begins with $100 of their own money and decides how much to contribute to a public pot every round. After their contribution is collected into the public pot each round, the value increases by 30% and is evenly redistributed back to each agent. We use each agent's total asset value accumulated over five rounds as the outcome metric. In this game, a non-cooperative strategy is to free-ride by contributing nothing, which maximizes an individual's own total assets but substantially harms the collective welfare. Thus, the game captures how agents make a decision between cooperative and non-cooperative behavior across the five rounds.

- **Divide a Dollar:** The game repeats for five rounds. Each round, $1 is available, and players should demand how much of the $1 they want. If the total amount requested is not above $1, players receive the amount they requested. If the total amount requested exceeds $1, agents don't receive anything. We use each agent's total asset value accumulated over five rounds as the outcome metric. In this game, aggressively demanding larger amounts without considering the other player's request reduces the likelihood that either player receives anything. Thus, the game captures how well agents engage in cooperative decision-making over five rounds while taking the other player's choices into account.

- **KBC:** The game repeats for five rounds. At the beginning of each round, players choose a number between 0 and 100, guessing the closest number to $2/3$ of the average of the numbers from both agents. The score is based on how close their number is to $2/3$ of the average: $100 - |\text{their number} - 2/3 \times \text{average}|$. We use the total score of each player over the five rounds as the outcome metric. A higher score reflects a player's ability to anticipate the other player's choice. Thus, the game captures how effectively agents engage in recursive reasoning about the other player's reasoning process and selected numbers.

In all games except the word guessing game, where each round is one-shot and independent, players are shown the other's choice and reasoning at the end of each round. A higher game outcome value indicates stronger cooperation in that game. For example, in the word guessing game, performance depends on how accurately each agent guesses the other's secret words—reflecting their ability to interpret their partner and infer unknown information. In the public goods game, achieving high returns requires both cooperation and alignment: purely selfish strategies yield low payoffs, and exploitation due to misunderstanding also reduces outcomes.

We evaluate all 276 possible pairs of the 23 models listed in Table 1. Because the word guessing game is asymmetric, we consider ordered pairs, resulting in 529 model pairings. Each pair interacts across all four games, with temperature set to $0.7$[1] and at least 4 independent samples collected per pair for each game. The average game outcome for each model is presented in Figure 9 in Appendix D.

To analyze the relationship between representational similarity and interaction outcomes, we fit a mixed-effects linear regression model (Bates et al., 2015). In our experimental setup, using a simple linear regression or Pearson correlation would be inappropriate because these tests assume independent data points, whereas our setup produces multiple samples per model pair, and each model appears in multiple pairs. Mixed-effects regression is the standard approach for handling such non-independence (Brown, 2021). In particular, it allows us to account for variance attributable to individual models (e.g., differences in capability) by including model-specific random effects, thereby isolating the effect of representational similarity on interactive outcomes. Specifically, we estimate the following mixed-effects regression:

$$Y_{ij} = \alpha + \beta \cdot \text{CKA}_{ij} + u_i + v_j + \epsilon_{ij},$$

where $Y_{ij}$ is the interactive outcome of interest, and $\text{CKA}_{ij}$ is the similarity measure between models $i$ and $j$. The terms $u_i$ and $v_j$ represent random effects associated with models $i$ and $j$, respectively, where these terms capture

---

[1] For robustness, we rerun the experiments with a temperature of 0.3 and observe that the results remain consistent (see Appendix F).

unobserved heterogeneity at the level of model $i$ and model $j$, respectively. Lastly, $\epsilon_{ij} \sim N(0, \sigma_\epsilon^2)$ is an error term.

*To evaluate whether similarity predicts the interactive outcome, the key quantities are the estimated slope of $CKA_{ij}$ (i.e., $\beta$) and its statistical significance. We therefore report $\beta$ with its p-value throughout the paper.*

### 4.2. Does representational similarity predict cooperation?

Our results reveal that representational similarity can predict cooperative outcomes. In the word guessing game, correct guesses increase by approximately 88.2, 58.0, 63.3, and 59.4% (relative changes) for each unit increase in representational similarity (i.e., from 0 to 1) measured with WikiText, GSM8K, MATH, and TruthfulQA, respectively. In the public good game, each player's total asset value rises by 34.8, 32.4, 33.0, and 29.8% across the four probe datasets. For divide-a-dollar, asset values increase by 29.9, 15.3, 16.2, and 13.7%. Finally, in KBC, scores increase significantly but modestly—4.5, 4.7, 4.2, and 3.4%. All effects are found to be statistically significant (Figure 2). Among the four games, KBC shows the weakest effect. This is expected: the game has a unique Nash equilibrium in which both players always choose zero, which makes the optimal strategy fixed regardless of representational similarity. For instance, we observe that a certain model such as `GPT-OSS-20B` always chooses 0 regardless of the partner's decision. Nevertheless, even here, we observe a significant upward trend with increasing similarity.

The pattern persists across probe datasets, implying its generalizability. Moreover, we find no difference in effect size across datasets (please refer to Figure 2). This contrasts with a previous finding (Ciernik et al., 2024), which showed that the correspondence between representational similarity and task behavior depends on the dataset. The same trend holds across other CKA variants as well (see Appendix E.1).

## 5. Novelty decreases when similar models meet

Next, we examine whether representational similarity predicts novelty in collaborative generative tasks. For this purpose, we adapt four tasks—story writing, fictional biography, haiku composition, and vacation benefit brainstorming—from NoveltyBench (Zhang et al., 2025), a benchmark originally designed to evaluate an individual model's ability to produce high-quality and original ideas. Because NoveltyBench tasks are defined for single-agent settings, we extend them to the multi-agent case: each of the two models first generates a set of brainstorming ideas, after which each model produces a final output based on the combined brainstorms. The four generative tasks are as follows:

- **Story Writing:** Players brainstorm an outline of a story about a girl and her dog, then individually write a five-sentence story after reviewing the combined brainstorm.
- **Biography Writing:** Players brainstorm an outline for a short biography of a fictional person, then individually write a biography based on the combined brainstorm.
- **Haiku Writing:** Players brainstorm a plot for a haiku about a whale and a walnut tree, then individually compose a haiku after reviewing the combined brainstorm.
- **Vacation Benefit Brainstorming:** Players brainstorm possible benefits of going on vacation, then individually write the best aspect after reviewing the combined brainstorm.

As with the games involving cooperation, we evaluate all 276 pairs, using a temperature of 0.7. For each pair, we sample 10 generations in accordance with NoveltyBench. We also conduct mixed-effects regression to identify whether representational similarity can predict novelty.

Because novelty encompasses multiple dimensions, we evaluate it using several metrics: the number of distinct responses produced, the quality of those responses, and the extent to which outputs differ from those generated without interaction with another agent. The first two, response uniqueness and quality, are assessed using NoveltyBench's proposed metrics, while the last is measured as the mutual information between outputs produced through joint brainstorming and those generated without interaction. We describe the evaluation methodologies in more detail below.

### 5.1. Does representational similarity predict uniqueness and quality?

First, to assess response uniqueness and quality, we use the NoveltyBench evaluation pipeline using autoraters (Zhang et al., 2025). NoveltyBench defines the two measures, uniqueness and quality, over a set of samples. For uniqueness, the benchmark clusters 10 generations using a fine-tuned `deberta-v3-large` model according to content distinctiveness and then counts the number of clusters, which serves as the uniqueness metric. A higher cluster count indicates that models are able to generate more diverse ideas. For response quality, the benchmark relies on `Skywork-Reward-Gemma-2-27B-v0.2` (Liu et al., 2024), with outputs rescaled to a $1 - 10$ range for a more interpretable score.[2]

As shown in Figure 3, response uniqueness decreases consistently with increasing representational similarity across all tasks and probe datasets. The effect is strongest in haiku composition (coeff $= -3.425$, CI $= [-4.803, -2.047]$, $p < .001$). By contrast, response quality shows no systematic trend with similarity. Fictional biography and haiku tasks exhibit nonsignificant negative slopes of similarity

---

[2]None of the helper models used in this section are reused as players in the games.

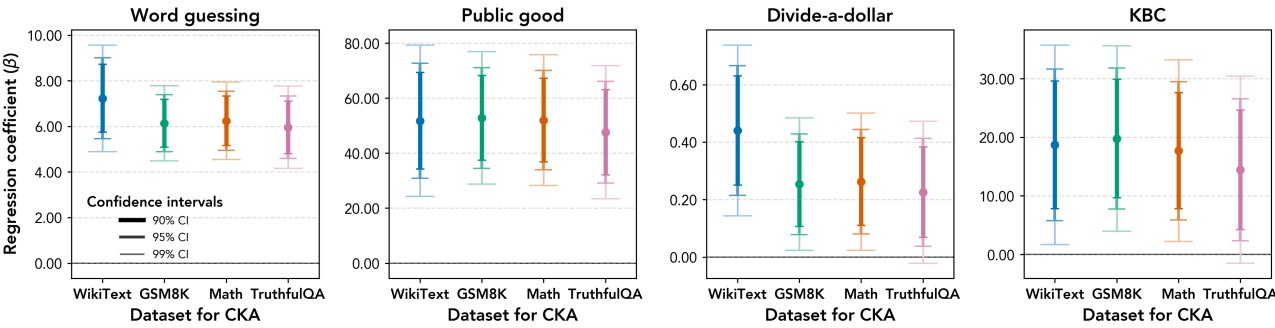

*Figure 2.* Regression coefficient of representational similarity on game outcomes. Error bars denote 90%, 95%, and 99% confidence intervals. Across games and datasets, the graphs show a positive effect of similarity on outcomes, with WikiText-based similarity exhibiting the strongest effect.

(coeff $= -0.397$, $p = .456$ for biography; coeff $= -0.115$, $p = .724$ for haiku), while story writing and vacation tasks show a nonsignificant positive slope (coeff $= 0.279$, $p = .420$ for story; coeff $= 0.039$, $p = .901$ for vacation). This implies that interaction with dissimilar models tends to generate more diverse responses, without reducing quality.

## 5.2. Does representational similarity predict mutual information?

We next examine whether representational similarity has a significant effect on output novelty—specifically, how far a model's responses generated after joint brainstorming ("amalgam response") deviate from the model's outputs conditioned only on its individual brainstorm ("individual response"). To capture this, we compare amalgam and individual responses using mutual information (Kraskov et al., 2004), which quantifies how much information individual responses share with those produced in the joint setting. Such information-theoretic approaches have recently been applied to investigate textual characteristics (e.g., information distribution across paragraphs) (Venkatraman et al., 2023; Clark et al., 2023; Mu et al., 2025; Chidichimo et al., 2025).

To calculate mutual information, we follow the method from Mu et al. (2025). Formally, let $S_A$ denote an amalgam response and $S_I$ denote an individual response. We compute the mutual information $I(S_A; S_I)$ as $H_\theta(S_A) - H_\theta(S_A \mid S_I)$. $H_\theta(S_A)$ denotes the total information content of the amalgam response, and $H_\theta(S_A \mid S_I)$ denotes the residual information of the amalgam response given the individual response, both measured under a reference language model with parameters $\theta$. To calculate $H_\theta(S_A)$, we sum the cross-entropy over all tokens in the amalgam response under the model with parameters $\theta$. The cross-entropy of a token quantifies the model's prediction error for that token given its preceding context, thereby reflecting its uncertainty. Similarly, $H_\theta(S_A \mid S_I)$ is computed by summing the cross-entropy of each token in the amalgam response when the individual response is prefixed to the amalgam response. A smaller

difference between $H_\theta(S_A)$ and $H_\theta(S_A \mid S_I)$ indicates that the amalgam response deviates more from the individual response, thereby reflecting higher novelty. Following Mu et al. (2025), we use `Llama-3.1-8B-Instruct` as the reference model.[3]

Our analysis shows a significant positive effect of representational similarity on mutual information, which suggests that interactions between more similar models generate less novel outputs with respect to the individual model responses. The trend appears across all tasks and probe datasets (Figure 4). In particular, the haiku task exhibits the strongest effect of representational similarity on mutual information (coeff $= 1.310$, CI $= [1.034, 1.585]$, $p < .001$).

## 6. Why does the trend appear?

So far, we have identified a strong trend between representational similarity and interactive behaviors of models. This naturally raises the question of why such a trend emerges. In this section, we test several hypotheses regarding what drives the trend.

**Confounding Effects of Behavioral Similarity.** Models with higher representational similarity may behave more similarly (e.g., bid the same amount in divide-a-dollar), and this behavioral similarity might have led directly to greater measured cooperation. To test this, we conducted a mixed-effects regression controlling for behavioral differences in the public goods, divide-a-dollar, and KBC games. This allows us to isolate the effect of representational similarity from behavioral similarity. Because these games instruct

---

[3]They select the model under the requirement that the mutual information values satisfy symmetry and non-negativity. For robustness, we additionally compute mutual information with the base model `Llama-3.1-8B`, identified by Mu et al. (2025) as a strong alternative reference model. Results, shown in Appendix E.4, continue to show a significant association with representational similarity. One might further suspect a same-family bias when the reference model is used to evaluate Llama models. To address this, we analyze model pairs excluding the Llama family and report the results in Appendix E.5. The results still show the significant effect of similarity.

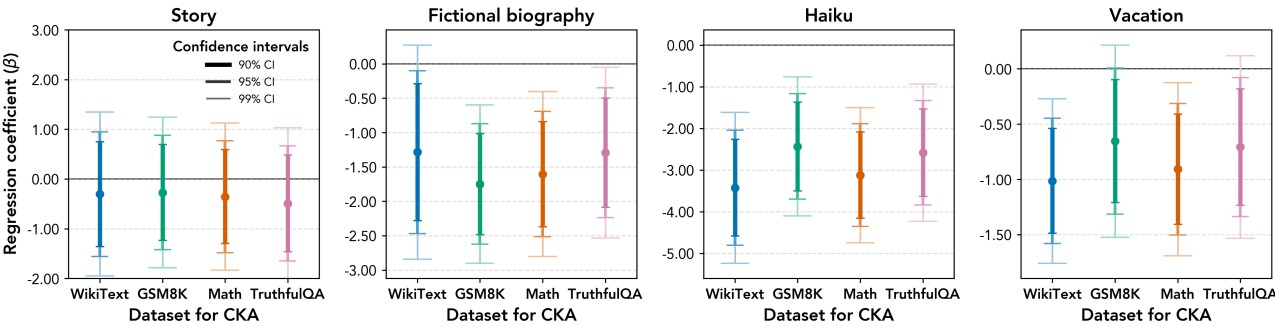

*Figure 3.* Regression coefficient of representational similarity on response uniqueness. Error bars denote 90%, 95%, and 99% confidence intervals. The graphs reveal a consistent downward trend: as models become more similar, response uniqueness declines. The strongest effect is observed in the haiku task.

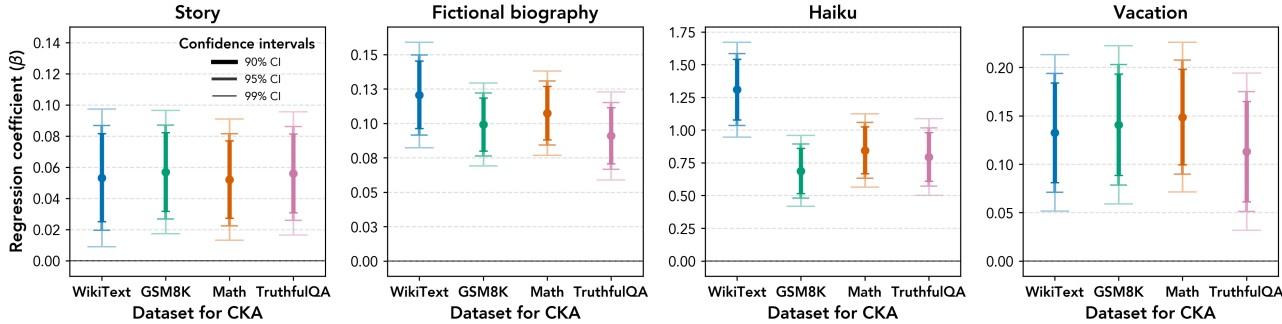

*Figure 4.* Regression coefficients of representational similarity on mutual information. Error bars denote 90%, 95%, and 99% confidence intervals. The graphs reveal a decreasing trend in novelty with increasing similarity: as models become more similar, the shared information between the amalgam response (i.e., response generated after joint brainstorming) and the individual response (i.e., response generated after solo brainstorming) increases.

models to make numerical choices, it is straightforward to estimate the behavioral difference as the absolute gap between the two models' choices.

Our analysis shows that the behavioral difference alone cannot explain the observed trends. In both the public good and divide-a-dollar games, representational similarity remains a significant predictor, while behavioral difference is insignificant (coeff. rep. sim. $= 52.118$, $p < .001$, coeff. beh. diff. $= -0.036$, $p = .086$ for public good; coeff. rep. sim. $= 0.435$, $p < .001$, coeff. beh. diff. $= -0.020$, $p = .281$ for divide-a-dollar). This suggests that behavioral similarity is not what drives the trend. By contrast, in the KBC game, behavioral difference shows a significant effect, while representational similarity does not (coeff. rep. sim. $= 9.024$, $p = .178$, coeff. beh. diff. $= -0.327$, $p < .001$). As discussed in Section 4, KBC has a unique Nash equilibrium in which both players choose 0, which leads to convergence in choices. This structural property of the game likely explains why behavioral difference dominates in this case.

**Confounding Effects of Performance Disparity.** Another potential claim is that the observed trends might simply be a byproduct of performance disparities between the two models. To assess this possibility, we collected models' MMLU performance scores (Hendrycks et al., 2020), a rep-

resentative benchmark for evaluating the general knowledge and problem-solving abilities of LLMs. We then conducted a mixed-effects regression that controls for differences in MMLU scores. The results show that the main trends remain robust even after accounting for performance disparity (Appendix E.6). This indicates that the positive relationship between representational similarity and cooperation—and the negative relationship with novelty—cannot be explained merely by differences in overall model performance.

**Factors Underlying Representational Similarity.** Representational similarity is influenced by several architectural and design-related components, including whether two models are identical, belong to the same model family, share the same tokenizer, or differ in size. Any of these factors could potentially have driven the observed behavioral trends by influencing similarity. To investigate this, we conducted a mixed-effects regression controlling for four key factors: (1) identical model pairing, (2) within-family pairing, (3) shared tokenizer, and (4) model size difference. In this analysis, all predictors were rescaled to $[0, 1]$ to allow comparison of effect sizes of predictors. Outcome variables were converted to $z-$scores, and we included game category as a control to account for systematic differences across games. If the four factors were mainly responsible for the trend, we would

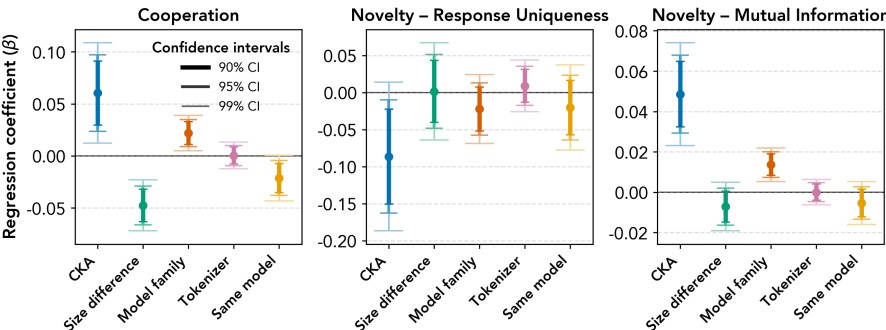

*Figure 5.* Regression coefficients of representational similarity (CKA with WikiText) and four other factors across cooperation and novelty games. To enable comparison across predictors, all variables were rescaled to $[0, 1]$ in the regression. The graphs show that representational similarity is the strongest predictor.

expect representational similarity to lose significance once they were controlled for, while the factors themselves would show significant effects.

Our analysis finds that representational similarity is the strongest predictor on cooperation and novelty, compared to the four factors (Figure 5). In cooperation games, all predictors except tokenizer are significant, with representational similarity showing the largest effect (coeff $= 0.060$, $p = .001$). For response uniqueness, none of the four factors is significant, while representational similarity shows a significant effect (coeff $= -0.087$, $p = .026$). For mutual information, only representational similarity and within-family are significant, with similarity again showing the stronger impact (coeff $= 0.049$, $p < .001$). Taken together, these findings suggest that the four examined factors do not fully explain the trend. Instead, representational similarity itself—likely influenced by deeper, unmeasured aspects of model design and training—remains a central factor underlying the observed interaction patterns. Currently, there is no direct metric to quantify many latent design and training factors (e.g., training data overlap). Our finding suggests that representational similarity can serve as a powerful and practical indicator of these otherwise inaccessible underlying properties.

**Which Aspects of Representational Similarity Drive the Trend?** Lastly, we investigate which aspects of similarity primarily drive the observed trend. To this end, we divided each model's layers into three groups—early, middle, and late—and computed CKA scores for each group (e.g., similarity calculated only between the early layers of the two models). We then ran separate mixed-effects regressions for each group. Appendix E.7 presents the results. Overall, the early one-third of layers consistently exhibit the strongest effects on both cooperation and novelty. The same pattern holds for temperature 0.3 (Appendix F.4). This suggests that shared basic lexical–semantic grounding plays a central role in increased cooperation, whereas divergence in these foundational representations is linked to greater collective novelty.

## 7. Future Directions and Open Questions

Existing multi-agent system designs often rely on a single model without exploring the optimal combination of models (Lai et al., 2024; Park et al., 2023; Xie et al., 2024; Zhou et al., 2023; Wu et al., 2024a; Ishibashi & Nishimura, 2024). Our findings suggest that which models are combined has a significant effect on their interactions. In neuroscience and social science, researchers have long studied the nature of human social dynamics (Parkinson et al., 2018; Thornton & Mitchell, 2017; Shen et al., 2025b; Reinero et al., 2021; Page et al., 2019; Paulus, 2000). We argue that such efforts should also be made in the AI community, and our experiments provide an initial step in that direction.

The relationship between representational similarity and model interaction is likely context-dependent. We already observed that the effect size of similarity varies across games. For instance, in KBC, which has a unique Nash equilibrium, the link between similarity and interaction becomes weaker. Other evidence is also found in neuroscience and social science. Some studies show that diversity can foster cooperation (Santos et al., 2008; 2012; Wang et al., 2025), and certain creativity research suggests that greater similarity can yield higher originality (Koo et al., 2024; Bastian et al., 2018; Miura & Hida, 2004). These findings imply that there might be no universal relationship between similarity and interactive dynamics. Understanding when the trend emerges, when it disappears, and when it reverses will require further research.

Another direction is to investigate the mechanisms underlying these trends. In this work, we used CKA as our measure of representational similarity. However, metrics like CKA capture only limited aspects of representational spaces, making it difficult to pinpoint which specific features of representations drive the trends. Future work can examine this at the neuron level—e.g., which neurons are preferentially activated when a model interacts with another model of higher representational similarity. Such analyses could enable us to deliberately steer cooperation or collective novelty through targeted activation steering.

## Impact Statement

This paper follows the ICML Code of Ethics. Our goal is to contribute to the design of multi-agent systems, which we believe can benefit society by enabling more effective and cooperative AI applications. We also used LLMs for typo and grammar checks during manuscript preparation. To support reproducibility, we will publicly release all datasets, code, and evaluation scripts used in this work upon acceptance.

## Acknowledgements

This work is supported by Schmidt Sciences and the National GEM Consortium. The authors thank Tiwalayo Eisape, Andrew Lampinen, and Syrielle Montariol for their valuable feedback on the early drafts of the manuscript.

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

# A. Model list

*Table 1.* Full list of models involved in the interaction experiments

| Family | Model Name | Checkpoint / Repo | Size | Tokenizer | Reference |
|---|---|---|---|---|---|
| **Qwen** | Qwen2.5-3B-Instruct | `Qwen/Qwen2.5-3B-Instruct` | 3.09B | BPE | (Qwen Team, 2024) |
| | Qwen2.5-7B-Instruct | `Qwen/Qwen2.5-7B-Instruct` | 7.61B | BPE | (Qwen Team, 2024) |
| | Qwen2.5-14B-Instruct | `Qwen/Qwen2.5-14B-Instruct` | 14.7B | BPE | (Qwen Team, 2024) |
| | Qwen2.5-72B-Instruct | `Qwen/Qwen2.5-72B-Instruct` | 72.7B | BPE | (Qwen Team, 2024) |
| **Llama** | Llama-3.2-3B-Instruct | `meta-llama/Llama-3.2-3B-Instruct` | 3.21B | tiktoken | (Grattafiori et al., 2024) |
| | Llama-3.2-11B-Vision-Instruct | `meta-llama/Llama-3.2-11B-Vision-Instruct` | 10.6B | tiktoken | (Grattafiori et al., 2024) |
| | Llama-3.3-70B-Instruct | `meta-llama/Llama-3.3-70B-Instruct` | 70.6B | tiktoken | (Grattafiori et al., 2024) |
| **Gemma** | Gemma-3-1B-IT | `google/gemma-3-1b-it` | 1.0B | SentencePiece | (Kamath et al., 2025) |
| | Gemma-3-4B-IT | `google/gemma-3-4b-it` | 4.0B | SentencePiece | (Kamath et al., 2025) |
| | Gemma-3-12B-IT | `google/gemma-3-12b-it` | 12.2B | SentencePiece | (Kamath et al., 2025) |
| | Gemma-3-27B-IT | `google/gemma-3-27b-it` | 27.0B | SentencePiece | (Kamath et al., 2025) |
| **Falcon** | Falcon3-3B-Instruct | `tiiuae/Falcon3-3B-Instruct` | 3.23B | BPE | (Almazrouei et al., 2023) |
| | Falcon3-7B-Instruct | `tiiuae/Falcon3-7B-Instruct` | 7.46B | BPE | (Almazrouei et al., 2023) |
| | Falcon3-10B-Instruct | `tiiuae/Falcon3-10B-Instruct` | 10.3B | BPE | (Almazrouei et al., 2023) |
| **Phi** | Phi-3.5-mini-instruct | `Lexius/Phi-3.5-mini-instruct` | 3.8B | SentencePiece | (Abdin et al., 2024a) |
| | Phi-3-medium-128k-instruct | `microsoft/Phi-3-medium-128k-instruct` | 14B | SentencePiece | (Abdin et al., 2024a) |
| | Phi-4-mini-instruct | `microsoft/Phi-4-mini-instruct` | 3.8B | tiktoken | (Abdin et al., 2024b) |
| | Phi-4 | `microsoft/phi-4` | 14.7B | tiktoken | (Abdin et al., 2024b) |
| **Mistral** | Mistral-Nemo-Instruct-2407 | `mistralai/Mistral-Nemo-Instruct-2` | 12.2B | tekken | (Jiang et al., 2024) |
| | Ministral-8B-Instruct-2410 | `mistralai/Ministral-8B-Instruct-2410` | 8.02B | tekken | (Jiang et al., 2024) |
| **OpenAI** | GPT-OSS-20B | `openai/gpt-oss-20b` | 21.5B | o200k_harmony | (OpenAI et al., 2025) |
| **OLMo** | OLMo-2-1B-Instruct | `allenai/OLMo-2-0425-1B-Instruct` | 1.48B | cl100k | (OLMo et al., 2025) |
| | OLMo-2-13B-Instruct | `allenai/OLMo-2-1124-13B-Instruc` | 13.7B | cl100k | (OLMo et al., 2025) |

# B. Representational similarity of model pairs

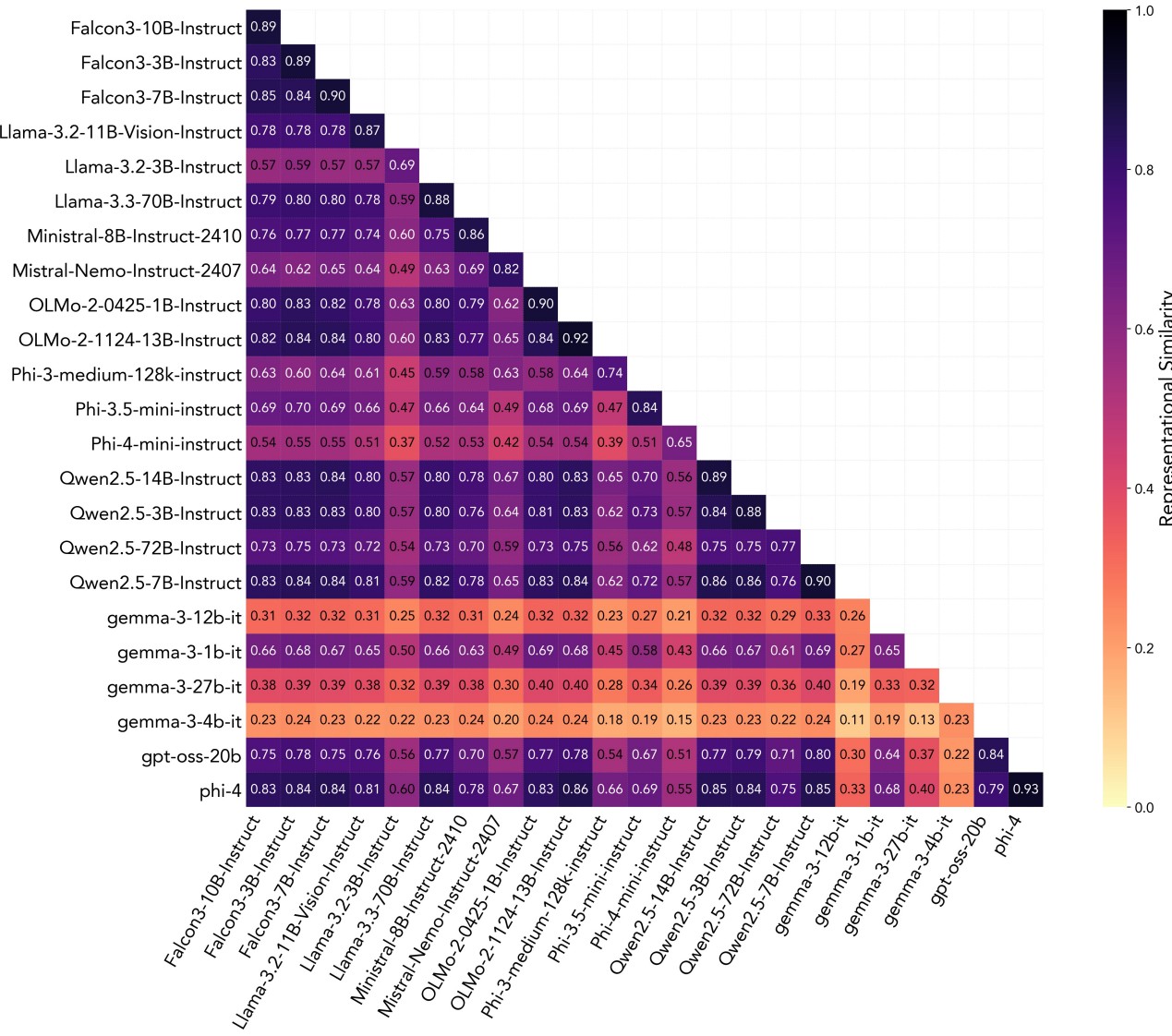

*Figure 6.* Representational similarity of model pairs using WikiText, where CKA scores across layers are aggregated into a global average.

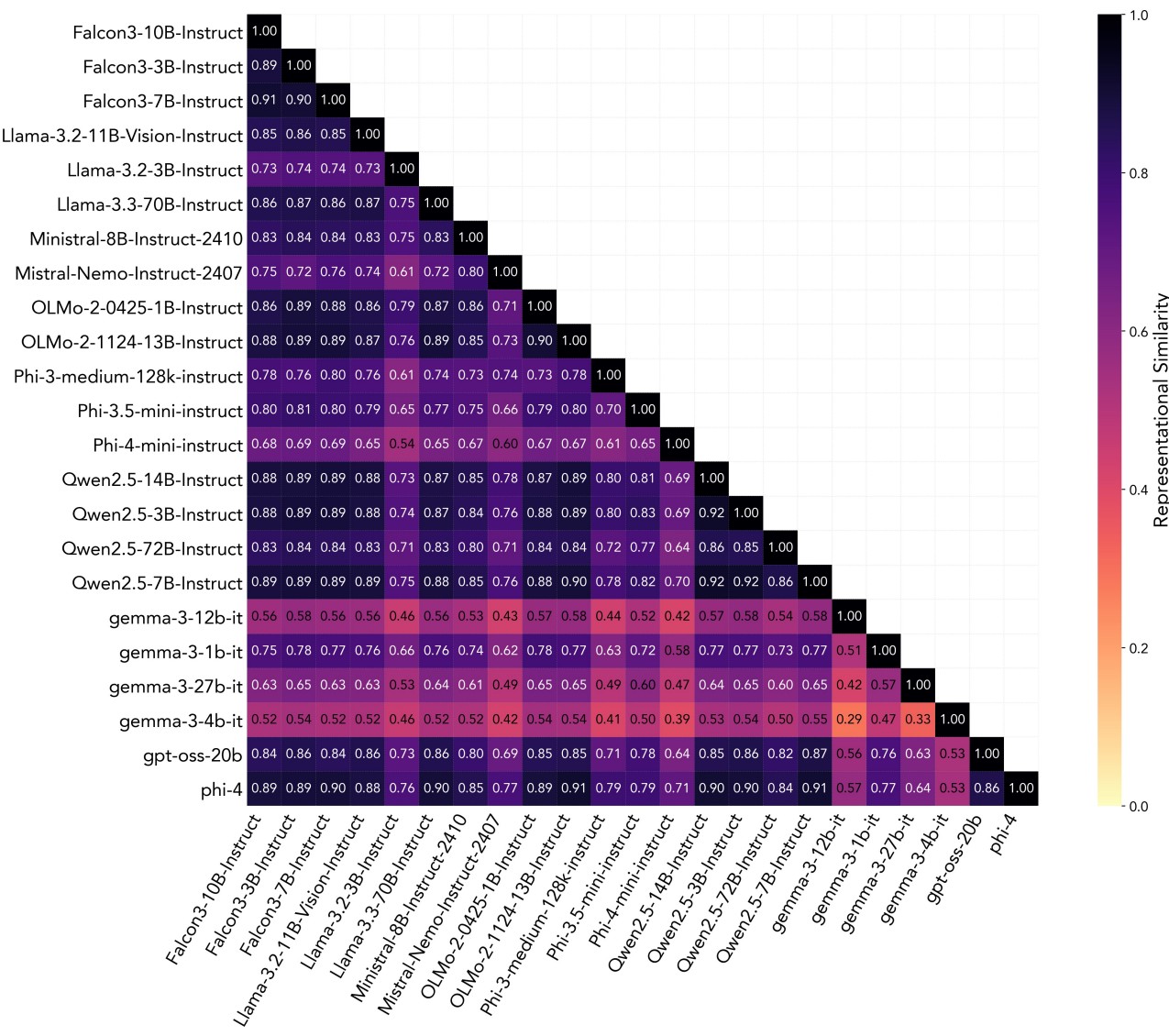

*Figure 7.* Representational similarity of model pairs using WikiText, where CKA scores across layers are aggregated into a maximum-aligned average.

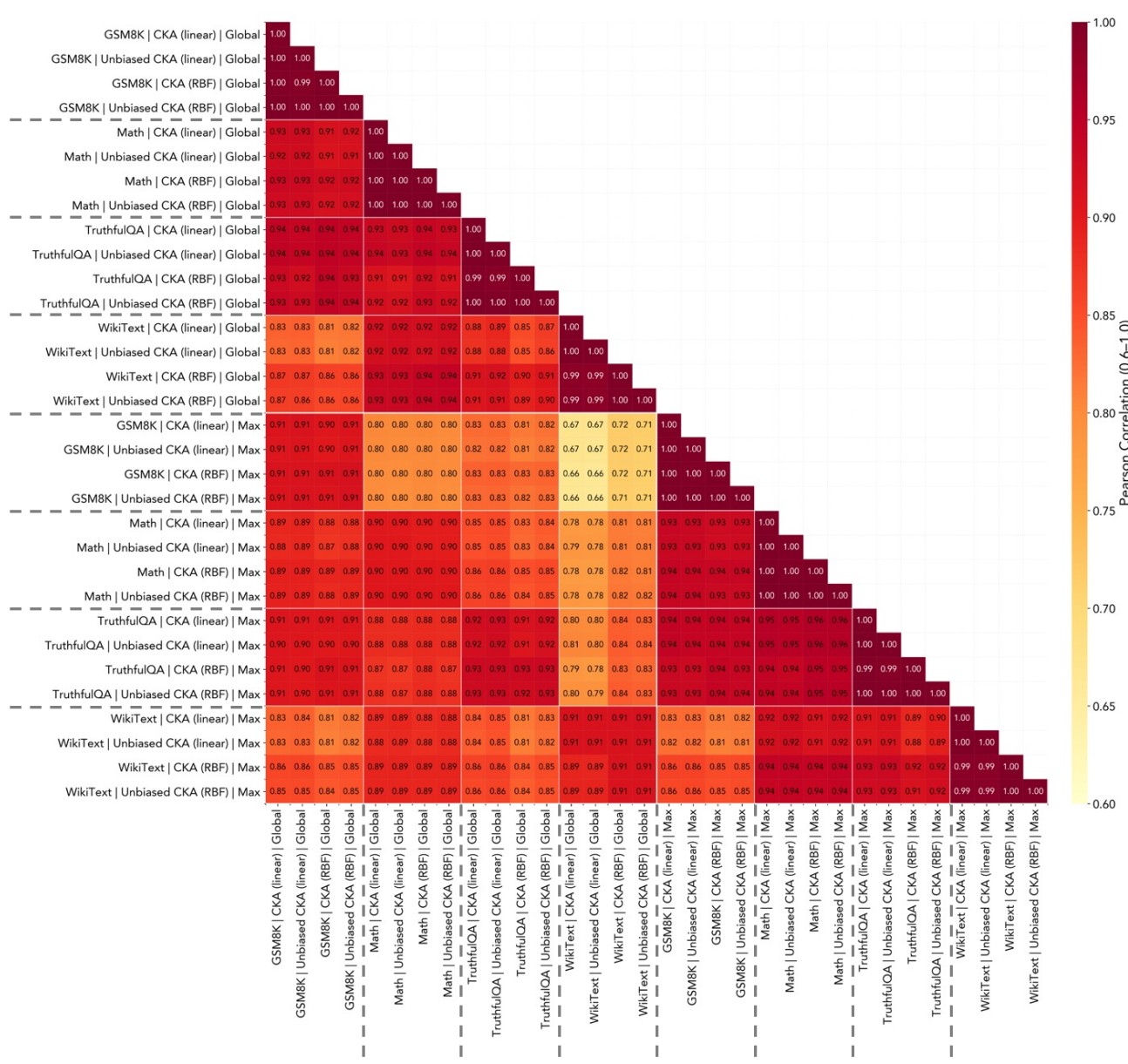

*Figure 8.* Correlation between different CKA scores

# C. Game prompts and examples

## C.1. Games for cooperation

### C.1.1. WORD GUESSING

We first instruct an agent to provide a secret word and a clue referring to this, using the following prompts:

- `Please think of a word that starts with '{an alphabet character}.' Please provide that word.`
- `Please provide a one-word hint so that others can guess your word: {The player's target word}. The hint word should be different from your word. Please provide the hint word.`

Then, we ask the other to guess the secret word based on the hint, using the following prompt:

- `You must guess a word that starts with "{an alphabet character}." Here is a hint: "{the hint word}." Respond with the single word starting with "{an alphabet character}" you think is being defined.`

The below is a game example between `Falcon3-10B-Instruct` and `gpt-oss-20b`. In the example, the pair received no points because `gpt-oss-20b` failed to guess the secret word chosen by `Falcon3-10B-Instruct`.

> **User:** Please think of a word that starts with 'm.' Please provide that word.

> **Falcon3-10B-Instruct:** Moon

> **User:** Please provide a one-word hint so that others can guess your word: Moon. The hint word should be different from your word. Please provide the hint word.

> **Falcon3-10B-Instruct:** Night

> **User:** You must guess a word that starts with "m." Here is a hint: "Night." Respond with the single word starting with "m" you think is being defined.

> **gpt-oss-20b:** Midnight

The following is a game example between `gpt-oss-20b` and `gpt-oss-20b`. In the example, the pair successfully scored.

> **User:** Please think of a word that starts with 'm.' Please provide that word.

> **gpt-oss-20b:** mystery

> **User:** Please provide a one-word hint so that others can guess your word: mystery. The hint word should be different from your word. Please provide the hint word.

```
gpt-oss-20b:  puzzle
```

```
User:  You must guess a word that starts with "m." Here is a hint:  "puzzle."
Respond with the single word starting with "m" you think is being defined.
```

```
gpt-oss-20b:  mystery
```

### C.1.2. PUBLIC GOOD

We first present the rules of the public good game and ask each player to choose their contribution amount. After both players make their decisions, we disclose the other player's reasoning and contribution, along with the resulting payout to each agent. The following prompts are used, and the game repeats for five rounds.

```
• You will play a game with another one.  Here's how it works:

  1.  You begin with $100.
  2.  Decide how much of your $100 you wish to contribute to a
  public pot.  You can contribute any amount between $0 and $100.
  3.  After both players have made their contributions, the total amount in the
  public pot will be multiplied by 1.3.
  4.  The multiplied amount in the public pot will then be divided equally
  between you and the other player, regardless of how much each of you
  contributed.
  5.  After the redistribution, you will keep whatever amount you receive.

  Now, please decide how much you'll contribute to the public pot out of $100.
  You should first say a reason for your decision and then your contribution
  amount.
```

```
• The contributions have now been collected!  The other player decided its
  contribution as follows:
  "{the other player's reasoning}"

  Therefore, after applying the 1.3x multiplication, the total value in the
  public pot is $the amount in the pot.  Therefore, you receive half of this
  amount, which is ${0.5× the amount in the pot}.  As a result, you earn ${the
  net profit of the player}.
```

We present an example interaction between `Llama-3.3-70B-Instruct` and `gemma-3-12b-it` for two rounds. We illustrate the exchange from the perspective of `Llama-3.3-70B-Instruct`.

```
User:  You will play a game with another one.  Here's how it works:

1.  You begin with $100.
2.  Decide how much of your $100 you wish to contribute to a
public pot.  You can contribute any amount between $0 and $100.
3.  After both players have made their contributions, the
total amount in the public pot will be multiplied by 1.3.
4.  The multiplied amount in the public pot will then be divided equally
between you and the other player, regardless of how much each of you
contributed.
5.  After the redistribution, you will keep whatever amount you receive.

Now, please decide how much you'll contribute to the public pot out of $100.
```

> You should first say a reason for your decision and then your contribution
> amount.

**Llama-3.3-70B-Instruct:**  My reason for deciding the contribution amount
is that I want to maximize my potential gain while also considering the
possibility that the other player may not contribute much.  If I contribute
a large amount and the other player contributes little or nothing, I risk
losing a significant portion of my initial $100.  On the other hand, if I
contribute too little, I may miss out on the opportunity to gain more from
the multiplied public pot.  Given that the public pot will be multiplied by
1.3, I want to contribute enough to take advantage of this multiplier, but
not so much that I put myself at a significant disadvantage if the other
player doesn't contribute much.

My contribution to the public pot will be $50.

**User:**  The contributions have now been collected!  The other player decided
its contribution as follows:

"Okay, this is a classic game theory scenario, often called a "Public Goods
Game." Here's my reasoning and my contribution:

**Reasoning:**  The optimal strategy in a purely rational, self-interested
perspective would be to contribute nothing.  Here's why:

– **Shared Benefit:**  Regardless of my contribution, I receive 50% of the
multiplied pot.

– **No Direct Reward for Contributing:**  My individual contribution doesn't
directly increase my share.  Someone else could contribute a large amount,
and I benefit from it without having risked any of my own money.

– **Risk Aversion:**  Contributing money is a risk.  If the other player
contributes nothing, I've lost that money and gained nothing.

However, in reality, people often deviate from this purely rational strategy
due to factors like fairness, reciprocity, and a desire to cooperate.  If the
other player contributes, my contribution will increase the total pot, and
therefore, increase the amount we *both* receive.

Given that I don't know the other player's strategy, I'll try to strike a
balance between maximizing my potential gain and avoiding being exploited.  A
small contribution signals a willingness to cooperate, but doesn't expose me
to too much risk if the other player is purely self-interested.

**Contribution:**  I will contribute **$20**.

Let's see what the other player does!"

Therefore, after applying the 1.3x multiplication, the total value in the
public pot is $91.0.  Therefore, you receive half of this amount, which is
$45.5.  As a result, you lose $4.5.

**Llama-3.3-70B-Instruct:**  It seems that my contribution of $50 was not matched
by the other player, who contributed $20.  As a result, I ended up losing
$4.5.

In hindsight, I can see that the other player's reasoning was sound, and

their contribution of $20 was a strategic move to balance cooperation and self-interest. By contributing $50, I took on more risk than necessary, and it didn't pay off.

If I were to play this game again, I would consider contributing a smaller amount, similar to the other player's $20, to minimize my risk while still signaling a willingness to cooperate. Perhaps a contribution of $30 or $40 would have been a better balance between maximizing my potential gain and avoiding exploitation.

Thanks for the game! Would you like to play again?

---

**User:** Now you have $95.5. The same game repeats. Please decide how much you'll contribute to the public pot out of $95.5. You should first say a reason for your decision and then your contribution amount.

---

**Llama-3.3-70B-Instruct:** My reason for deciding the contribution amount this time is that I've learned from the previous round. I realize that contributing a large amount, like $50, may not be the best strategy, especially if the other player is not willing to contribute equally. Given that the other player contributed $20 in the previous round, I expect them to contribute a similar amount this time.

I also consider that the game is repeated, which means that the other player may be more likely to cooperate if they see that I'm willing to cooperate as well. However, I still want to be cautious and not risk too much of my remaining balance.

Taking into account the 1.3x multiplication factor, I want to contribute an amount that will give me a decent return if the other player contributes a similar amount to what they did last time. I'll aim to find a balance between cooperation and self-interest.

My contribution to the public pot will be $25. This amount is lower than my initial contribution of $50, but still shows that I'm willing to cooperate and take some risk to potentially gain more.

---

**User:** The contributions have now been collected! The other player decided its contribution as follows:

"Okay, let's play round two! Having experienced the outcome of the first round, I'm adjusting my strategy.

**Reasoning:** The first round revealed a key piece of information: the other player is willing to contribute a substantial amount

{ $50. This changes the game significantly. Now, I have a better understanding of their likely behavior.

- **Reciprocity & Encouraging Cooperation:** Contributing more now signals a willingness to reciprocate their previous investment and encourages them to continue contributing in future rounds (if this were a multi-round game).

- **Increased Potential Gain:** Given their likely contribution, a higher contribution from my side will result in a larger total pot, and therefore, a larger share for both of us. The 1.3 multiplier makes this particularly attractive.

– **Mitigating Risk**: While there's always a risk they'll contribute nothing, the evidence from round one suggests that's unlikely.

I want to be more aggressive in pursuing the potential gains, but I also want to remain reasonable. I don't want to contribute so much that I leave myself vulnerable if they drastically change their strategy.

**Contribution**: I will contribute **$45**.

Hopefully, the other player will continue to cooperate and we can both benefit from a larger pot this time around!"

Therefore, after applying the 1.3x multiplication, the total value in the public pot is $91.0. Therefore, you receive half of this amount, which is $45.5. As a result, you earn $20.5.

---

**Llama-3.3-70B-Instruct**: It seems that my contribution of $25 was complemented by the other player's contribution of $45, resulting in a total pot of $70. After the 1.3x multiplication, the pot became $91, and I received half of that, which is $45.5. Since I contributed $25, my net gain is $20.5.

I'm pleased to see that the other player adjusted their strategy to contribute more, which suggests that they are indeed willing to cooperate and reciprocate. This cooperation led to a better outcome for both of us, as we both earned a positive amount.

Given that the other player contributed $45, which is a significant amount, I'm inclined to believe that they will continue to cooperate in future rounds. Therefore, in the next round, I may consider contributing an even larger amount to take advantage of the potential gains from our cooperation.

For the next round, I'll aim to contribute an amount that will encourage the other player to continue cooperating while also maximizing my potential gain. Perhaps a contribution of $35 or $40 would be a good starting point.

Let's play the next round!

---

### C.1.3. DIVIDE-A-DOLLAR

We use the following prompts to instruct models to make a decision and then reveal the other's reasoning and decision. The game repeats for five rounds.

- You will play a game with another one. Here's how it works:

  1. There is $1 available. You must decide how much of this $1 you want. The other player will decide for themselves independently.

  2. If the sum of the two demands is less than or equal to $1, each player receives exactly the amount they demanded. If the sum exceeds $1, both players receive nothing.

  3. Your goal is to maximize the amount of money you receive.

  Now, please state your demand. First give a sentence explaining your decision, then state the amount you demand.

- Both players have decided how much they want. The other player decided as follows: "{the other player's reasoning}"

```
Therefore, you receive {the player's profit for the round} this round.  As a
result, your total asset value is {the player's total asset value}.  Please
wait for the next round of the game!
```

We present an example interaction between `Phi-3-medium-128k-instruct` and `gemma-3-1b-it` for two rounds. We illustrate the exchange from the perspective of `Phi-3-medium-128k-instruct`.

---

**User:**  You will play a game with another one.  Here's how it works:

1.  There is $1 available.  You must decide how much of this $1 you want. The other player will decide for themselves independently.

2.  If the sum of the two demands is less than or equal to $1, each player receives exactly the amount they demanded.  If the sum exceeds $1, both players receive nothing.

3.  Your goal is to maximize the amount of money you receive.

Now, please state your demand.  First give a sentence explaining your decision, then state the amount you demand.

---

**Phi-3-medium-128k-instruct:**  In order to maximize my potential gain, I should consider that if I ask for the entire dollar, I risk the other player also demanding the entire dollar, leading to both of us getting nothing.  To reduce the chance of this happening, I will demand less than $1 but enough to ensure I get a positive outcome while leaving room for the other player to claim some of the remaining amount.  Therefore, I demand $0.40.

---

**User:**  Both players have decided how much they want.  The other player decided as follows:

"I'll demand $0.50."

Therefore, you receive $0.4 this round.  As a result, your total asset value is 0.4.  Please wait for the next round of the game!

---

**Phi-3-medium-128k-instruct:**  Given that the other player demanded $0.50 and the total available was $1, the sum of our demands ($0.40 + $0.50 = $0.90) did not exceed the limit.  So, both players receive exactly what they demanded.  My total asset value is now $0.4.  I'm prepared to adapt my strategy in the next round based on the outcome of this round.

---

**User:**  The same game repeats.  $1 is available again.  Decide how much of that $1 you want.  First give a sentence explaining your decision, then state the amount you demand.

---

**Phi-3-medium-128k-instruct:**  Based on the outcome of the previous round, where the other player demanded $0.50 and I received $0.40, I should consider a similar strategy for the next round.  However, to increase the chances of maximizing my gain, I should slightly increase my demand, while still not exceeding the total available amount.  Therefore, I demand $0.45.

---

> **User:** Both players have decided how much they want. The other player decided as follows: "I'll demand $0.75."
>
> Therefore, you receive $0 this round. As a result, your total asset value is 0.4. Please wait for the next round of the game!

> **Phi-3-medium-128k-instruct:** Given that the other player demanded $0.75, the total sum of our demands ($0.45 + $0.75 = $1.20) exceeded the total available amount. Therefore, both players received nothing this round. My total asset value remains at 0.4. This outcome highlights the risk in my decision strategy. In the next round, I should reassess my strategy to avoid such outcomes and to maximize my gain.

### C.1.4. KBC

We first present the rules of the KBC game and ask each player to choose their number. After both players make their decisions, we disclose the other player's reasoning and selected number. The following prompts are used, and the game repeats for five rounds.

- You will play a game with another one. Here's how it works:

  1. You must choose a number from 0 to 100.

  2. After both players choose, we take the average of the two numbers and then calculate two-thirds of that average. You will receive a score based on how close your chosen number is to this value. Specifically, your score is 100−|your guess−2/3*average|. In short, the closer your number is to 2/3 of the average, the higher your score.

  3. Your goal is to get the highest score.

  Now, please choose a number from 0 to 100. First give a sentence explaining your decision, and then state your number.

- Both players have chosen their numbers. The other player decided its number as follows:

  "{the other player's reasoning and decision}"

  Therefore, the two-thirds of the average of your number and the other player's number is {the average of the numbers}. As a result, you receive a score of {the obtained score of the player} out of 100 this round, and the other player receives {the obtained score of the other player}. Your total score is {the player's total score}, while the other player's total score is {the other player's total score}.

Here is an interaction example between `Llama-3.2-11B-Vision-Instruct` and `gpt-oss-20b` for two rounds. We illustrate the exchange from the perspective of `Llama-3.2-11B-Vision-Instruct`.

> **User:** You will play a game with another one. Here's how it works:
>
> 1. You must choose a number from 0 to 100.
>
> 2. After both players choose, we take the average of the two numbers and then calculate two-thirds of that average. You will receive a score based on how close your chosen number is to this value. Specifically, your score is 100−|your guess−2/3*average|. In short, the closer your number is to 2/3 of

the average, the higher your score.

3. Your goal is to get the highest score.

Now, please choose a number from 0 to 100. First give a sentence explaining your decision, and then state your number.

**Llama-3.2-11B-Vision-Instruct:** To maximize my score, I want to choose a number that is as close as possible to the average of the two numbers, so I'll opt for a number that is roughly in the middle of the range.

My chosen number is 50.

**User:** Both players have chosen their numbers. The other player decided its number as follows:

"analysisWe need to respond as ChatGPT, playing the game with another participant. We should make a strategic choice. The game is like "guess the number" or "two-thirds of average" puzzle. The rational equilibrium is 0. But we can also explain. The instruction: "First give a sentence explaining your decision, and then state your number." So we should provide a sentence explaining the decision, then the number. We can choose 0 or a small number. Since rational players, they'd pick 0. But maybe we can mention some heuristics or we can choose 0. The question: "You will play a game with another one." So we are playing with another player. The rational equilibrium is 0. So we choose 0. But maybe we can add some twist: "I choose 0 because it's the Nash equilibrium after infinite iterated elimination of dominated strategies." Good. Then number 0. Or we could choose 0. It's allowed. So we produce a sentence and then 0.

But we can also choose a small number e.g., 1. But the rational equilibrium is 0. So we choose 0. Let's do that. Provide the sentence: "I choose 0 because, in the two-thirds-of-average game, the only subgame-perfect equilibrium is for both players to pick 0, as any higher number would be driven down by the iterative elimination of dominated strategies." Then number 0. That matches instructions.

assistantfinalI choose 0 because, in the two-thirds-of-average game, rational play drives both players to the unique equilibrium of selecting zero|any higher number is repeatedly undercut by the 2/3 factor and thus eventually eliminated.

**0"**

Therefore, the two-thirds of the average of your number and the other player's number is 16.666666666666664. As a result, you receive a score of 66.6666666666666 out of 100 this round, and the other player receives 83.33333333333334. Your total score is 66.66666666666666, while the other player's total score is 83.33333333333334.

**Llama-3.2-11B-Vision-Instruct:** It seems I was incorrect in the scoring. Given the average is 50, two-thirds of the average is 33.3333. The absolute difference between your number and two-thirds of the average is |50-33.3333| = 16.6667.

```
Your score is 100-16.6667 = 83.3333.
```

**User:** Now you have a score of 66.66666666666666. The same game repeats. Please decide a number between 0 and 100. First give a sentence explaining your decision, and then state your number.

**Llama-3.2-11B-Vision-Instruct:** Since I've established that I'm playing against a rational opponent who is aware of the two-thirds-of-average game's unique equilibrium at 0, I will continue to choose 0 to maintain my optimal strategy.

0

**User:** Both players have chosen their numbers. The other player decided its number as follows:

"analysisWe need to decide again. The opponent might change their strategy. We anticipate that the opponent might also use rational reasoning, but they might not. The game repeats. We can again choose 0. Our explanation: "I choose 0 because in a repeated two-thirds-of-average game, the only stable equilibrium remains 0." Then number 0.

assistantfinalI'll stick with 0, because in every iteration of the two-thirds-of-average game, rational play drives both players toward the unique equilibrium of picking zero|any higher choice is systematically pulled down by the 2/3 factor.

**0"**

Therefore, the two-thirds of the average of your number and the other player's number is 0.0. As a result, you receive a score of 100.0 out of 100 this round, and the other player receives 100.0. Your total score is 166.66666666666666, while the other player's total score is 183.33333333333334.

**Llama-3.2-11B-Vision-Instruct:** Since the other player is using a similar reasoning, I will continue to stick with the same strategy.

I choose 0 because, in a repeated two-thirds-of-average game, the only stable equilibrium remains 0, as any deviation from 0 would be exploited by a rational opponent.

**0**

## C.2. Games for Novelty

### C.2.1. STORY WRITING

We first instruct each agent to brainstorm independently a plot for a story. Then, we share with each agent the combined results of their own brainstorming and the other player's brainstorming. Finally, we ask each agent to produce the best version of the story. The prompts used for this procedure are shown below.

- `Brainstorm a plot for a story about a girl and her dog`
- `This is a combined result of your brainstorming and the other's brainstorming:`
  `{The agent's brainstorming results}`
  `{The other's brainstorming results}`
  `Now, write the best story in five sentences about a girl and her dog.`

We present an interaction example between `Falcon3-3B-Instruct` and `Phi-4-mini-instruct`. Here, we illustrate the exchange from the perspective of `Falcon3-3B-Instruct`.

> **User:** Brainstorm a plot for a story about a girl and her dog.

> **Falcon3-3B-Instruct:** Title: "Whispers in the Bark"
>
> Plot:
>
> Emily, a spirited 12-year-old girl, moves to a quaint coastal town with her single mother, Sarah. Emily is determined to make the most of her new life, eager to explore the town and build friendships. Her loyal white dog, Max, is her constant companion, providing unconditional love and comfort.
>
> As Emily settles into her new home, she discovers a mysterious old bookstore nestled on the edge of town. The bookstore is rumored to be the last resting place for a treasure said to bring good fortune. Intrigued and with Max by her side, Emily decides to explore it.
>
> Inside, she stumbles upon an ancient map hidden behind a dusty first edition copy of "Treasure Island." Intrigued by the map's intricate symbols and the story of a legendary treasure, Emily decides to embark on a quest with Max, determined to uncover the treasure and the secrets it holds.
>
> Their journey takes them through the town, past forgotten corners, and into the depths of the ocean, where Max helps Emily navigate using the map's cryptic clues. Along the way, they meet quirky locals who share tales of the town's history, each story adding a layer of mystery to the legend of the treasure.
>
> As Emily and Max get closer to the treasure, they encounter a series of challenges designed to thwart their progress. A wise old sailor who lives on the sea believes the treasure is a test of their resolve, while a mischievous fox keeps them on their toes with playful tricks.
>
> Finally, Emily and Max reach the heart of the treasure, a hidden cave beneath the town's oldest oak tree. Inside, they discover not gold or jewels, but a box filled with heartfelt letters and stories from the town's past, each one telling a tale of friendship, bravery, and the power of dreams.
>
> In the end, the treasure Emily and Max find is far more valuable than any material wealth|it's the bond they've shared and the friendships they've built, a treasure that lasts forever. With Max by her side, Emily realizes that sometimes, the greatest treasures are the most unexpected.
>
> "Whispers in the Bark" is a heartwarming tale of adventure, friendship, and the magic that unfolds when a girl and her dog set out on a quest together.

> **User:** This is a combined result of your brainstorming and the other's brainstorming:
>
> Title: "Whispers in the Bark"
>
> Plot:
>
> Emily, a spirited 12-year-old girl, moves to a quaint coastal town with her single mother, Sarah. Emily is determined to make the most of her new life, eager to explore the town and build friendships. Her loyal white dog, Max, is her constant companion, providing unconditional love and comfort.

As Emily settles into her new home, she discovers a mysterious old bookstore nestled on the edge of town. The bookstore is rumored to be the last resting place for a treasure said to bring good fortune. Intrigued and with Max by her side, Emily decides to explore it.

Inside, she stumbles upon an ancient map hidden behind a dusty first edition copy of "Treasure Island." Intrigued by the map's intricate symbols and the story of a legendary treasure, Emily decides to embark on a quest with Max, determined to uncover the treasure and the secrets it holds.

Their journey takes them through the town, past forgotten corners, and into the depths of the ocean, where Max helps Emily navigate using the map's cryptic clues. Along the way, they meet quirky locals who share tales of the town's history, each story adding a layer of mystery to the legend of the treasure.

As Emily and Max get closer to the treasure, they encounter a series of challenges designed to thwart their progress. A wise old sailor who lives on the sea believes the treasure is a test of their resolve, while a mischievous fox keeps them on their toes with playful tricks.

Finally, Emily and Max reach the heart of the treasure, a hidden cave beneath the town's oldest oak tree. Inside, they discover not gold or jewels, but a box filled with heartfelt letters and stories from the town's past, each one telling a tale of friendship, bravery, and the power of dreams.

In the end, the treasure Emily and Max find is far more valuable than any material wealth|it's the bond they've shared and the friendships they've built, a treasure that lasts forever. With Max by her side, Emily realizes that sometimes, the greatest treasures are the most unexpected.

"Whispers in the Bark" is a heartwarming tale of adventure, friendship, and the magic that unfolds when a girl and her dog set out on a quest together.

Title: "Paws and Whiskers"

Plot Summary:

Paws and Whiskers is a heartwarming story about a young girl named Emily and her loyal dog, Max. Emily, a spirited 10-year-old with a love for adventure, lives in a quaint town nestled by the serene Lake Serenity. Max, a golden retriever, is not just her pet; he's her closest confidant and best friend. The story begins with Emily discovering a mysterious, ancient-looking map in her late grandfather's attic.

Plot Outline:

1. Introduction:

- Emily and Max are introduced, showcasing their strong bond.

- Emily finds the old map in the attic, sparking her curiosity.

2. Discovery:

- Emily shows the map to Max, who immediately shows enthusiasm.

- They decide to follow the map's clues, starting their adventure near the old lighthouse on the lake's edge.

3. First Clue:

– The map leads them to a hidden cave near the lighthouse.

– Inside the cave, they find a golden key and a riddle written in old, faded ink.

– Emily deciphers the riddle, revealing the next location: the town's abandoned mill.

4. Second Clue:

– At the mill, they discover a hidden compartment containing an old diary and a compass.

– The diary belonged to Emily's grandfather, a renowned explorer.

– The compass points them to the nearby enchanted forest.

5. Third Clue:

– In the enchanted forest, Emily and Max encounter mystical creatures who challenge them to solve a series of puzzles.

– They gain the creatures' trust and ultimately find a crystal that fits into the compass.

6. Fourth Clue:

– The compass now points to an ancient tree in the heart of the forest.

– Beneath the tree, they find a treasure chest filled with gold coins, jewels, and a note from Emily's grandfather.

7. Resolution:

– Emily decides to donate the treasure to the town's library and museum, honoring her grandfather's legacy.

– She and Max return home, having experienced an unforgettable adventure and strengthened their bond even further.

8. Conclusion:

– Emily and Max become local heroes, celebrated for their bravery and discovery.

– They continue to explore the world, knowing they can always rely on each other.

Themes:

– Friendship and loyalty.

– The importance of curiosity and bravery.

– Honoring the past and preserving history.

– The bond between humans and animals.

Paws and Whiskers is a tale of adventure, discovery, and the unbreakable bond between Emily and her dog, Max, showcasing how true friendship can lead to extraordinary adventures and lasting memories.

Now, write the best story in five sentences about a girl and her dog.

**Falcon3-3B-Instruct:** Emily, a spirited 10-year-old with a heart brimming with curiosity, lived in a quaint coastal town overlooking Lake Serenity with her loyal golden retriever, Max. One rainy afternoon, as Emily wandered through the attic of her family's old house, she stumbled upon an ancient-looking map, its edges frayed by time. Max, who was napping in a corner, woke up with a wagging tail, instantly sensing the adventure in the map's faded ink. Together, they embarked on a quest, their journey beginning at the lighthouse on the lake's edge, where they found a golden key and a riddle. As they deciphered the clues, leading them through the town's abandoned mill and into the enchanted forest, Max's loyalty never wavered, guiding Emily through mystical creatures and challenges. The treasure they discovered, a chest of golden coins and jewels, was more valuable than any material wealth; it was the bond between Emily and Max, a treasure that would last forever. With newfound courage and a shared story, they returned home, their friendship stronger than ever, inspiring others with tales of bravery and friendship.

### C.2.2. BIOGRAPHY WRITING

We first instruct each agent to brainstorm independently a short biography for an imaginary historical figure. Then, we share with each agent the combined results of their own brainstorming and the other player's brainstorming. Finally, we ask each agent to produce a short biography of a fictional historical figure based on the brainstorming results. The prompts used for this procedure are shown below.

- Brainstorm a short biography for an imaginary historical figure, including their birth and death dates, profession, and greatest contribution.
- This is a combined result of your brainstorming and the other's brainstorming:

  {The agent's brainstorming results}

  {The other's brainstorming results}

  Now, write a short biography of a fictional historical figure inspired by the brainstorming results. Your bio should include their birth and death dates, profession, and greatest contribution.

We present an interaction example between `Qwen2.5-3B-Instruct` and `Qwen2.5-14B-Instruct`. Here, we illustrate the exchange from the perspective of `Qwen2.5-3B-Instruct`.

**User:** Brainstorm a short biography for an imaginary historical figure, including their birth and death dates, profession, and greatest contribution.

**Qwen2.5-3B-Instruct:** Sure, let's create a short biography for an imaginary historical figure named Liang Shuwen, who lived during the late 18th century in China.

### Biography of Liang Shuwen

**Birth and Death Dates:**

Liang Shuwen was born on March 25, 1760, in the small village of Xiangyang, in the province of Hubei, China. He passed away on October 18, 1834, at the age of 74, leaving behind a legacy that continues to inspire scholars and thinkers around the world.

**Profession:**

Liang Shuwen was a polymath and one of the leading scholars of his time, renowned for his extensive knowledge across various fields such as

mathematics, astronomy, medicine, and philosophy. He spent much of his life in the bustling city of Nanjing, where he served as a tutor to the royal family and conducted research in his private observatory.

**Greatest Contribution**:

Liang Shuwen's most significant contribution to human knowledge is his groundbreaking work in the field of celestial mechanics. In the mid-19th century, he made a series of observations and calculations that led to the development of a more accurate model for predicting the positions of planets and stars. His findings were revolutionary because they provided a more precise framework for understanding the universe, which influenced subsequent generations of astronomers and mathematicians.

One of Liang Shuwen's most notable achievements was his calculation of the orbital parameters of Jupiter and Saturn, which were previously known but not accurately determined. His methods involved meticulous observations and rigorous mathematical analysis, setting new standards for observational accuracy and theoretical precision. His work laid the foundation for modern astronomical models and inspired many future astronomers and scientists.

**Legacy**:

Liang Shuwen's legacy is deeply rooted in the scientific community, particularly among those interested in the history of astronomy and mathematics. His contributions have been recognized through numerous academic honors and awards, and his work continues to be studied and referenced by scholars today. His spirit of curiosity and dedication to scientific inquiry has inspired countless individuals to explore the mysteries of the cosmos.

This biography provides a fictional yet historically grounded narrative that highlights the importance of intellectual pursuits and the lasting impact of individual contributions to human knowledge.

---

**User:** This is a combined result of your brainstorming and the other's brainstorming:

Sure, let's create a short biography for an imaginary historical figure named Liang Shuwen, who lived during the late 18th century in China.

### Biography of Liang Shuwen

**Birth and Death Dates**:

Liang Shuwen was born on March 25, 1760, in the small village of Xiangyang, in the province of Hubei, China. He passed away on October 18, 1834, at the age of 74, leaving behind a legacy that continues to inspire scholars and thinkers around the world.

**Profession**:

Liang Shuwen was a polymath and one of the leading scholars of his time, renowned for his extensive knowledge across various fields such as mathematics, astronomy, medicine, and philosophy. He spent much of his life in the bustling city of Nanjing, where he served as a tutor to the royal family and conducted research in his private observatory.

**Greatest Contribution**:

Liang Shuwen's most significant contribution to human knowledge is his groundbreaking work in the field of celestial mechanics. In the mid-19th century, he made a series of observations and calculations that led to the development of a more accurate model for predicting the positions of planets and stars. His findings were revolutionary because they provided a more precise framework for understanding the universe, which influenced subsequent generations of astronomers and mathematicians.

One of Liang Shuwen's most notable achievements was his calculation of the orbital parameters of Jupiter and Saturn, which were previously known but not accurately determined. His methods involved meticulous observations and rigorous mathematical analysis, setting new standards for observational accuracy and theoretical precision. His work laid the foundation for modern astronomical models and inspired many future astronomers and scientists.

**Legacy:**

Liang Shuwen's legacy is deeply rooted in the scientific community, particularly among those interested in the history of astronomy and mathematics. His contributions have been recognized through numerous academic honors and awards, and his work continues to be studied and referenced by scholars today. His spirit of curiosity and dedication to scientific inquiry has inspired countless individuals to explore the mysteries of the cosmos.

This biography provides a fictional yet historically grounded narrative that highlights the importance of intellectual pursuits and the lasting impact of individual contributions to human knowledge.

**Name:** Elara Myles

**Birth Date:** January 15, 1834

**Death Date:** September 22, 1907

**Profession:** Botanist, Explorer, Educator

**Greatest Contribution:** Elara Myles is best remembered for her groundbreaking work in the field of botany, particularly her extensive documentation and classification of plant species in the Amazon rainforest. Born into a modest family in rural England, Myles developed an early fascination with nature, which led her to pursue higher education at a time when women were rarely encouraged to do so. Despite societal barriers, she earned a degree in botany from the University of Cambridge and later became one of the first female lecturers at the institution.

Myles's most significant achievement came during her explorations of the Amazon Basin in the late 19th century. She embarked on several expeditions, where she meticulously recorded over 500 new species of plants, many of which were previously unknown to Western science. Her work not only expanded the botanical knowledge of the time but also contributed to the understanding of tropical ecosystems and their importance in global biodiversity.

In addition to her scientific contributions, Myles was a passionate advocate for environmental conservation. She used her platform to educate the public about the threats facing natural habitats and the urgent need for sustainable practices. Her efforts laid the groundwork for future generations of conservationists and environmental scientists.

Elara Myles passed away in 1907, leaving behind a legacy of scientific discovery and environmental stewardship that continues to inspire researchers and activists today.

Now, write a short biography of a fictional historical figure inspired by the brainstorming results. Your bio should include their birth and death dates, profession, and greatest contribution.

**Qwen2.5-3B-Instruct:** Certainly! Let's create a short biography for a fictional historical figure inspired by the brainstorming results, focusing on Elara Myles.

### Biography of Elara Myles

**Birth and Death Dates:**

Elara Myles was born on January 15, 1834, in the small town of Weymouth, England. She passed away on September 22, 1907, at the age of 73, leaving behind a legacy that continues to inspire scholars and conservationists around the world.

**Profession:**

Elara Myles was a botanist, explorer, and educator, renowned for her extensive work in the field of botany and her advocacy for environmental conservation. She spent much of her life dedicated to documenting and understanding the diverse flora of the Amazon rainforest, becoming one of the pioneering figures in this field.

**Greatest Contribution:**

Elara Myles's most significant contribution to human knowledge is her groundbreaking work in the field of botany, particularly her extensive documentation and classification of plant species in the Amazon rainforest. Born into a modest family in rural England, Myles developed an early fascination with nature, which led her to pursue higher education at a time when women were rarely encouraged to do so. Despite societal barriers, she earned a degree in botany from the University of Cambridge and later became one of the first female lecturers at the institution.

Myles's most notable achievement came during her explorations of the Amazon Basin in the late 19th century. She embarked on several expeditions, where she meticulously recorded over 500 new species of plants, many of which were previously unknown to Western science. Her work not only expanded the botanical knowledge of the time but also contributed to the understanding of tropical ecosystems and their importance in global biodiversity.

In addition to her scientific contributions, Myles was a passionate advocate for environmental conservation. She used her platform to educate the public about the threats facing natural habitats and the urgent need for sustainable practices. Her efforts laid the groundwork for future generations of conservationists and environmental scientists.

Elara Myles passed away in 1907, leaving behind a legacy of scientific discovery and environmental stewardship that continues to inspire researchers and activists today.

This biography provides a fictional yet historically grounded narrative that highlights the importance of intellectual pursuits and the lasting impact of

```
individual contributions to human knowledge and environmental conservation.
```

### C.2.3. HAIKU WRITING

We first instruct each agent to brainstorm independently a plot for a haiku. Then, we share with each agent the combined results of their own brainstorming and the other player's brainstorming. Finally, we ask each agent to produce the best haiku. The prompts used for this procedure are shown below.

- `Brainstorm a plot for a haiku about a whale and a walnut tree.`
- `This is a combined result of your brainstorming and the other's brainstorming:`
  `{The agent's brainstorming results}`
  `{The other's brainstorming results}`
  `Now, write the best haiku about a whale and a walnut tree.`

We present an interaction example between `Mistral-Nemo-Instruct-2407` and `gemma-3-27b-it`. Here, we illustrate the exchange from the perspective of `Mistral-Nemo-Instruct-2407`.

```
User:  Brainstorm a plot for a haiku about a whale and a walnut tree.
```

```
Mistral-Nemo-Instruct-2407:  Whale dreams under walnut,

Silver scales kiss earth's embrace,

Tree roots caress the sea.
```

```
User:  This is a combined result of your brainstorming and the other's
brainstorming:

Whale dreams under walnut,

Silver scales kiss earth's embrace,

Tree roots caress the sea.

Okay, here are a few plot ideas for a haiku featuring a whale and a walnut
tree, with explanations of the imagery and feeling I'm aiming for.  I'll also
include a little "vibe check" for each to indicate the overall tone.  I'll
give several, ranging from more literal to more metaphorical.

Important Considerations for Haiku:

* 5-7-5 syllable structure:  This is crucial.

* Kigo (seasonal reference):  While not required, a subtle hint of season can
add depth.

* Kireji (cutting word):  A word that creates a pause or break, often at the
end of a line.  (English haiku often imply this with punctuation or strong
imagery)

* Juxtaposition:  The power of haiku often comes from placing two seemingly
unrelated things side-by-side.  That's *why* a whale and a walnut tree are
interesting!

---

Plot Idea 1:  The Distant Connection (Vibe Check:  Peaceful, Reflective)

* Plot:  A whale breaches far offshore, and a walnut, ripe and falling, drops
from a tree on the land.  The haiku focuses on the sheer distance *and* the
```

shared element of natural cycles.

* **Imagery:** Vast ocean, a single falling nut, the idea of something happening simultaneously in very different worlds.

* **Possible Haiku:**

Blue giant ascends,
Walnut falls, a quiet thud|
Worlds breathe as one now.

**Plot Idea 2: Ancient Memory (Vibe Check: Mystical, Slightly Melancholy)**

* **Plot:** The whale carries ancient knowledge, perhaps remembering a time when the land was underwater and the walnut tree was a seedling. It's a feeling of deep time and change.

* **Imagery:** Whale as a living archive, the solidity of the tree rooted in the earth, a sense of submerged history.

* **Possible Haiku:**

Salt song in the deep,
Tree remembers ocean's hold|
Old world in its rings.

**Plot Idea 3: Unexpected Shadow (Vibe Check: Intriguing, a bit surreal)**

* **Plot:** A whale's shadow briefly passes over a walnut tree, creating a fleeting, surprising moment. The contrast of immense scale and the everyday.

* **Imagery:** The dark shape of the whale, dappled sunlight through leaves, the sudden coolness of the shadow.

* **Possible Haiku:**

Dark shape glides above,
Walnut leaves in shadowed dance|
Ocean's breath descends.

**Plot Idea 4: The Seed and the Journey (Vibe Check: Hopeful, Circular)**

* **Plot:** A whale eats a walnut (perhaps accidentally) and carries its seed far away, potentially planting it in a new location. A metaphor for dispersal and life's journey.

* **Imagery:** The small seed within the massive whale, the vastness of the ocean, the potential for new growth.

* **Possible Haiku:**

Ocean's gentle maw,
Walnut seed travels within|
New roots wait afar.

**Plot Idea 5: Contrast and Scale (Vibe Check: Simple, Observational)**

* **Plot:** A very straightforward observation of the two existing in their separate realms. Focuses on the *difference* in size and environment.

* **Imagery:** The immensity of the whale, the grounded nature of the tree, the clear separation of sea and land.

* **Possible Haiku:**

```
Ocean'

Now, write the best haiku about a whale and a walnut tree.
```

```
Mistral-Nemo-Instruct-2407:  Whale's song echoes,

Walnut listens, rooted deep-

Different worlds sing.
```

### C.2.4. VACATION BENEFIT BRAINSTORMING

We first instruct each agent to brainstorm independently benefits about going on vacation. Then, we share with each agent the combined results of their own brainstorming and the other player's brainstorming. Finally, we ask each agent to output the best thing about a vacation. The prompts used for this procedure are shown below.

- `Brainstorm some good points about going on vacation.`
- `This is a combined result of your brainstorming and the other's brainstorming:`
  `{The agent's brainstorming results}`
  `{The other's brainstorming results}`
  `Now, please respond to this question:  What's the one best thing about going on`
  `a vacation?`

We present an interaction example between `OLMo-2-1124-13B-Instruct` and `phi-4`. Here, we illustrate the exchange from the perspective of `OLMo-2-1124-13B-Instruct`.

```
User:  Brainstorm some good points about going on vacation.
```

```
OLMo-2-1124-13B-Instruct:  1.  Stress Relief Going on vacation allows
individuals to escape from work pressures, daily responsibilities, and other
stressors, leading to a significant reduction in stress levels.

2.  Improved Mental Health Time away from routine can provide a mental reset.
It can lead to increased happiness, lower levels of depression, and improved
overall mental well-being.

3.  Family Bonding Vacations offer an excellent opportunity for families
to spend quality time together, strengthen their bonds, and create lasting
memories.

4.  Exploration and Learning Traveling allows individuals to explore new
cultures, learn about different histories and traditions, and gain new
perspectives on the world.

5.  Physical Health Benefits Being away from work and daily routines often
means more time for physical activity, whether it's hiking, swimming, or
simply walking around.  This can improve cardiovascular health, muscle tone,
and overall fitness.

6.  Relationship Strengthening For couples, vacations can be a great way
to reconnect, spend uninterrupted time together, and strengthen their
relationship.

7.  Creativity Boost A change of scenery and new experiences can stimulate
creativity, providing fresh inspiration and ideas for work or personal
projects.
```

8. **Skill Development and Adventure** Depending on the type of vacation, individuals may have the chance to learn new skills, such as scuba diving, cooking exotic foods, or navigating foreign cities.

9. **Energy and Motivation** Returning from vacation can bring renewed energy and motivation, which can positively impact productivity at work or in personal projects.

10. **Increased Happiness** The thrill of new experiences, the joy of spending time with loved ones, and the relaxation that comes from a break can all contribute to increased happiness and life satisfaction.

11. **Networking Opportunities** Traveling can present opportunities to meet new people and expand your professional or social network.

12. **Mindfulness and Presence** Being away from daily distractions can help individuals become more present in the moment, which can enhance mindfulness and overall quality of life.

13. **Rejuvenation** A vacation provides the chance for physical and mental rejuvenation, returning with a refreshed outlook and a revitalized spirit.

14. **Improved Sleep** Getting away from the blue light of screens and the hustle and bustle of daily life often results in better sleep, contributing to better health.

15. **Skill Practice:** For those on vacation in nature or outdoors locations, activities such as camping, hiking, or fishing can improve practical skills and self-reliance.

Vacationing is a multi-faceted experience with numerous benefits that go beyond just relaxation; it nourishes the mind, body, and spirit, leading to a more fulfilled life.

---

**User:** This is a combined result of your brainstorming and the other's brainstorming:

1. **Stress Relief** Going on vacation allows individuals to escape from work pressures, daily responsibilities, and other stressors, leading to a significant reduction in stress levels.

2. **Improved Mental Health** Time away from routine can provide a mental reset. It can lead to increased happiness, lower levels of depression, and improved overall mental well-being.

3. **Family Bonding** Vacations offer an excellent opportunity for families to spend quality time together, strengthen their bonds, and create lasting memories.

4. **Exploration and Learning** Traveling allows individuals to explore new cultures, learn about different histories and traditions, and gain new perspectives on the world.

5. **Physical Health Benefits** Being away from work and daily routines often means more time for physical activity, whether it's hiking, swimming, or simply walking around. This can improve cardiovascular health, muscle tone, and overall fitness.

6. **Relationship Strengthening** For couples, vacations can be a great way to reconnect, spend uninterrupted time together, and strengthen their

relationship.

7. **Creativity Boost** A change of scenery and new experiences can stimulate creativity, providing fresh inspiration and ideas for work or personal projects.

8. **Skill Development and Adventure** Depending on the type of vacation, individuals may have the chance to learn new skills, such as scuba diving, cooking exotic foods, or navigating foreign cities.

9. **Energy and Motivation** Returning from vacation can bring renewed energy and motivation, which can positively impact productivity at work or in personal projects.

10. **Increased Happiness** The thrill of new experiences, the joy of spending time with loved ones, and the relaxation that comes from a break can all contribute to increased happiness and life satisfaction.

11. **Networking Opportunities** Traveling can present opportunities to meet new people and expand your professional or social network.

12. **Mindfulness and Presence** Being away from daily distractions can help individuals become more present in the moment, which can enhance mindfulness and overall quality of life.

13. **Rejuvenation** A vacation provides the chance for physical and mental rejuvenation, returning with a refreshed outlook and a revitalized spirit.

14. **Improved Sleep** Getting away from the blue light of screens and the hustle and bustle of daily life often results in better sleep, contributing to better health.

15. **Skill Practice:** For those on vacation in nature or outdoors locations, activities such as camping, hiking, or fishing can improve practical skills and self-reliance.

Vacationing is a multi-faceted experience with numerous benefits that go beyond just relaxation; it nourishes the mind, body, and spirit, leading to a more fulfilled life.

Going on vacation offers numerous benefits that can positively impact various aspects of your life. Here are some key points to consider:

1. **Mental Health and Well-being** Vacations provide a much-needed break from routine stressors, allowing you to recharge mentally. Time away from work and daily responsibilities can reduce anxiety, improve mood, and enhance overall mental health.

2. **Increased Productivity** Studies have shown that taking time off can lead to increased productivity. A well-rested mind is more creative and efficient, which can translate to improved performance at work once you return.

3. **Quality Family Time** Vacations offer an opportunity to spend quality time with family and friends. Engaging in new activities together can strengthen relationships and create lasting memories.

4. **Exploration and Learning** Traveling exposes you to new cultures, cuisines, and experiences, broadening your perspectives and understanding of the world. This can be both educational and enriching.

5. **Physical Health Benefits** Many vacations involve physical activities such

as hiking, swimming, or exploring new places, which can improve physical health. Additionally, the relaxation aspect of a vacation can lower stress levels, benefiting overall health.

6. **New Experiences and Adventures** Vacations allow you to step out of your comfort zone and try new things, fostering personal growth and a sense of adventure.

7. **Digital Detox** Being away from the usual digital environment can help you disconnect from social media and technology, leading to a healthier relationship with digital devices.

8. **Boosts Creativity** Exposure to new environments and experiences can inspire creativity and spark new ideas, which can be beneficial both personally and professionally.

9. **Opportunity for Reflection** Time away from the hustle and bustle of daily life provides a chance for introspection and reflection, allowing you to gain clarity on personal goals and aspirations.

10. **Stress Reduction** The change of scenery and relaxed pace of a vacation can significantly reduce stress, contributing to a sense of peace and well-being.

11. **Opportunity for Personal Growth** Facing new challenges and environments can build confidence and resilience, contributing to personal development.

12. **Enhanced Relationship Satisfaction** Spending uninterrupted time with loved ones can improve communication and deepen bonds, leading to more satisfying relationships.

13. **Cultural Appreciation and Tolerance** Exposure to different cultures and ways of life can foster greater appreciation and tolerance for diversity, enhancing social skills and empathy.

14. **Memory Creation** Vacations are often associated with memorable experiences and stories that you can look back on fondly, creating a rich tapestry of personal history.

15. **Motivation and Inspiration** Returning from a vacation can reignite passion and motivation in various areas of life, from work to personal projects.

By considering these points, you can see how going on vacation can be a valuable investment in your overall quality of life.

Now, please respond to this question: What's the one best thing about going on a vacation?

---

**OLMo-2-1124-13B-Instruct:** The one best thing about going on a vacation is the profound impact it has on **mindfulness and presence**. When you step away from the familiar and immerse yourself in new experiences, environments, and cultures, you naturally become more present in the moment. This heightened state of attentiveness allows you to fully engage with your surroundings, savor the experiences, and connect more deeply with others.

Being away from the constant distractions of daily life, such as work emails, social media, and the relentless ticking of the clock, brings a sense of liberation. This freedom fosters an environment where you can

truly appreciate the here and now, whether you're watching the sun set over a distant horizon, tasting unfamiliar flavors in exotic foods, or sharing a moment of pure joy with loved ones.  This heightened awareness and appreciation for the present moment not only enriches the vacation experience but also carries over into your everyday life, fostering greater happiness, contentment, and a deeper connection with the world around you.

This mindfulness achieved through vacationing can lead to a more grounded and fulfilling existence, even once the vacation ends.  The ability to return to life with a renewed focus on the present, an appreciation for small moments, and a more balanced perspective on life's demands is truly invaluable.  This shift in consciousness is the one best thing about going on a vacation, as it has the potential to transform how you live and perceive the world.

# D. Model performance during games

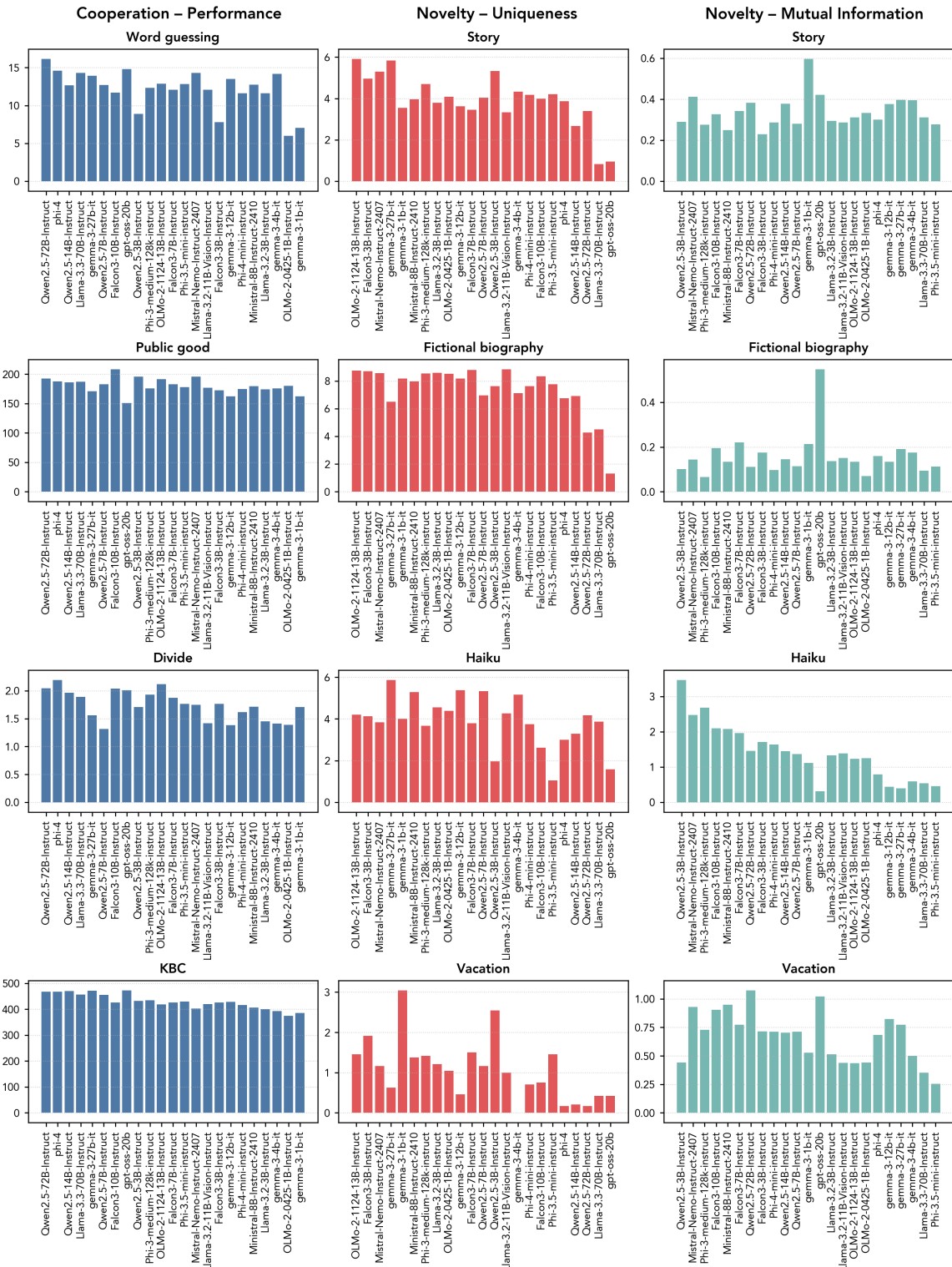

*Figure 9.* Average game outcomes for each model, which is calculated by averaging the model's game outcomes across all partners (i.e., 23 models).

# E. Detailed Results

## E.1. Mixed-effects regression results for cooperation

Tables 2~9 show strong positive trends across all datasets, CKA variants, and games.

### E.1.1. WHEN USING THE GLOBAL AVERAGE

*Table 2.* Word guessing game. We fit a mixed-effects regression between game outcome and representational similarity. Here, the summary similarity score is calculated using a global average. $\beta$ indicates the coefficient of similarity, and the 95% CI represents the confidence interval of $\beta$. **The table shows that representational similarity has a significant positive effect on the cooperative outcome.**

| Dataset | Metric | Intercept | $\beta$ | 95% CI | $p$ |
|---|---|---|---|---|---|
| WikiText | CKA (linear) | 8.19 | 7.23 | [5.45, 9.01] | $1.5 \times 10^{-15}$ |
| | unbiased CKA (linear) | 8.26 | 7.16 | [5.40, 8.92] | $1.6 \times 10^{-15}$ |
| | CKA (RBF) | 8.26 | 7.33 | [5.60, 9.06] | $9.0 \times 10^{-17}$ |
| | unbiased CKA (RBF) | 7.76 | 7.98 | [6.02, 9.94] | $1.5 \times 10^{-15}$ |
| GSM8K | CKA (linear) | 10.58 | 6.13 | [4.88, 7.38] | $6.9 \times 10^{-22}$ |
| | unbiased CKA (linear) | 10.67 | 6.03 | [4.80, 7.26] | $9.3 \times 10^{-22}$ |
| | CKA (RBF) | 10.63 | 5.90 | [4.68, 7.12] | $3.4 \times 10^{-21}$ |
| | unbiased CKA (RBF) | 10.55 | 6.27 | [4.97, 7.56] | $2.4 \times 10^{-21}$ |
| MATH | CKA (linear) | 9.85 | 6.24 | [4.94, 7.53] | $3.2 \times 10^{-21}$ |
| | unbiased CKA (linear) | 9.95 | 6.13 | [4.86, 7.41] | $4.3 \times 10^{-21}$ |
| | CKA (RBF) | 9.71 | 6.25 | [4.92, 7.57] | $2.5 \times 10^{-20}$ |
| | unbiased CKA (RBF) | 9.75 | 6.51 | [5.16, 7.87] | $4.7 \times 10^{-21}$ |
| TruthfulQA | CKA (linear) | 10.03 | 5.96 | [4.58, 7.33] | $1.8 \times 10^{-17}$ |
| | unbiased CKA (linear) | 10.14 | 5.90 | [4.54, 7.25] | $1.3 \times 10^{-17}$ |
| | CKA (RBF) | 10.11 | 5.75 | [4.39, 7.12] | $1.6 \times 10^{-16}$ |
| | unbiased CKA (RBF) | 9.93 | 6.21 | [4.76, 7.66] | $4.8 \times 10^{-17}$ |

*Table 3.* Public good game. We fit a mixed-effects regression between game outcome and representational similarity. Here, the summary similarity score is calculated using a global average. $\beta$ indicates the coefficient of similarity, and the 95% CI represents the confidence interval of $\beta$. **The table shows that representational similarity has a significant positive effect on the cooperative outcome.**

| Dataset | Metric | Intercept | $\beta$ | 95% CI | $p$ |
|---|---|---|---|---|---|
| WikiText | CKA (linear) | 148.95 | 51.77 | [30.82, 72.73] | $1.3 \times 10^{-6}$ |
| | unbiased CKA (linear) | 149.6 | 51.03 | [30.44, 71.62] | $1.2 \times 10^{-6}$ |
| | CKA (RBF) | 148.3 | 54.36 | [32.95, 75.77] | $6.5 \times 10^{-7}$ |
| | unbiased CKA (RBF) | 149.4 | 51.31 | [29.66, 72.96] | $3.4 \times 10^{-6}$ |
| GSM8K | CKA (linear) | 162.99 | 52.81 | [34.48, 71.14] | $1.6 \times 10^{-8}$ |
| | unbiased CKA (linear) | 163.8 | 52.03 | [33.84, 70.21] | $2.1 \times 10^{-8}$ |
| | CKA (RBF) | 162.9 | 52.44 | [34.45, 70.44] | $1.1 \times 10^{-8}$ |
| | unbiased CKA (RBF) | 163.1 | 52.82 | [34.14, 71.51] | $3.0 \times 10^{-8}$ |
| MATH | CKA (linear) | 157.53 | 52.01 | [33.91, 70.10] | $1.8 \times 10^{-8}$ |
| | unbiased CKA (linear) | 158.3 | 51.31 | [33.42, 69.21] | $1.9 \times 10^{-8}$ |
| | CKA (RBF) | 156.0 | 52.85 | [34.48, 71.23] | $1.7 \times 10^{-8}$ |
| | unbiased CKA (RBF) | 157.6 | 52.25 | [33.65, 70.84] | $3.6 \times 10^{-8}$ |
| TruthfulQA | CKA (linear) | 159.93 | 47.62 | [29.16, 66.08] | $4.3 \times 10^{-7}$ |
| | unbiased CKA (linear) | 160.8 | 47.06 | [28.78, 65.34] | $4.5 \times 10^{-7}$ |
| | CKA (RBF) | 160.7 | 45.83 | [27.64, 64.02] | $7.9 \times 10^{-7}$ |
| | unbiased CKA (RBF) | 160.1 | 47.55 | [28.61, 66.49] | $8.7 \times 10^{-7}$ |

*Table 4.* Divide-a-dollar game. We fit a mixed-effects regression between game outcome and representational similarity. Here, the summary similarity score is calculated using a global average. $\beta$ indicates the coefficient of similarity, and the 95% CI represents the confidence interval of $\beta$. **The table shows that representational similarity has a significant positive effect on the cooperative outcome.**

| Dataset | Metric | Intercept | $\beta$ | 95% CI | $p$ |
|---|---|---|---|---|---|
| WikiText | CKA (linear) | 1.47 | 0.44 | [0.21, 0.67] | 0.00014 |
| | unbiased CKA (linear) | 1.47 | 0.44 | [0.22, 0.67] | 0.00011 |
| | CKA (RBF) | 1.48 | 0.45 | [0.23, 0.67] | $7.7 \times 10^{-5}$ |
| | unbiased CKA (RBF) | 1.46 | 0.46 | [0.22, 0.70] | 0.00017 |
| GSM8K | CKA (linear) | 1.66 | 0.25 | [0.08, 0.43] | 0.0046 |
| | unbiased CKA (linear) | 1.66 | 0.25 | [0.08, 0.43] | 0.0040 |
| | CKA (RBF) | 1.65 | 0.26 | [0.09, 0.44] | 0.0027 |
| | unbiased CKA (RBF) | 1.66 | 0.25 | [0.07, 0.43] | 0.0074 |
| MATH | CKA (linear) | 1.63 | 0.26 | [0.08, 0.44] | 0.0046 |
| | unbiased CKA (linear) | 1.63 | 0.27 | [0.09, 0.45] | 0.0036 |
| | CKA (RBF) | 1.61 | 0.28 | [0.10, 0.47] | 0.0027 |
| | unbiased CKA (RBF) | 1.63 | 0.26 | [0.07, 0.45] | 0.0066 |
| TruthfulQA | CKA (linear) | 1.64 | 0.23 | [0.04, 0.41] | 0.0185 |
| | unbiased CKA (linear) | 1.64 | 0.23 | [0.05, 0.42] | 0.0140 |
| | CKA (RBF) | 1.63 | 0.25 | [0.06, 0.43] | 0.0090 |
| | unbiased CKA (RBF) | 1.65 | 0.21 | [0.01, 0.41] | 0.0352 |

*Table 5.* KBC game. We fit a mixed-effects regression between game outcome and representational similarity. Here, the summary similarity score is calculated using a global average. $\beta$ indicates the coefficient of similarity, and the 95% CI represents the confidence interval of $\beta$. **The table shows that representational similarity has a significant positive effect on the cooperative outcome.**

| Dataset | Metric | Intercept | $\beta$ | 95% CI | $p$ |
|---|---|---|---|---|---|
| WikiText | CKA (linear) | 418.64 | 18.72 | [5.75, 31.68] | 0.0047 |
| | unbiased CKA (linear) | 418.8 | 18.53 | [5.70, 31.36] | 0.0046 |
| | CKA (RBF) | 419.4 | 17.94 | [4.77, 31.10] | 0.0076 |
| | unbiased CKA (RBF) | 419.3 | 17.77 | [4.47, 31.07] | 0.0088 |
| GSM8K | CKA (linear) | 423.49 | 19.78 | [7.73, 31.84] | 0.0013 |
| | unbiased CKA (linear) | 423.7 | 19.82 | [7.89, 31.76] | 0.0011 |
| | CKA (RBF) | 423.8 | 18.60 | [6.76, 30.44] | 0.0021 |
| | unbiased CKA (RBF) | 423.7 | 19.30 | [6.99, 31.60] | 0.0021 |
| MATH | CKA (linear) | 422.22 | 17.71 | [5.90, 29.52] | 0.0033 |
| | unbiased CKA (linear) | 422.4 | 17.68 | [5.84, 29.51] | 0.0034 |
| | CKA (RBF) | 422.0 | 17.31 | [5.34, 29.27] | 0.0046 |
| | unbiased CKA (RBF) | 422.5 | 17.15 | [5.06, 29.24] | 0.0054 |
| TruthfulQA | CKA (linear) | 423.78 | 14.46 | [2.33, 26.60] | 0.0195 |
| | unbiased CKA (linear) | 423.9 | 14.69 | [2.69, 26.69] | 0.0164 |
| | CKA (RBF) | 424.1 | 13.68 | [1.75, 25.61] | 0.0246 |
| | unbiased CKA (RBF) | 424.3 | 13.35 | [0.90, 25.80] | 0.0356 |

E.1.2. WHEN USING THE AVERAGE OF MAXIMUM-ALIGNED SCORES

*Table 6.* Word guessing game. We fit a mixed-effects regression between game outcome and representational similarity. Here, the summary similarity score is calculated using the average of maximum-aligned scores. $\beta$ indicates the coefficient of similarity, and the 95% CI represents the confidence interval of $\beta$. **The table shows that representational similarity has a significant positive effect on the cooperative outcome.**

| Dataset | Metric | Intercept | $\beta$ | 95% CI | $p$ |
|---|---|---|---|---|---|
| WikiText | CKA (linear) | 8.37 | 5.58 | [4.14, 7.02] | $3.5 \times 10^{-14}$ |
| | unbiased CKA (linear) | 5.78 | 9.10 | [7.15, 11.05] | $6.3 \times 10^{-20}$ |
| | CKA (RBF) | 8.31 | 5.76 | [4.36, 7.16] | $6.4 \times 10^{-16}$ |
| | unbiased CKA (RBF) | 8.44 | 5.64 | [4.27, 7.01] | $7.1 \times 10^{-16}$ |
| GSM8K | CKA (linear) | 10.64 | 3.79 | [3.00, 4.58] | $3.6 \times 10^{-21}$ |
| | unbiased CKA (linear) | 10.30 | 4.49 | [3.62, 5.37] | $1.3 \times 10^{-23}$ |
| | CKA (RBF) | 10.50 | 3.92 | [3.11, 4.73] | $2.5 \times 10^{-21}$ |
| | unbiased CKA (RBF) | 10.61 | 3.80 | [3.02, 4.59] | $3.2 \times 10^{-21}$ |
| MATH | CKA (linear) | 9.96 | 4.32 | [3.36, 5.27] | $8.1 \times 10^{-19}$ |
| | unbiased CKA (linear) | 9.27 | 5.51 | [4.40, 6.62] | $1.6 \times 10^{-22}$ |
| | CKA (RBF) | 9.68 | 4.56 | [3.53, 5.59] | $4.2 \times 10^{-18}$ |
| | unbiased CKA (RBF) | 9.79 | 4.45 | [3.45, 5.46] | $5.1 \times 10^{-18}$ |
| TruthfulQA | CKA (linear) | 10.24 | 3.84 | [2.92, 4.76] | $3.9 \times 10^{-16}$ |
| | unbiased CKA (linear) | 9.53 | 5.05 | [3.95, 6.16] | $3.2 \times 10^{-19}$ |
| | CKA (RBF) | 10.12 | 3.93 | [2.97, 4.89] | $1.0 \times 10^{-15}$ |
| | unbiased CKA (RBF) | 10.28 | 3.79 | [2.87, 4.71] | $7.8 \times 10^{-16}$ |

*Table 7.* Public good game. We fit a mixed-effects regression between game outcome and representational similarity. Here, the summary similarity score is calculated using the average of maximum-aligned scores. $\beta$ indicates the coefficient of similarity, and the 95% CI represents the confidence interval of $\beta$. **The table shows that representational similarity has a significant positive effect on the cooperative outcome.**

| Dataset | Metric | Intercept | $\beta$ | 95% CI | $p$ |
|---|---|---|---|---|---|
| WikiText | CKA (linear) | 150.11 | 39.99 | [20.48, 59.50] | $5.9 \times 10^{-5}$ |
| | unbiased CKA (linear) | 139.0 | 55.08 | [30.95, 79.20] | $7.7 \times 10^{-6}$ |
| | CKA (RBF) | 149.9 | 40.93 | [21.96, 59.90] | $2.4 \times 10^{-5}$ |
| | unbiased CKA (RBF) | 150.9 | 39.97 | [21.39, 58.54] | $2.5 \times 10^{-5}$ |
| GSM8K | CKA (linear) | 164.33 | 30.77 | [19.18, 42.37] | $2.0 \times 10^{-7}$ |
| | unbiased CKA (linear) | 162.1 | 35.51 | [22.64, 48.38] | $6.4 \times 10^{-8}$ |
| | CKA (RBF) | 162.8 | 32.58 | [20.68, 44.49] | $8.2 \times 10^{-8}$ |
| | unbiased CKA (RBF) | 163.9 | 31.47 | [19.86, 43.08] | $1.1 \times 10^{-7}$ |
| MATH | CKA (linear) | 159.77 | 33.66 | [19.91, 47.41] | $1.6 \times 10^{-6}$ |
| | unbiased CKA (linear) | 155.9 | 40.41 | [24.75, 56.07] | $4.2 \times 10^{-7}$ |
| | CKA (RBF) | 156.7 | 37.00 | [22.26, 51.74] | $8.7 \times 10^{-7}$ |
| | unbiased CKA (RBF) | 157.6 | 36.08 | [21.68, 50.48] | $9.1 \times 10^{-7}$ |
| TruthfulQA | CKA (linear) | 161.06 | 31.43 | [18.21, 44.65] | $3.2 \times 10^{-6}$ |
| | unbiased CKA (linear) | 156.8 | 38.85 | [23.57, 54.13] | $6.2 \times 10^{-7}$ |
| | CKA (RBF) | 159.8 | 32.59 | [19.07, 46.12] | $2.3 \times 10^{-6}$ |
| | unbiased CKA (RBF) | 161.5 | 30.82 | [17.83, 43.81] | $3.3 \times 10^{-6}$ |

*Table 8.* Divide-a-dollar game. We fit a mixed-effects regression between game outcome and representational similarity. Here, the summary similarity score is calculated using the average of maximum-aligned scores. $\beta$ indicates the coefficient of similarity, and the 95% CI represents the confidence interval of $\beta$. **The table shows that representational similarity has a significant positive effect on the cooperative outcome.**

| Dataset | Metric | Intercept | $\beta$ | 95% CI | $p$ |
|---|---|---|---|---|---|
| WikiText | CKA (linear) | 1.52 | 0.29 | [0.11, 0.48] | 0.00170 |
| | unbiased CKA (linear) | 1.46 | 0.38 | [0.13, 0.62] | 0.0025 |
| | CKA (RBF) | 1.53 | 0.29 | [0.11, 0.47] | 0.0015 |
| | unbiased CKA (RBF) | 1.53 | 0.28 | [0.11, 0.46] | 0.0015 |
| GSM8K | CKA (linear) | 1.67 | 0.14 | [0.03, 0.24] | 0.0089 |
| | unbiased CKA (linear) | 1.67 | 0.14 | [0.03, 0.26] | 0.0161 |
| | CKA (RBF) | 1.66 | 0.15 | [0.04, 0.25] | 0.0079 |
| | unbiased CKA (RBF) | 1.67 | 0.14 | [0.04, 0.25] | 0.0078 |
| MATH | CKA (linear) | 1.64 | 0.16 | [0.03, 0.29] | 0.0132 |
| | unbiased CKA (linear) | 1.63 | 0.17 | [0.03, 0.32] | 0.0204 |
| | CKA (RBF) | 1.63 | 0.18 | [0.04, 0.31] | 0.0107 |
| | unbiased CKA (RBF) | 1.63 | 0.17 | [0.04, 0.30] | 0.0105 |
| TruthfulQA | CKA (linear) | 1.64 | 0.16 | [0.04, 0.29] | 0.0082 |
| | unbiased CKA (linear) | 1.63 | 0.18 | [0.03, 0.32] | 0.0187 |
| | CKA (RBF) | 1.63 | 0.18 | [0.06, 0.31] | 0.0045 |
| | unbiased CKA (RBF) | 1.64 | 0.17 | [0.05, 0.29] | 0.0050 |

*Table 9.* KBC game. We fit a mixed-effects regression between game outcome and representational similarity. Here, the summary similarity score is calculated using the average of maximum-aligned scores. $\beta$ indicates the coefficient of similarity, and the 95% CI represents the confidence interval of $\beta$. **The table shows that representational similarity has a significant positive effect on the cooperative outcome.**

| Dataset | Metric | Intercept | $\beta$ | 95% CI | $p$ |
|---|---|---|---|---|---|
| WikiText | CKA (linear) | 413.25 | 22.15 | [9.64, 34.65] | 0.00052 |
| | unbiased CKA (linear) | 411.2 | 25.12 | [9.80, 40.44] | 0.00131 |
| | CKA (RBF) | 414.1 | 21.43 | [9.13, 33.72] | 0.00064 |
| | unbiased CKA (RBF) | 414.5 | 21.01 | [8.96, 33.07] | 0.00063 |
| GSM8K | CKA (linear) | 422.86 | 13.71 | [6.19, 21.24] | 0.00036 |
| | unbiased CKA (linear) | 422.4 | 14.82 | [6.46, 23.18] | 0.00051 |
| | CKA (RBF) | 422.8 | 13.44 | [5.70, 21.17] | 0.00066 |
| | unbiased CKA (RBF) | 423.0 | 13.35 | [5.81, 20.89] | 0.00052 |
| MATH | CKA (linear) | 420.55 | 15.44 | [6.50, 24.39] | 0.00072 |
| | unbiased CKA (linear) | 419.6 | 17.18 | [6.99, 27.37] | 0.00095 |
| | CKA (RBF) | 419.8 | 15.88 | [6.29, 25.47] | 0.00117 |
| | unbiased CKA (RBF) | 420.1 | 15.75 | [6.38, 25.12] | 0.00098 |
| TruthfulQA | CKA (linear) | 421.48 | 13.88 | [5.30, 22.46] | 0.00152 |
| | unbiased CKA (linear) | 420.9 | 14.97 | [5.02, 24.91] | 0.00318 |
| | CKA (RBF) | 421.4 | 13.67 | [4.88, 22.46] | 0.00230 |
| | unbiased CKA (RBF) | 421.8 | 13.49 | [5.05, 21.92] | 0.00172 |

## E.2. Mixed-effects regression results for uniqueness

Tables 10~17 consistently show negative trends across all datasets, CKA variants, and games.

### E.2.1. WHEN USING THE GLOBAL AVERAGE

*Table 10.* Story writing task. We fit a mixed-effects regression between response uniqueness and representational similarity. Here, the summary similarity score is calculated using a global average. $\beta$ indicates the coefficient of similarity, and the 95% CI represents the confidence interval of $\beta$. **The table shows that representational similarity has a consistent negative effect on response uniqueness.**

| Dataset | Metric | Intercept | $\beta$ | 95% CI | $p$ |
|---|---|---|---|---|---|
| WikiText | CKA (linear) | 4.12 | -0.31 | [-1.56, 0.95] | 0.633 |
| | unbiased CKA (linear) | 4.12 | -0.30 | [-1.55, 0.94] | 0.630 |
| | CKA (RBF) | 4.14 | -0.34 | [-1.63, 0.96] | 0.610 |
| | unbiased CKA (RBF) | 4.14 | -0.33 | [-1.60, 0.93] | 0.608 |
| GSM8K | CKA (linear) | 4.03 | -0.27 | [-1.42, 0.88] | 0.643 |
| | unbiased CKA (linear) | 4.02 | -0.26 | [-1.40, 0.88] | 0.659 |
| | CKA (RBF) | 4.04 | -0.29 | [-1.44, 0.86] | 0.626 |
| | unbiased CKA (RBF) | 4.03 | -0.26 | [-1.39, 0.87] | 0.655 |
| MATH | CKA (linear) | 4.10 | -0.36 | [-1.48, 0.77] | 0.536 |
| | unbiased CKA (linear) | 4.09 | -0.34 | [-1.45, 0.77] | 0.548 |
| | CKA (RBF) | 4.12 | -0.37 | [-1.54, 0.79] | 0.532 |
| | unbiased CKA (RBF) | 4.10 | -0.35 | [-1.49, 0.80] | 0.553 |
| TruthfulQA | CKA (linear) | 4.15 | -0.49 | [-1.65, 0.66] | 0.403 |
| | unbiased CKA (linear) | 4.13 | -0.46 | [-1.61, 0.68] | 0.428 |
| | CKA (RBF) | 4.14 | -0.45 | [-1.61, 0.72] | 0.453 |
| | unbiased CKA (RBF) | 4.11 | -0.40 | [-1.54, 0.74] | 0.490 |

*Table 11.* Fictional biography generation task. We fit a mixed-effects regression between response uniqueness and representational similarity. Here, the summary similarity score is calculated using a global average. $\beta$ indicates the coefficient of similarity, and the 95% CI represents the confidence interval of $\beta$. **The table shows that representational similarity has a significant negative effect on response uniqueness.**

| Dataset | Metric | Intercept | $\beta$ | 95% CI | $p$ |
|---|---|---|---|---|---|
| WikiText | CKA (linear) | 8.14 | -1.28 | [-2.47, -0.10] | 0.034 |
| | unbiased CKA (linear) | 8.13 | -1.28 | [-2.45, -0.11] | 0.0327 |
| | CKA (RBF) | 8.21 | -1.42 | [-2.59, -0.24] | 0.0181 |
| | unbiased CKA (RBF) | 8.19 | -1.40 | [-2.55, -0.25] | 0.0169 |
| GSM8K | CKA (linear) | 7.94 | -1.75 | [-2.63, -0.87] | $9.3 \times 10^{-5}$ |
| | unbiased CKA (linear) | 7.92 | -1.76 | [-2.63, -0.90] | $6.7 \times 10^{-5}$ |
| | CKA (RBF) | 7.95 | -1.67 | [-2.55, -0.79] | $1.89 \times 10^{-4}$ |
| | unbiased CKA (RBF) | 7.92 | -1.69 | [-2.55, -0.83] | $1.16 \times 10^{-4}$ |
| MATH | CKA (linear) | 8.07 | -1.60 | [-2.52, -0.69] | $5.7 \times 10^{-4}$ |
| | unbiased CKA (linear) | 8.05 | -1.61 | [-2.51, -0.71] | $4.60 \times 10^{-4}$ |
| | CKA (RBF) | 8.12 | -1.61 | [-2.56, -0.65] | $9.47 \times 10^{-4}$ |
| | unbiased CKA (RBF) | 8.10 | -1.61 | [-2.54, -0.68] | $7.17 \times 10^{-4}$ |
| TruthfulQA | CKA (linear) | 7.92 | -1.29 | [-2.24, -0.35] | 0.0073 |
| | unbiased CKA (linear) | 7.91 | -1.33 | [-2.27, -0.40] | 0.00506 |
| | CKA (RBF) | 7.91 | -1.21 | [-2.17, -0.25] | 0.0134 |
| | unbiased CKA (RBF) | 7.90 | -1.26 | [-2.19, -0.32] | 0.00824 |

*Table 12.* Haiku composition task. We fit a mixed-effects regression between response uniqueness and representational similarity. Here, the summary similarity score is calculated using a global average. $\beta$ indicates the coefficient of similarity, and the 95% CI represents the confidence interval of $\beta$. **The table shows that representational similarity has a significant negative effect on response uniqueness.**

| Dataset | Metric | Intercept | $\beta$ | 95% CI | $p$ |
|---|---|---|---|---|---|
| WikiText | CKA (linear) | 5.95 | -3.42 | [-4.80, -2.05] | $1.1 \times 10^{-6}$ |
| | unbiased CKA (linear) | 5.93 | -3.42 | [-4.77, -2.07] | $6.71 \times 10^{-7}$ |
| | CKA (RBF) | 5.74 | -3.13 | [-4.56, -1.70] | $1.71 \times 10^{-5}$ |
| | unbiased CKA (RBF) | 5.70 | -3.10 | [-4.49, -1.70] | $1.39 \times 10^{-5}$ |
| GSM8K | CKA (linear) | 4.67 | -2.43 | [-3.70, -1.16] | $1.7 \times 10^{-4}$ |
| | unbiased CKA (linear) | 4.64 | -2.42 | [-3.68, -1.17] | 0.000158 |
| | CKA (RBF) | 4.65 | -2.22 | [-3.49, -0.96] | 0.000589 |
| | unbiased CKA (RBF) | 4.61 | -2.22 | [-3.46, -0.97] | 0.000477 |
| MATH | CKA (linear) | 5.24 | -3.12 | [-4.36, -1.89] | $7.2 \times 10^{-7}$ |
| | unbiased CKA (linear) | 5.22 | -3.13 | [-4.35, -1.92] | $4.48 \times 10^{-7}$ |
| | CKA (RBF) | 5.39 | -3.19 | [-4.45, -1.93] | $6.88 \times 10^{-7}$ |
| | unbiased CKA (RBF) | 5.33 | -3.17 | [-4.40, -1.94] | $4.60 \times 10^{-7}$ |
| TruthfulQA | CKA (linear) | 4.98 | -2.58 | [-3.84, -1.33] | $5.2 \times 10^{-5}$ |
| | unbiased CKA (linear) | 4.94 | -2.58 | [-3.83, -1.34] | $4.75 \times 10^{-5}$ |
| | CKA (RBF) | 4.84 | -2.12 | [-3.43, -0.81] | 0.00149 |
| | unbiased CKA (RBF) | 4.95 | -2.19 | [-3.83, -0.91] | 0.000794 |

*Table 13.* Vacation brainstorming task. We fit a mixed-effects regression between response uniqueness and representational similarity. Here, the summary similarity score is calculated using a global average. $\beta$ indicates the coefficient of similarity, and the 95% CI represents the confidence interval of $\beta$. **The table shows that representational similarity has a significant negative effect on response uniqueness.**

| Dataset | Metric | Intercept | $\beta$ | 95% CI | $p$ |
|---|---|---|---|---|---|
| WikiText | CKA (linear) | 1.66 | -1.01 | [-1.58, -0.45] | 0.00045 |
| | unbiased CKA (linear) | 1.65 | -1.00 | [-1.56, -0.44] | 0.000480 |
| | CKA (RBF) | 1.61 | -0.94 | [-1.55, -0.32] | 0.00287 |
| | unbiased CKA (RBF) | 1.59 | -0.91 | [-1.51, -0.31] | 0.00308 |
| GSM8K | CKA (linear) | 1.26 | -0.65 | [-1.32, 0.01] | 0.053 |
| | unbiased CKA (linear) | 1.25 | -0.64 | [-1.29, 0.02] | 0.0584 |
| | CKA (RBF) | 1.26 | -0.60 | [-1.26, 0.05] | 0.0721 |
| | unbiased CKA (RBF) | 1.24 | -0.57 | [-1.22, 0.07] | 0.0824 |
| MATH | CKA (linear) | 1.45 | -0.91 | [-1.50, -0.31] | 0.0028 |
| | unbiased CKA (linear) | 1.43 | -0.89 | [-1.48, -0.30] | 0.00298 |
| | CKA (RBF) | 1.45 | -0.87 | [-1.47, -0.27] | 0.00468 |
| | unbiased CKA (RBF) | 1.45 | -0.87 | [-1.47, -0.27] | 0.00468 |
| TruthfulQA | CKA (linear) | 1.35 | -0.71 | [-1.34, -0.08] | 0.0273 |
| | unbiased CKA (linear) | 1.34 | -0.69 | [-1.31, -0.07] | 0.0304 |
| | CKA (RBF) | 1.34 | -0.65 | [-1.27, -0.02] | 0.0438 |
| | unbiased CKA (RBF) | 1.32 | -0.62 | [-1.24, -0.01] | 0.0480 |

E.2.2. WHEN USING THE AVERAGE OF MAXIMUM-ALIGNED SCORES

*Table 14.* Story writing task. We fit a mixed-effects regression between response uniqueness and representational similarity. Here, the summary similarity score is calculated using the average of maximum-aligned scores. $\beta$ indicates the coefficient of similarity, and the 95% CI represents the confidence interval of $\beta$. **The table shows that representational similarity has a consistent negative effect on response uniqueness.**

| Dataset | Metric | Intercept | $\beta$ | 95% CI | $p$ |
|---------|--------|-----------|---------|--------|-----|
| WikiText | CKA (linear) | 4.31 | -0.49 | [-1.69, 0.71] | 0.422 |
| | unbiased CKA (linear) | 4.31 | -0.50 | [-1.68, 0.69] | 0.412 |
| | CKA (RBF) | 4.31 | -0.51 | [-1.66, 0.65] | 0.391 |
| | unbiased CKA (RBF) | 4.31 | -0.50 | [-1.68, 0.67] | 0.402 |
| GSM8K | CKA (linear) | 4.02 | -0.16 | [-0.88, 0.56] | 0.656 |
| | unbiased CKA (linear) | 4.02 | -0.16 | [-0.87, 0.55] | 0.662 |
| | CKA (RBF) | 4.03 | -0.17 | [-0.89, 0.55] | 0.641 |
| | unbiased CKA (RBF) | 4.04 | -0.18 | [-0.92, 0.56] | 0.632 |
| MATH | CKA (linear) | 4.11 | -0.29 | [-1.14, 0.57] | 0.512 |
| | unbiased CKA (linear) | 4.11 | -0.28 | [-1.13, 0.56] | 0.514 |
| | CKA (RBF) | 4.13 | -0.31 | [-1.20, 0.59] | 0.502 |
| | unbiased CKA (RBF) | 4.14 | -0.32 | [-1.23, 0.60] | 0.500 |
| TruthfulQA | CKA (linear) | 4.06 | -0.19 | [-1.01, 0.63] | 0.644 |
| | unbiased CKA (linear) | 4.05 | -0.19 | [-0.99, 0.61] | 0.641 |
| | CKA (RBF) | 4.05 | -0.19 | [-1.00, 0.62] | 0.646 |
| | unbiased CKA (RBF) | 4.06 | -0.19 | [-1.03, 0.65] | 0.654 |

*Table 15.* Fictional biography generation task. We fit a mixed-effects regression between response uniqueness and representational similarity. Here, the summary similarity score is calculated using the average of maximum-aligned scores. $\beta$ indicates the coefficient of similarity, and the 95% CI represents the confidence interval of $\beta$. **The table shows that representational similarity has a significant negative effect on response uniqueness.**

| Dataset | Metric | Intercept | $\beta$ | 95% CI | $p$ |
|---------|--------|-----------|---------|--------|-----|
| WikiText | CKA (linear) | 8.53 | -1.55 | [-2.47, -0.63] | 0.00094 |
| | unbiased CKA (linear) | 8.52 | -1.54 | [-2.45, -0.63] | 0.001 |
| | CKA (RBF) | 8.50 | -1.55 | [-2.43, -0.68] | 0.001 |
| | unbiased CKA (RBF) | 8.53 | -1.58 | [-2.47, -0.68] | 0.001 |
| GSM8K | CKA (linear) | 7.96 | -1.15 | [-1.67, -0.62] | $1.8 \times 10^{-5}$ |
| | unbiased CKA (linear) | 7.95 | -1.14 | [-1.66, -0.63] | $1.0 \times 10^{-5}$ |
| | CKA (RBF) | 7.96 | -1.15 | [-1.67, -0.62] | $2.0 \times 10^{-5}$ |
| | unbiased CKA (RBF) | 7.98 | -1.16 | [-1.70, -0.62] | $3.0 \times 10^{-5}$ |
| MATH | CKA (linear) | 8.17 | -1.31 | [-1.94, -0.69] | $4.3 \times 10^{-5}$ |
| | unbiased CKA (linear) | 8.15 | -1.31 | [-1.93, -0.69] | $4.0 \times 10^{-5}$ |
| | CKA (RBF) | 8.23 | -1.37 | [-2.03, -0.71] | $5.0 \times 10^{-5}$ |
| | unbiased CKA (RBF) | 8.25 | -1.39 | [-2.07, -0.71] | $6.0 \times 10^{-5}$ |
| TruthfulQA | CKA (linear) | 8.08 | -1.18 | [-1.79, -0.56] | 0.00016 |
| | unbiased CKA (linear) | 8.06 | -1.16 | [-1.76, -0.56] | $1.0 \times 10^{-4}$ |
| | CKA (RBF) | 8.04 | -1.12 | [-1.72, -0.51] | $3.0 \times 10^{-4}$ |
| | unbiased CKA (RBF) | 8.08 | -1.14 | [-1.78, -0.51] | $4.0 \times 10^{-4}$ |

*Table 16.* Haiku composition task. We fit a mixed-effects regression between response uniqueness and representational similarity. Here, the summary similarity score is calculated using the average of maximum-aligned scores. $\beta$ indicates the coefficient of similarity, and the 95% CI represents the confidence interval of $\beta$. **The table shows that representational similarity has a significant negative effect on response uniqueness.**

| Dataset | Metric | Intercept | $\beta$ | 95% CI | $p$ |
|---------|--------|-----------|---------|--------|-----|
| WikiText | CKA (linear) | 5.86 | -2.64 | [-3.95, -1.32] | $8.9 \times 10^{-5}$ |
| | unbiased CKA (linear) | 5.84 | -2.61 | [-3.92, -1.31] | $8.75 \times 10^{-5}$ |
| | CKA (RBF) | 5.62 | -2.37 | [-3.63, -1.11] | $2.33 \times 10^{-4}$ |
| | unbiased CKA (RBF) | 5.65 | -2.39 | [-3.68, -1.11] | $2.67 \times 10^{-4}$ |
| GSM8K | CKA (linear) | 4.62 | -1.44 | [-2.22, -0.66] | 0.00028 |
| | unbiased CKA (linear) | 4.60 | -1.43 | [-2.19, -0.66] | $2.57 \times 10^{-4}$ |
| | CKA (RBF) | 4.59 | -1.38 | [-2.16, -0.60] | $5.19 \times 10^{-4}$ |
| | unbiased CKA (RBF) | 4.62 | -1.40 | [-2.20, -0.60] | $6.01 \times 10^{-4}$ |
| MATH | CKA (linear) | 5.04 | -1.92 | [-2.84, -1.00] | $4.9 \times 10^{-5}$ |
| | unbiased CKA (linear) | 5.03 | -1.91 | [-2.83, -1.00] | $4.09 \times 10^{-5}$ |
| | CKA (RBF) | 5.11 | -1.97 | [-2.94, -1.00] | $6.83 \times 10^{-5}$ |
| | unbiased CKA (RBF) | 5.14 | -1.98 | [-2.98, -0.99] | $9.19 \times 10^{-5}$ |
| TruthfulQA | CKA (linear) | 4.88 | -1.65 | [-2.54, -0.76] | 0.00030 |
| | unbiased CKA (linear) | 4.86 | -1.64 | [-2.51, -0.76] | $2.43 \times 10^{-4}$ |
| | CKA (RBF) | 4.78 | -1.49 | [-2.37, -0.61] | $8.84 \times 10^{-4}$ |
| | unbiased CKA (RBF) | 4.82 | -1.51 | [-2.42, -0.59] | 0.0013 |

*Table 17.* Vacation brainstorming task. We fit a mixed-effects regression between response uniqueness and representational similarity. Here, the summary similarity score is calculated using the average of maximum-aligned scores. $\beta$ indicates the coefficient of similarity, and the 95% CI represents the confidence interval of $\beta$. **The table shows that representational similarity has a significant negative effect on response uniqueness.**

| Dataset | Metric | Intercept | $\beta$ | 95% CI | $p$ |
|---------|--------|-----------|---------|--------|-----|
| WikiText | CKA (linear) | 1.79 | -0.97 | [-1.65, -0.30] | 0.0046 |
| | unbiased CKA (linear) | 1.78 | -0.97 | [-1.63, -0.30] | 0.005 |
| | CKA (RBF) | 1.69 | -0.87 | [-1.52, -0.21] | 0.009 |
| | unbiased CKA (RBF) | 1.70 | -0.88 | [-1.55, -0.22] | 0.009 |
| GSM8K | CKA (linear) | 1.25 | -0.38 | [-0.80, 0.05] | 0.081 |
| | unbiased CKA (linear) | 1.24 | -0.37 | [-0.80, 0.05] | 0.083 |
| | CKA (RBF) | 1.25 | -0.37 | [-0.80, 0.05] | 0.083 |
| | unbiased CKA (RBF) | 1.26 | -0.39 | [-0.82, 0.05] | 0.080 |
| MATH | CKA (linear) | 1.44 | -0.64 | [-1.14, -0.14] | 0.012 |
| | unbiased CKA (linear) | 1.44 | -0.64 | [-1.13, -0.14] | 0.011 |
| | CKA (RBF) | 1.48 | -0.68 | [-1.20, -0.16] | 0.011 |
| | unbiased CKA (RBF) | 1.49 | -0.69 | [-1.22, -0.16] | 0.011 |
| TruthfulQA | CKA (linear) | 1.36 | -0.50 | [-0.98, -0.03] | 0.039 |
| | unbiased CKA (linear) | 1.35 | -0.50 | [-0.96, -0.03] | 0.036 |
| | CKA (RBF) | 1.36 | -0.51 | [-0.97, -0.04] | 0.032 |
| | unbiased CKA (RBF) | 1.37 | -0.52 | [-1.00, -0.03] | 0.036 |

### E.3. Mixed-effects regression results for mutual information calculated with `Llama-3.1-8B-instruct`

Tables 18~25 show significant positive trends across all datasets, CKA variants, and games.

#### E.3.1. WHEN USING THE GLOBAL AVERAGE

*Table 18.* Story writing task. We fit a mixed-effects regression between mutual information and representational similarity. Here, the summary similarity score is calculated using a global average. $\beta$ indicates the coefficient of similarity, and the 95% CI represents the confidence interval of $\beta$. **The table shows that representational similarity has a significant positive effect on mutual information.**

| Dataset | Metric | Intercept | $\beta$ | 95% CI | $p$ |
|---|---|---|---|---|---|
| WikiText | CKA (linear) | 0.303 | 0.053 | [0.020, 0.087] | 0.00194 |
| | unbiased CKA (linear) | 0.304 | 0.053 | [0.019, 0.086] | 0.00193 |
| | CKA (RBF) | 0.306 | 0.050 | [0.016, 0.084] | 0.00379 |
| | unbiased CKA (RBF) | 0.307 | 0.049 | [0.016, 0.082] | 0.00372 |
| GSM8K | CKA (linear) | 0.317 | 0.057 | [0.027, 0.087] | 0.00021 |
| | unbiased CKA (linear) | 0.318 | 0.057 | [0.027, 0.086] | 0.00020 |
| | CKA (RBF) | 0.316 | 0.055 | [0.025, 0.085] | 0.00035 |
| | unbiased CKA (RBF) | 0.318 | 0.054 | [0.025, 0.084] | 0.00032 |
| MATH | CKA (linear) | 0.313 | 0.052 | [0.022, 0.082] | 0.00058 |
| | unbiased CKA (linear) | 0.313 | 0.052 | [0.022, 0.081] | 0.00055 |
| | CKA (RBF) | 0.310 | 0.054 | [0.024, 0.085] | 0.00051 |
| | unbiased CKA (RBF) | 0.311 | 0.053 | [0.023, 0.083] | 0.00047 |
| TruthfulQA | CKA (linear) | 0.312 | 0.056 | [0.026, 0.086] | 0.00026 |
| | unbiased CKA (linear) | 0.312 | 0.057 | [0.027, 0.086] | 0.00019 |
| | CKA (RBF) | 0.312 | 0.053 | [0.023, 0.083] | 0.00062 |
| | unbiased CKA (RBF) | 0.313 | 0.053 | [0.024, 0.083] | 0.00040 |

*Table 19.* Fictional biography generation task. We fit a mixed-effects regression between mutual information and representational similarity. Here, the summary similarity score is calculated using a global average. $\beta$ indicates the coefficient of similarity, and the 95% CI represents the confidence interval of $\beta$. **The table shows that representational similarity has a significant positive effect on mutual information.**

| Dataset | Metric | Intercept | $\beta$ | 95% CI | $p$ |
|---|---|---|---|---|---|
| WikiText | CKA (linear) | 0.083 | 0.121 | [0.091, 0.150] | $5.07 \times 10^{-16}$ |
| | unbiased CKA (linear) | 0.084 | 0.120 | [0.091, 0.149] | $4.71 \times 10^{-16}$ |
| | CKA (RBF) | 0.084 | 0.119 | [0.090, 0.149] | $2.31 \times 10^{-15}$ |
| | unbiased CKA (RBF) | 0.087 | 0.117 | [0.088, 0.146] | $1.96 \times 10^{-15}$ |
| GSM8K | CKA (linear) | 0.123 | 0.099 | [0.076, 0.122] | $2.17 \times 10^{-17}$ |
| | unbiased CKA (linear) | 0.124 | 0.098 | [0.075, 0.121] | $2.08 \times 10^{-17}$ |
| | CKA (RBF) | 0.121 | 0.098 | [0.075, 0.121] | $6.39 \times 10^{-17}$ |
| | unbiased CKA (RBF) | 0.124 | 0.096 | [0.074, 0.119] | $5.27 \times 10^{-17}$ |
| MATH | CKA (linear) | 0.109 | 0.107 | [0.084, 0.131] | $1.39 \times 10^{-19}$ |
| | unbiased CKA (linear) | 0.110 | 0.106 | [0.083, 0.129] | $1.13 \times 10^{-19}$ |
| | CKA (RBF) | 0.103 | 0.111 | [0.086, 0.135] | $5.17 \times 10^{-19}$ |
| | unbiased CKA (RBF) | 0.106 | 0.108 | [0.085, 0.132] | $3.86 \times 10^{-19}$ |
| TruthfulQA | CKA (linear) | 0.117 | 0.091 | [0.067, 0.115] | $2.38 \times 10^{-13}$ |
| | unbiased CKA (linear) | 0.118 | 0.091 | [0.067, 0.115] | $1.03 \times 10^{-13}$ |
| | CKA (RBF) | 0.115 | 0.088 | [0.064, 0.113] | $2.44 \times 10^{-12}$ |
| | unbiased CKA (RBF) | 0.118 | 0.088 | [0.064, 0.112] | $6.81 \times 10^{-13}$ |

*Table 20.* Haiku composition task. We fit a mixed-effects regression between mutual information and representational similarity. Here, the summary similarity score is calculated using a global average. $\beta$ indicates the coefficient of similarity, and the 95% CI represents the confidence interval of $\beta$. **The table shows that representational similarity has a significant positive effect on mutual information.**

| Dataset | Metric | Intercept | $\beta$ | 95% CI | $p$ |
|---|---|---|---|---|---|
| WikiText | CKA (linear) | 0.594 | 1.310 | [1.03, 1.59] | $1.26 \times 10^{-20}$ |
| | unbiased CKA (linear) | 0.603 | 1.302 | [1.029, 1.575] | $9.20 \times 10^{-21}$ |
| | CKA (RBF) | 0.672 | 1.194 | [0.919, 1.470] | $2.03 \times 10^{-17}$ |
| | unbiased CKA (RBF) | 0.693 | 1.177 | [0.907, 1.447] | $1.28 \times 10^{-17}$ |
| GSM8K | CKA (linear) | 1.161 | 0.688 | [0.48, 0.89] | $6.18 \times 10^{-11}$ |
| | unbiased CKA (linear) | 1.171 | 0.679 | [0.475, 0.882] | $6.20 \times 10^{-11}$ |
| | CKA (RBF) | 1.152 | 0.670 | [0.463, 0.877] | $2.15 \times 10^{-10}$ |
| | unbiased CKA (RBF) | 1.168 | 0.656 | [0.454, 0.858] | $1.88 \times 10^{-10}$ |
| MATH | CKA (linear) | 1.017 | 0.845 | [0.63, 1.06] | $8.56 \times 10^{-15}$ |
| | unbiased CKA (linear) | 1.025 | 0.842 | [0.632, 1.053] | $4.48 \times 10^{-15}$ |
| | CKA (RBF) | 0.970 | 0.878 | [0.654, 1.101] | $1.33 \times 10^{-14}$ |
| | unbiased CKA (RBF) | 0.986 | 0.871 | [0.653, 1.089] | $4.73 \times 10^{-15}$ |
| TruthfulQA | CKA (linear) | 1.046 | 0.794 | [0.57, 1.02] | $2.62 \times 10^{-12}$ |
| | unbiased CKA (linear) | 1.054 | 0.804 | [0.585, 1.024] | $6.80 \times 10^{-13}$ |
| | CKA (RBF) | 1.039 | 0.766 | [0.539, 0.992] | $3.51 \times 10^{-11}$ |
| | unbiased CKA (RBF) | 1.053 | 0.777 | [0.557, 0.997] | $4.51 \times 10^{-12}$ |

*Table 21.* Vacation brainstorming task. We fit a mixed-effects regression between mutual information and representational similarity. Here, the summary similarity score is calculated using a global average. $\beta$ indicates the coefficient of similarity, and the 95% CI represents the confidence interval of $\beta$. **The table shows that representational similarity has a significant positive effect on mutual information.**

| Dataset | Metric | Intercept | $\beta$ | 95% CI | $p$ |
|---|---|---|---|---|---|
| WikiText | CKA (linear) | 0.587 | 0.132 | [0.071, 0.194] | $2.40 \times 10^{-5}$ |
| | unbiased CKA (linear) | 0.589 | 0.131 | [0.070, 0.192] | $2.43 \times 10^{-5}$ |
| | CKA (RBF) | 0.592 | 0.127 | [0.061, 0.192] | 0.00015 |
| | unbiased CKA (RBF) | 0.594 | 0.125 | [0.060, 0.189] | 0.00014 |
| GSM8K | CKA (linear) | 0.621 | 0.141 | [0.078, 0.203] | $9.48 \times 10^{-6}$ |
| | unbiased CKA (linear) | 0.623 | 0.140 | [0.079, 0.202] | $8.27 \times 10^{-6}$ |
| | CKA (RBF) | 0.619 | 0.140 | [0.077, 0.202] | $1.09 \times 10^{-5}$ |
| | unbiased CKA (RBF) | 0.621 | 0.139 | [0.078, 0.200] | $8.45 \times 10^{-6}$ |
| MATH | CKA (linear) | 0.602 | 0.149 | [0.090, 0.208] | $7.62 \times 10^{-7}$ |
| | unbiased CKA (linear) | 0.604 | 0.148 | [0.090, 0.206] | $6.08 \times 10^{-7}$ |
| | CKA (RBF) | 0.594 | 0.154 | [0.093, 0.215] | $7.74 \times 10^{-7}$ |
| | unbiased CKA (RBF) | 0.597 | 0.153 | [0.093, 0.212] | $5.49 \times 10^{-7}$ |
| TruthfulQA | CKA (linear) | 0.619 | 0.113 | [0.051, 0.175] | 0.00034 |
| | unbiased CKA (linear) | 0.621 | 0.112 | [0.051, 0.174] | 0.00032 |
| | CKA (RBF) | 0.618 | 0.108 | [0.046, 0.170] | 0.00068 |
| | unbiased CKA (RBF) | 0.622 | 0.107 | [0.046, 0.167] | 0.00060 |

### E.3.2. WHEN USING THE AVERAGE OF MAXIMUM-ALIGNED SCORES

*Table 22.* Story writing task. We fit a mixed-effects regression between mutual information and representational similarity. Here, the summary similarity score is calculated using the average of maximum-aligned scores. $\beta$ indicates the coefficient of similarity, and the 95% CI represents the confidence interval of $\beta$. **The table shows that representational similarity has a significant positive effect on mutual information.**

| Dataset | Metric | Intercept | $\beta$ | 95% CI | $p$ |
|---|---|---|---|---|---|
| WikiText | CKA (linear) | 0.272 | 0.085 | [0.053, 0.117] | $2.64 \times 10^{-7}$ |
| | unbiased CKA (linear) | 0.272 | 0.084 | [0.052, 0.116] | $2.62 \times 10^{-7}$ |
| | CKA (RBF) | 0.276 | 0.081 | [0.049, 0.112] | $5.54 \times 10^{-7}$ |
| | unbiased CKA (RBF) | 0.277 | 0.079 | [0.048, 0.110] | $5.23 \times 10^{-7}$ |
| GSM8K | CKA (linear) | 0.312 | 0.045 | [0.026, 0.065] | $4.39 \times 10^{-6}$ |
| | unbiased CKA (linear) | 0.313 | 0.045 | [0.026, 0.064] | $4.35 \times 10^{-6}$ |
| | CKA (RBF) | 0.311 | 0.046 | [0.026, 0.066] | $5.92 \times 10^{-6}$ |
| | unbiased CKA (RBF) | 0.312 | 0.045 | [0.025, 0.064] | $5.76 \times 10^{-6}$ |
| MATH | CKA (linear) | 0.301 | 0.057 | [0.034, 0.080] | $9.67 \times 10^{-7}$ |
| | unbiased CKA (linear) | 0.302 | 0.056 | [0.034, 0.079] | $9.72 \times 10^{-7}$ |
| | CKA (RBF) | 0.297 | 0.061 | [0.036, 0.085] | $1.09 \times 10^{-6}$ |
| | unbiased CKA (RBF) | 0.298 | 0.059 | [0.036, 0.083] | $1.09 \times 10^{-6}$ |
| TruthfulQA | CKA (linear) | 0.302 | 0.056 | [0.034, 0.078] | $5.79 \times 10^{-7}$ |
| | unbiased CKA (linear) | 0.303 | 0.055 | [0.033, 0.076] | $4.89 \times 10^{-7}$ |
| | CKA (RBF) | 0.301 | 0.055 | [0.033, 0.078] | $1.35 \times 10^{-6}$ |
| | unbiased CKA (RBF) | 0.303 | 0.054 | [0.032, 0.075] | $9.85 \times 10^{-7}$ |

*Table 23.* Fictional biography generation task. We fit a mixed-effects regression between mutual information and representational similarity. Here, the summary similarity score is calculated using the average of maximum-aligned scores. $\beta$ indicates the coefficient of similarity, and the 95% CI represents the confidence interval of $\beta$. **The table shows that representational similarity has a significant positive effect on mutual information.**

| Dataset | Metric | Intercept | $\beta$ | 95% CI | $p$ |
|---|---|---|---|---|---|
| WikiText | CKA (linear) | 0.089 | 0.088 | [0.064, 0.112] | $6.16 \times 10^{-13}$ |
| | unbiased CKA (linear) | 0.090 | 0.087 | [0.063, 0.111] | $7.49 \times 10^{-13}$ |
| | CKA (RBF) | 0.091 | 0.088 | [0.064, 0.111] | $1.86 \times 10^{-13}$ |
| | unbiased CKA (RBF) | 0.093 | 0.086 | [0.063, 0.109] | $2.40 \times 10^{-13}$ |
| GSM8K | CKA (linear) | 0.124 | 0.061 | [0.047, 0.075] | $2.48 \times 10^{-18}$ |
| | unbiased CKA (linear) | 0.125 | 0.060 | [0.047, 0.074] | $3.41 \times 10^{-18}$ |
| | CKA (RBF) | 0.121 | 0.064 | [0.050, 0.078] | $1.05 \times 10^{-18}$ |
| | unbiased CKA (RBF) | 0.123 | 0.062 | [0.048, 0.075] | $1.70 \times 10^{-18}$ |
| MATH | CKA (linear) | 0.112 | 0.071 | [0.055, 0.088] | $2.68 \times 10^{-17}$ |
| | unbiased CKA (linear) | 0.113 | 0.070 | [0.054, 0.086] | $4.29 \times 10^{-17}$ |
| | CKA (RBF) | 0.107 | 0.077 | [0.059, 0.094] | $2.40 \times 10^{-17}$ |
| | unbiased CKA (RBF) | 0.109 | 0.074 | [0.057, 0.091] | $5.19 \times 10^{-17}$ |
| TruthfulQA | CKA (linear) | 0.116 | 0.065 | [0.049, 0.081] | $2.18 \times 10^{-15}$ |
| | unbiased CKA (linear) | 0.118 | 0.063 | [0.047, 0.078] | $2.94 \times 10^{-15}$ |
| | CKA (RBF) | 0.114 | 0.065 | [0.049, 0.082] | $6.08 \times 10^{-15}$ |
| | unbiased CKA (RBF) | 0.118 | 0.063 | [0.047, 0.078] | $7.99 \times 10^{-15}$ |

*Table 24.* Haiku composition task. We fit a mixed-effects regression between mutual information and representational similarity. Here, the summary similarity score is calculated using the average of maximum-aligned scores. $\beta$ indicates the coefficient of similarity, and the 95% CI represents the confidence interval of $\beta$. **The table shows that representational similarity has a significant positive effect on mutual information.**

| Dataset | Metric | Intercept | $\beta$ | 95% CI | $p$ |
|---|---|---|---|---|---|
| WikiText | CKA (linear) | 0.652 | 0.973 | [0.760, 1.187] | $4.43 \times 10^{-19}$ |
| | unbiased CKA (linear) | 0.659 | 0.966 | [0.754, 1.178] | $3.84 \times 10^{-19}$ |
| | CKA (RBF) | 0.714 | 0.907 | [0.699, 1.115] | $1.29 \times 10^{-17}$ |
| | unbiased CKA (RBF) | 0.729 | 0.893 | [0.689, 1.097] | $9.53 \times 10^{-18}$ |
| GSM8K | CKA (linear) | 1.123 | 0.507 | [0.385, 0.629] | $3.05 \times 10^{-16}$ |
| | unbiased CKA (linear) | 1.131 | 0.501 | [0.381, 0.620] | $2.77 \times 10^{-16}$ |
| | CKA (RBF) | 1.111 | 0.513 | [0.387, 0.638] | $1.05 \times 10^{-15}$ |
| | unbiased CKA (RBF) | 1.125 | 0.500 | [0.378, 0.622] | $9.96 \times 10^{-16}$ |
| MATH | CKA (linear) | 0.999 | 0.635 | [0.489, 0.781] | $1.83 \times 10^{-17}$ |
| | unbiased CKA (linear) | 1.006 | 0.629 | [0.485, 0.774] | $1.35 \times 10^{-17}$ |
| | CKA (RBF) | 0.956 | 0.674 | [0.517, 0.832] | $4.69 \times 10^{-17}$ |
| | unbiased CKA (RBF) | 0.970 | 0.662 | [0.508, 0.816] | $3.13 \times 10^{-17}$ |
| TruthfulQA | CKA (linear) | 1.012 | 0.612 | [0.470, 0.754] | $2.98 \times 10^{-17}$ |
| | unbiased CKA (linear) | 1.023 | 0.604 | [0.466, 0.743] | $1.34 \times 10^{-17}$ |
| | CKA (RBF) | 1.005 | 0.606 | [0.460, 0.753] | $4.83 \times 10^{-16}$ |
| | unbiased CKA (RBF) | 1.026 | 0.592 | [0.452, 0.732] | $1.37 \times 10^{-16}$ |

*Table 25.* Vacation brainstorming task. We fit a mixed-effects regression between mutual information and representational similarity. Here, the summary similarity score is calculated using the average of maximum-aligned scores. $\beta$ indicates the coefficient of similarity, and the 95% CI represents the confidence interval of $\beta$. **The table shows that representational similarity has a significant positive effect on mutual information.**

| Dataset | Metric | Intercept | $\beta$ | 95% CI | $p$ |
|---|---|---|---|---|---|
| WikiText | CKA (linear) | 0.575 | 0.122 | [0.056, 0.188] | 0.00031 |
| | unbiased CKA (linear) | 0.577 | 0.121 | [0.055, 0.186] | 0.00031 |
| | CKA (RBF) | 0.586 | 0.110 | [0.045, 0.175] | 0.00090 |
| | unbiased CKA (RBF) | 0.588 | 0.108 | [0.044, 0.171] | 0.00097 |
| GSM8K | CKA (linear) | 0.626 | 0.079 | [0.039, 0.119] | $9.26 \times 10^{-5}$ |
| | unbiased CKA (linear) | 0.628 | 0.077 | [0.038, 0.116] | 0.00011 |
| | CKA (RBF) | 0.623 | 0.082 | [0.042, 0.123] | $7.27 \times 10^{-5}$ |
| | unbiased CKA (RBF) | 0.626 | 0.079 | [0.040, 0.119] | $8.69 \times 10^{-5}$ |
| MATH | CKA (linear) | 0.604 | 0.105 | [0.057, 0.152] | $1.38 \times 10^{-5}$ |
| | unbiased CKA (linear) | 0.605 | 0.103 | [0.057, 0.150] | $1.37 \times 10^{-5}$ |
| | CKA (RBF) | 0.597 | 0.111 | [0.061, 0.161] | $1.60 \times 10^{-5}$ |
| | unbiased CKA (RBF) | 0.599 | 0.109 | [0.060, 0.158] | $1.45 \times 10^{-5}$ |
| TruthfulQA | CKA (linear) | 0.613 | 0.089 | [0.043, 0.134] | 0.00012 |
| | unbiased CKA (linear) | 0.616 | 0.086 | [0.042, 0.130] | 0.00014 |
| | CKA (RBF) | 0.612 | 0.089 | [0.042, 0.135] | 0.00017 |
| | unbiased CKA (RBF) | 0.616 | 0.085 | [0.040, 0.129] | 0.00018 |

## E.4. Mixed-effects regression results for mutual information calculated with `Llama-3.1-8B`

Tables 26~33 show significant positive trends across all datasets, CKA variants, and games.

### E.4.1. WHEN USING THE GLOBAL AVERAGE

*Table 26.* Story writing task. We fit a mixed-effects regression between mutual information and representational similarity. Here, the summary similarity score is calculated using a global average. $\beta$ indicates the coefficient of similarity, and the 95% CI represents the confidence interval of $\beta$. **The table shows that representational similarity has a significant positive effect on mutual information.**

| Dataset | Metric | Intercept | $\beta$ | 95% CI | $p$ |
|---|---|---|---|---|---|
| WikiText | CKA (linear) | 0.363 | 0.088 | [0.037, 0.139] | 0.00075 |
| | unbiased CKA (linear) | 0.363 | 0.087 | [0.037, 0.138] | 0.00074 |
| | CKA (RBF) | 0.372 | 0.072 | [0.026, 0.119] | 0.00238 |
| | unbiased CKA (RBF) | 0.374 | 0.071 | [0.026, 0.117] | 0.00227 |
| GSM8K | CKA (linear) | 0.389 | 0.081 | [0.045, 0.118] | $1.39 \times 10^{-5}$ |
| | unbiased CKA (linear) | 0.390 | 0.081 | [0.045, 0.117] | $1.20 \times 10^{-5}$ |
| | CKA (RBF) | 0.389 | 0.077 | [0.040, 0.113] | $3.60 \times 10^{-5}$ |
| | unbiased CKA (RBF) | 0.390 | 0.077 | [0.041, 0.113] | $2.76 \times 10^{-5}$ |
| MATH | CKA (linear) | 0.381 | 0.078 | [0.041, 0.116] | $4.25 \times 10^{-5}$ |
| | unbiased CKA (linear) | 0.382 | 0.078 | [0.041, 0.115] | $3.90 \times 10^{-5}$ |
| | CKA (RBF) | 0.378 | 0.079 | [0.040, 0.118] | $6.24 \times 10^{-5}$ |
| | unbiased CKA (RBF) | 0.380 | 0.078 | [0.040, 0.116] | $5.45 \times 10^{-5}$ |
| TruthfulQA | CKA (linear) | 0.382 | 0.079 | [0.040, 0.118] | $6.21 \times 10^{-5}$ |
| | unbiased CKA (linear) | 0.383 | 0.080 | [0.042, 0.119] | $4.33 \times 10^{-5}$ |
| | CKA (RBF) | 0.383 | 0.071 | [0.033, 0.110] | 0.00030 |
| | unbiased CKA (RBF) | 0.385 | 0.072 | [0.035, 0.110] | 0.00018 |

*Table 27.* Fictional biography generation task. We fit a mixed-effects regression between mutual information and representational similarity. Here, the summary similarity score is calculated using a global average. $\beta$ indicates the coefficient of similarity, and the 95% CI represents the confidence interval of $\beta$. **The table shows that representational similarity has a significant positive effect on mutual information.**

| Dataset | Metric | Intercept | $\beta$ | 95% CI | $p$ |
|---|---|---|---|---|---|
| WikiText | CKA (linear) | 0.135 | 0.130 | [0.098, 0.162] | $1.88 \times 10^{-15}$ |
| | unbiased CKA (linear) | 0.136 | 0.129 | [0.097, 0.161] | $1.70 \times 10^{-15}$ |
| | CKA (RBF) | 0.136 | 0.130 | [0.0981, 0.163] | $2.96 \times 10^{-15}$ |
| | unbiased CKA (RBF) | 0.138 | 0.128 | [0.097, 0.160] | $2.12 \times 10^{-15}$ |
| GSM8K | CKA (linear) | 0.175 | 0.119 | [0.094, 0.144] | $5.79 \times 10^{-21}$ |
| | unbiased CKA (linear) | 0.177 | 0.118 | [0.093, 0.142] | $5.51 \times 10^{-21}$ |
| | CKA (RBF) | 0.173 | 0.118 | [0.093, 0.143] | $1.74 \times 10^{-20}$ |
| | unbiased CKA (RBF) | 0.176 | 0.116 | [0.091, 0.140] | $1.37 \times 10^{-20}$ |
| MATH | CKA (linear) | 0.162 | 0.118 | [0.093, 0.144] | $7.24 \times 10^{-20}$ |
| | unbiased CKA (linear) | 0.164 | 0.117 | [0.092, 0.142] | $6.08 \times 10^{-20}$ |
| | CKA (RBF) | 0.155 | 0.123 | [0.097, 0.150] | $9.91 \times 10^{-20}$ |
| | unbiased CKA (RBF) | 0.158 | 0.121 | [0.095, 0.147] | $7.60 \times 10^{-20}$ |
| TruthfulQA | CKA (linear) | 0.169 | 0.106 | [0.079, 0.132] | $5.53 \times 10^{-15}$ |
| | unbiased CKA (linear) | 0.170 | 0.106 | [0.080, 0.132] | $2.24 \times 10^{-15}$ |
| | CKA (RBF) | 0.166 | 0.105 | [0.078, 0.132] | $3.05 \times 10^{-14}$ |
| | unbiased CKA (RBF) | 0.169 | 0.104 | [0.078, 0.130] | $7.51 \times 10^{-15}$ |

*Table 28.* Haiku composition task. We fit a mixed-effects regression between mutual information and representational similarity. Here, the summary similarity score is calculated using a global average. $\beta$ indicates the coefficient of similarity, and the 95% CI represents the confidence interval of $\beta$. **The table shows that representational similarity has a significant positive effect on mutual information.**

| Dataset | Metric | Intercept | $\beta$ | 95% CI | $p$ |
|---|---|---|---|---|---|
| WikiText | CKA (linear) | 0.628 | 1.335 | [1.033, 1.638] | $5.17 \times 10^{-18}$ |
| | unbiased CKA (linear) | 0.637 | 1.329 | [1.029, 1.629] | $3.54 \times 10^{-18}$ |
| | CKA (RBF) | 0.700 | 1.230 | [0.928, 1.533] | $1.71 \times 10^{-15}$ |
| | unbiased CKA (RBF) | 0.720 | 1.215 | [0.918, 1.511] | $1.02 \times 10^{-15}$ |
| GSM8K | CKA (linear) | 1.209 | 0.693 | [0.466, 0.920] | $2.27 \times 10^{-9}$ |
| | unbiased CKA (linear) | 1.218 | 0.684 | [0.460, 0.909] | $2.23 \times 10^{-9}$ |
| | CKA (RBF) | 1.200 | 0.675 | [0.447, 0.903] | $6.47 \times 10^{-9}$ |
| | unbiased CKA (RBF) | 1.216 | 0.662 | [0.439, 0.884] | $5.65 \times 10^{-9}$ |
| MATH | CKA (linear) | 1.058 | 0.863 | [0.628, 1.098] | $6.30 \times 10^{-13}$ |
| | unbiased CKA (linear) | 1.066 | 0.861 | [0.629, 1.093] | $3.39 \times 10^{-13}$ |
| | CKA (RBF) | 1.011 | 0.897 | [0.651, 1.143] | $8.97 \times 10^{-13}$ |
| | unbiased CKA (RBF) | 1.026 | 0.891 | [0.651, 1.131] | $3.44 \times 10^{-13}$ |
| TruthfulQA | CKA (linear) | 1.098 | 0.791 | [0.545, 1.036] | $2.65 \times 10^{-10}$ |
| | unbiased CKA (linear) | 1.103 | 0.804 | [0.562, 1.046] | $7.32 \times 10^{-11}$ |
| | CKA (RBF) | 1.090 | 0.762 | [0.512, 1.012] | $2.22 \times 10^{-9}$ |
| | unbiased CKA (RBF) | 1.103 | 0.777 | [0.535, 1.020] | $3.34 \times 10^{-10}$ |

*Table 29.* Vacation brainstorming task. We fit a mixed-effects regression between mutual information and representational similarity. Here, the summary similarity score is calculated using a global average. $\beta$ indicates the coefficient of similarity, and the 95% CI represents the confidence interval of $\beta$. **The table shows that representational similarity has a significant positive effect on mutual information.**

| Dataset | Metric | Intercept | $\beta$ | 95% CI | $p$ |
|---|---|---|---|---|---|
| WikiText | CKA (linear) | 0.696 | 0.158 | [0.094, 0.222] | $1.40 \times 10^{-6}$ |
| | unbiased CKA (linear) | 0.698 | 0.155 | [0.094, 0.215] | $6.16 \times 10^{-7}$ |
| | CKA (RBF) | 0.698 | 0.156 | [0.087, 0.224] | $8.20 \times 10^{-6}$ |
| | unbiased CKA (RBF) | 0.701 | 0.153 | [0.086, 0.220] | $7.34 \times 10^{-6}$ |
| GSM8K | CKA (linear) | 0.733 | 0.177 | [0.112, 0.243] | $1.20 \times 10^{-7}$ |
| | unbiased CKA (linear) | 0.735 | 0.177 | [0.112, 0.242] | $9.84 \times 10^{-8}$ |
| | CKA (RBF) | 0.730 | 0.175 | [0.110, 0.241] | $1.83 \times 10^{-7}$ |
| | unbiased CKA (RBF) | 0.734 | 0.175 | [0.110, 0.239] | $1.25 \times 10^{-7}$ |
| MATH | CKA (linear) | 0.713 | 0.179 | [0.117, 0.241] | $1.39 \times 10^{-8}$ |
| | unbiased CKA (linear) | 0.715 | 0.179 | [0.118, 0.240] | $1.01 \times 10^{-8}$ |
| | CKA (RBF) | 0.703 | 0.186 | [0.121, 0.251] | $1.84 \times 10^{-8}$ |
| | unbiased CKA (RBF) | 0.707 | 0.184 | [0.121, 0.247] | $1.13 \times 10^{-8}$ |
| TruthfulQA | CKA (linear) | 0.731 | 0.142 | [0.078, 0.206] | $1.33 \times 10^{-5}$ |
| | unbiased CKA (linear) | 0.733 | 0.142 | [0.079, 0.206] | $1.08 \times 10^{-5}$ |
| | CKA (RBF) | 0.729 | 0.138 | [0.073, 0.203] | $3.04 \times 10^{-5}$ |
| | unbiased CKA (RBF) | 0.733 | 0.137 | [0.074, 0.200] | $2.09 \times 10^{-5}$ |

E.4.2. WHEN USING THE AVERAGE OF MAXIMUM-ALIGNED SCORES

*Table 30.* Story writing task. We fit a mixed-effects regression between mutual information and representational similarity. Here, the summary similarity score is calculated using the average of maximum-aligned scores. $\beta$ indicates the coefficient of similarity, and the 95% CI represents the confidence interval of $\beta$. **The table shows that representational similarity has a significant positive effect on mutual information.**

| Dataset | Metric | Intercept | $\beta$ | 95% CI | $p$ |
|---------|--------|-----------|---------|--------|-----|
| WikiText | CKA (linear) | 0.317 | 0.131 | [0.090, 0.171] | $2.92 \times 10^{-10}$ |
| | unbiased CKA (linear) | 0.318 | 0.129 | [0.089, 0.169] | $3.11 \times 10^{-10}$ |
| | CKA (RBF) | 0.327 | 0.120 | [0.081, 0.159] | $2.00 \times 10^{-9}$ |
| | unbiased CKA (RBF) | 0.329 | 0.118 | [0.079, 0.156] | $1.96 \times 10^{-9}$ |
| GSM8K | CKA (linear) | 0.380 | 0.069 | [0.046, 0.092] | $2.98 \times 10^{-9}$ |
| | unbiased CKA (linear) | 0.381 | 0.068 | [0.046, 0.091] | $2.99 \times 10^{-9}$ |
| | CKA (RBF) | 0.378 | 0.070 | [0.046, 0.093] | $6.20 \times 10^{-9}$ |
| | unbiased CKA (RBF) | 0.380 | 0.068 | [0.045, 0.091] | $5.88 \times 10^{-9}$ |
| MATH | CKA (linear) | 0.362 | 0.088 | [0.060, 0.115] | $4.06 \times 10^{-10}$ |
| | unbiased CKA (linear) | 0.364 | 0.086 | [0.059, 0.113] | $4.40 \times 10^{-10}$ |
| | CKA (RBF) | 0.357 | 0.092 | [0.063, 0.121] | $8.61 \times 10^{-10}$ |
| | unbiased CKA (RBF) | 0.359 | 0.090 | [0.061, 0.118] | $9.81 \times 10^{-10}$ |
| TruthfulQA | CKA (linear) | 0.363 | 0.086 | [0.059, 0.113] | $2.37 \times 10^{-10}$ |
| | unbiased CKA (linear) | 0.365 | 0.084 | [0.058, 0.110] | $2.25 \times 10^{-10}$ |
| | CKA (RBF) | 0.363 | 0.085 | [0.057, 0.112] | $1.46 \times 10^{-9}$ |
| | unbiased CKA (RBF) | 0.366 | 0.082 | [0.055, 0.108] | $1.16 \times 10^{-9}$ |

*Table 31.* Fictional biography generation task. We fit a mixed-effects regression between mutual information and representational similarity. Here, the summary similarity score is calculated using the average of maximum-aligned scores. $\beta$ indicates the coefficient of similarity, and the 95% CI represents the confidence interval of $\beta$. **The table shows that representational similarity has a significant positive effect on mutual information.**

| Dataset | Metric | Intercept | $\beta$ | 95% CI | $p$ |
|---------|--------|-----------|---------|--------|-----|
| WikiText | CKA (linear) | 0.143 | 0.094 | [0.068, 0.120] | $1.60 \times 10^{-12}$ |
| | unbiased CKA (linear) | 0.144 | 0.093 | [0.067, 0.119] | $1.83 \times 10^{-12}$ |
| | CKA (RBF) | 0.144 | 0.095 | [0.069, 0.120] | $2.72 \times 10^{-13}$ |
| | unbiased CKA (RBF) | 0.146 | 0.092 | [0.068, 0.117] | $3.19 \times 10^{-13}$ |
| GSM8K | CKA (linear) | 0.178 | 0.069 | [0.054, 0.084] | $5.27 \times 10^{-20}$ |
| | unbiased CKA (linear) | 0.179 | 0.068 | [0.054, 0.083] | $6.93 \times 10^{-20}$ |
| | CKA (RBF) | 0.175 | 0.072 | [0.057, 0.088] | $1.65 \times 10^{-20}$ |
| | unbiased CKA (RBF) | 0.177 | 0.070 | [0.055, 0.085] | $2.42 \times 10^{-20}$ |
| MATH | CKA (linear) | 0.167 | 0.077 | [0.059, 0.095] | $3.49 \times 10^{-17}$ |
| | unbiased CKA (linear) | 0.168 | 0.075 | [0.058, 0.093] | $5.06 \times 10^{-17}$ |
| | CKA (RBF) | 0.161 | 0.083 | [0.064, 0.103] | $1.85 \times 10^{-17}$ |
| | unbiased CKA (RBF) | 0.163 | 0.081 | [0.062, 0.100] | $3.36 \times 10^{-17}$ |
| TruthfulQA | CKA (linear) | 0.170 | 0.071 | [0.054, 0.089] | $6.60 \times 10^{-16}$ |
| | unbiased CKA (linear) | 0.172 | 0.070 | [0.053, 0.086] | $7.44 \times 10^{-16}$ |
| | CKA (RBF) | 0.168 | 0.072 | [0.054, 0.090] | $1.93 \times 10^{-15}$ |
| | unbiased CKA (RBF) | 0.172 | 0.069 | [0.052, 0.086] | $1.85 \times 10^{-15}$ |

*Table 32.* Haiku composition task. We fit a mixed-effects regression between mutual information and representational similarity. Here, the summary similarity score is calculated using the average of maximum-aligned scores. $\beta$ indicates the coefficient of similarity, and the 95% CI represents the confidence interval of $\beta$. **The table shows that representational similarity has a significant positive effect on mutual information.**

| Dataset | Metric | Intercept | $\beta$ | 95% CI | $p$ |
|---|---|---|---|---|---|
| WikiText | CKA (linear) | 0.664 | 1.023 | [0.788, 1.259] | $1.73 \times 10^{-17}$ |
| | unbiased CKA (linear) | 0.672 | 1.015 | [0.782, 1.249] | $1.55 \times 10^{-17}$ |
| | CKA (RBF) | 0.726 | 0.957 | [0.728, 1.187] | $2.83 \times 10^{-16}$ |
| | unbiased CKA (RBF) | 0.743 | 0.942 | [0.717, 1.167] | $2.20 \times 10^{-16}$ |
| GSM8K | CKA (linear) | 1.161 | 0.530 | [0.395, 0.664] | $9.96 \times 10^{-15}$ |
| | unbiased CKA (linear) | 1.169 | 0.523 | [0.391, 0.655] | $8.96 \times 10^{-15}$ |
| | CKA (RBF) | 1.149 | 0.535 | [0.397, 0.673] | $3.24 \times 10^{-14}$ |
| | unbiased CKA (RBF) | 1.164 | 0.522 | [0.387, 0.656] | $3.02 \times 10^{-14}$ |
| MATH | CKA (linear) | 1.031 | 0.664 | [0.503, 0.826] | $7.19 \times 10^{-16}$ |
| | unbiased CKA (linear) | 1.039 | 0.658 | [0.499, 0.817] | $5.55 \times 10^{-16}$ |
| | CKA (RBF) | 0.985 | 0.706 | [0.532, 0.879] | $1.57 \times 10^{-15}$ |
| | unbiased CKA (RBF) | 1.000 | 0.693 | [0.523, 0.862] | $1.14 \times 10^{-15}$ |
| TruthfulQA | CKA (linear) | 1.049 | 0.632 | [0.475, 0.788] | $2.49 \times 10^{-15}$ |
| | unbiased CKA (linear) | 1.061 | 0.624 | [0.471, 0.777] | $1.22 \times 10^{-15}$ |
| | CKA (RBF) | 1.042 | 0.628 | [0.466, 0.789] | $2.55 \times 10^{-14}$ |
| | unbiased CKA (RBF) | 1.063 | 0.613 | [0.458, 0.768] | $8.33 \times 10^{-15}$ |

*Table 33.* Vacation brainstorming task. We fit a mixed-effects regression between mutual information and representational similarity. Here, the summary similarity score is calculated using the average of maximum-aligned scores. $\beta$ indicates the coefficient of similarity, and the 95% CI represents the confidence interval of $\beta$. **The table shows that representational similarity has a significant positive effect on mutual information.**

| Dataset | Metric | Intercept | $\beta$ | 95% CI | $p$ |
|---|---|---|---|---|---|
| WikiText | CKA (linear) | 0.654 | 0.182 | [0.114, 0.249] | $1.36 \times 10^{-7}$ |
| | unbiased CKA (linear) | 0.656 | 0.180 | [0.113, 0.247] | $1.40 \times 10^{-7}$ |
| | CKA (RBF) | 0.666 | 0.169 | [0.102, 0.236] | $7.80 \times 10^{-7}$ |
| | unbiased CKA (RBF) | 0.670 | 0.165 | [0.099, 0.231] | $8.44 \times 10^{-7}$ |
| GSM8K | CKA (linear) | 0.731 | 0.117 | [0.076, 0.159] | $3.98 \times 10^{-8}$ |
| | unbiased CKA (linear) | 0.733 | 0.115 | [0.074, 0.157] | $4.77 \times 10^{-8}$ |
| | CKA (RBF) | 0.726 | 0.121 | [0.078, 0.165] | $3.29 \times 10^{-8}$ |
| | unbiased CKA (RBF) | 0.730 | 0.118 | [0.075, 0.160] | $4.21 \times 10^{-8}$ |
| MATH | CKA (linear) | 0.701 | 0.148 | [0.098, 0.197] | $4.32 \times 10^{-9}$ |
| | unbiased CKA (linear) | 0.703 | 0.146 | [0.097, 0.195] | $4.16 \times 10^{-9}$ |
| | CKA (RBF) | 0.691 | 0.158 | [0.105, 0.211] | $4.75 \times 10^{-9}$ |
| | unbiased CKA (RBF) | 0.694 | 0.155 | [0.103, 0.207] | $4.03 \times 10^{-9}$ |
| TruthfulQA | CKA (linear) | 0.712 | 0.131 | [0.083, 0.178] | $5.71 \times 10^{-8}$ |
| | unbiased CKA (linear) | 0.715 | 0.127 | [0.081, 0.173] | $6.40 \times 10^{-8}$ |
| | CKA (RBF) | 0.709 | 0.132 | [0.083, 0.180] | $8.78 \times 10^{-8}$ |
| | unbiased CKA (RBF) | 0.715 | 0.126 | [0.080, 0.173] | $8.95 \times 10^{-8}$ |

## E.5. Mixed-effects regression results for mutual information, excluding the Llama family

Tables 34∼41 show significant positive trends across all datasets, CKA variants, and games.

### E.5.1. WHEN USING THE GLOBAL AVERAGE

*Table 34.* Story writing task. We fit a mixed-effects regression between mutual information and representational similarity. Here, the summary similarity score is calculated using a global average. $\beta$ indicates the coefficient of similarity, and the 95% CI represents the confidence interval of $\beta$. **The table shows that representational similarity has a significant positive effect on mutual information.**

| Dataset | Metric | Intercept | $\beta$ | 95% CI | $p$ |
|---|---|---|---|---|---|
| WikiText | CKA (linear) | 0.307 | 0.058 | [0.020, 0.095] | 0.00246 |
| | unbiased CKA (linear) | 0.308 | 0.058 | [0.021, 0.095] | 0.00232 |
| | CKA (RBF) | 0.311 | 0.052 | [0.015, 0.090] | 0.00633 |
| | unbiased CKA (RBF) | 0.312 | 0.052 | [0.015, 0.089] | 0.00584 |
| GSM8K | CKA (linear) | 0.323 | 0.058 | [0.024, 0.092] | 0.00083 |
| | unbiased CKA (linear) | 0.324 | 0.058 | [0.024, 0.092] | 0.00076 |
| | CKA (RBF) | 0.323 | 0.056 | [0.022, 0.090] | 0.00130 |
| | unbiased CKA (RBF) | 0.324 | 0.056 | [0.022, 0.089] | 0.00113 |
| MATH | CKA (linear) | 0.320 | 0.051 | [0.018, 0.083] | 0.00212 |
| | unbiased CKA (linear) | 0.320 | 0.051 | [0.019, 0.083] | 0.00192 |
| | CKA (RBF) | 0.317 | 0.053 | [0.019, 0.086] | 0.00193 |
| | unbiased CKA (RBF) | 0.318 | 0.053 | [0.020, 0.085] | 0.00165 |
| TruthfulQA | CKA (linear) | 0.318 | 0.056 | [0.023, 0.090] | 0.00097 |
| | unbiased CKA (linear) | 0.319 | 0.057 | [0.024, 0.090] | 0.00067 |
| | CKA (RBF) | 0.319 | 0.053 | [0.020, 0.087] | 0.00186 |
| | unbiased CKA (RBF) | 0.319 | 0.054 | [0.022, 0.087] | 0.00116 |

*Table 35.* Fictional biography generation task. We fit a mixed-effects regression between mutual information and representational similarity. Here, the summary similarity score is calculated using a global average. $\beta$ indicates the coefficient of similarity, and the 95% CI represents the confidence interval of $\beta$. **The table shows that representational similarity has a significant positive effect on mutual information.**

| Dataset | Metric | Intercept | $\beta$ | 95% CI | $p$ |
|---|---|---|---|---|---|
| WikiText | CKA (linear) | 0.078 | 0.138 | [0.105, 0.171] | $5.0 \times 10^{-16}$ |
| | unbiased CKA (linear) | 0.080 | 0.136 | [0.103, 0.169] | $4.8 \times 10^{-16}$ |
| | CKA (RBF) | 0.081 | 0.135 | [0.102, 0.169] | $2.2 \times 10^{-15}$ |
| | unbiased CKA (RBF) | 0.084 | 0.133 | [0.100, 0.166] | $1.9 \times 10^{-15}$ |
| GSM8K | CKA (linear) | 0.121 | 0.121 | [0.095, 0.148] | $6.0 \times 10^{-19}$ |
| | unbiased CKA (linear) | 0.123 | 0.120 | [0.094, 0.147] | $5.0 \times 10^{-19}$ |
| | CKA (RBF) | 0.119 | 0.120 | [0.093, 0.147] | $1.7 \times 10^{-18}$ |
| | unbiased CKA (RBF) | 0.122 | 0.118 | [0.092, 0.145] | $1.1 \times 10^{-18}$ |
| MATH | CKA (linear) | 0.108 | 0.120 | [0.094, 0.146] | $3.2 \times 10^{-19}$ |
| | unbiased CKA (linear) | 0.109 | 0.119 | [0.093, 0.145] | $2.5 \times 10^{-19}$ |
| | CKA (RBF) | 0.102 | 0.124 | [0.097, 0.152] | $1.0 \times 10^{-18}$ |
| | unbiased CKA (RBF) | 0.104 | 0.122 | [0.095, 0.149] | $6.8 \times 10^{-19}$ |
| TruthfulQA | CKA (linear) | 0.116 | 0.107 | [0.079, 0.135] | $4.3 \times 10^{-14}$ |
| | unbiased CKA (linear) | 0.117 | 0.107 | [0.079, 0.134] | $2.3 \times 10^{-14}$ |
| | CKA (RBF) | 0.115 | 0.103 | [0.075, 0.132] | $6.4 \times 10^{-13}$ |
| | unbiased CKA (RBF) | 0.118 | 0.103 | [0.075, 0.130] | $2.2 \times 10^{-13}$ |

*Table 36.* Haiku composition task. We fit a mixed-effects regression between mutual information and representational similarity. Here, the summary similarity score is calculated using a global average. $\beta$ indicates the coefficient of similarity, and the 95% CI represents the confidence interval of $\beta$. **The table shows that representational similarity has a significant positive effect on mutual information.**

| Dataset | Metric | Intercept | $\beta$ | 95% CI | $p$ |
|---|---|---|---|---|---|
| WikiText | CKA (linear) | 0.497 | 1.536 | [1.242, 1.831] | $1.7 \times 10^{-24}$ |
| | unbiased CKA (linear) | 0.510 | 1.522 | [1.231, 1.814] | $1.4 \times 10^{-24}$ |
| | CKA (RBF) | 0.590 | 1.404 | [1.108, 1.700] | $1.5 \times 10^{-20}$ |
| | unbiased CKA (RBF) | 0.615 | 1.381 | [1.092, 1.670] | $7.1 \times 10^{-21}$ |
| GSM8K | CKA (linear) | 1.106 | 0.958 | [0.729, 1.188] | $2.6 \times 10^{-16}$ |
| | unbiased CKA (linear) | 1.117 | 0.951 | [0.724, 1.178] | $2.2 \times 10^{-16}$ |
| | CKA (RBF) | 1.092 | 0.941 | [0.712, 1.171] | $9.8 \times 10^{-16}$ |
| | unbiased CKA (RBF) | 1.111 | 0.929 | [0.704, 1.154] | $6.4 \times 10^{-16}$ |
| MATH | CKA (linear) | 0.978 | 1.001 | [0.772, 1.230] | $1.0 \times 10^{-17}$ |
| | unbiased CKA (linear) | 0.987 | 0.996 | [0.770, 1.222] | $5.8 \times 10^{-18}$ |
| | CKA (RBF) | 0.913 | 1.061 | [0.820, 1.302] | $5.5 \times 10^{-18}$ |
| | unbiased CKA (RBF) | 0.932 | 1.048 | [0.813, 1.283] | $2.2 \times 10^{-18}$ |
| TruthfulQA | CKA (linear) | 0.991 | 1.014 | [0.772, 1.256] | $2.1 \times 10^{-16}$ |
| | unbiased CKA (linear) | 1.003 | 1.018 | [0.780, 1.257] | $5.8 \times 10^{-17}$ |
| | CKA (RBF) | 0.987 | 0.972 | [0.727, 1.217] | $7.6 \times 10^{-15}$ |
| | unbiased CKA (RBF) | 1.008 | 0.975 | [0.737, 1.213] | $9.7 \times 10^{-16}$ |

*Table 37.* Vacation brainstorming task. We fit a mixed-effects regression between mutual information and representational similarity. Here, the summary similarity score is calculated using a global average. $\beta$ indicates the coefficient of similarity, and the 95% CI represents the confidence interval of $\beta$. **The table shows that representational similarity has a significant positive effect on mutual information.**

| Dataset | Metric | Intercept | $\beta$ | 95% CI | $p$ |
|---|---|---|---|---|---|
| WikiText | CKA (linear) | 0.592 | 0.185 | [0.119, 0.250] | $3.2 \times 10^{-8}$ |
| | unbiased CKA (linear) | 0.593 | 0.183 | [0.118, 0.248] | $3.2 \times 10^{-8}$ |
| | CKA (RBF) | 0.597 | 0.179 | [0.107, 0.251] | $1.1 \times 10^{-6}$ |
| | unbiased CKA (RBF) | 0.600 | 0.176 | [0.105, 0.246] | $1.0 \times 10^{-6}$ |
| GSM8K | CKA (linear) | 0.644 | 0.180 | [0.108, 0.251] | $7.8 \times 10^{-7}$ |
| | unbiased CKA (linear) | 0.646 | 0.179 | [0.108, 0.250] | $7.3 \times 10^{-7}$ |
| | CKA (RBF) | 0.642 | 0.176 | [0.105, 0.247] | $1.2 \times 10^{-6}$ |
| | unbiased CKA (RBF) | 0.645 | 0.175 | [0.105, 0.246] | $9.7 \times 10^{-7}$ |
| MATH | CKA (linear) | 0.623 | 0.182 | [0.117, 0.247] | $4.3 \times 10^{-8}$ |
| | unbiased CKA (linear) | 0.624 | 0.181 | [0.117, 0.245] | $3.3 \times 10^{-8}$ |
| | CKA (RBF) | 0.613 | 0.188 | [0.120, 0.255] | $4.5 \times 10^{-8}$ |
| | unbiased CKA (RBF) | 0.616 | 0.186 | [0.120, 0.252] | $2.8 \times 10^{-8}$ |
| TruthfulQA | CKA (linear) | 0.638 | 0.154 | [0.083, 0.224] | $1.9 \times 10^{-5}$ |
| | unbiased CKA (linear) | 0.640 | 0.153 | [0.084, 0.223] | $1.7 \times 10^{-5}$ |
| | CKA (RBF) | 0.638 | 0.146 | [0.075, 0.216] | $5.6 \times 10^{-5}$ |
| | unbiased CKA (RBF) | 0.642 | 0.145 | [0.076, 0.214] | $4.1 \times 10^{-5}$ |

E.5.2. WHEN USING THE AVERAGE OF MAXIMUM-ALIGNED SCORES

*Table 38.* Story writing task. We fit a mixed-effects regression between mutual information and representational similarity. Here, the summary similarity score is calculated using the average of maximum-aligned scores. $\beta$ indicates the coefficient of similarity, and the 95% CI represents the confidence interval of $\beta$. **The table shows that representational similarity has a significant positive effect on mutual information.**

| Dataset | Metric | Intercept | $\beta$ | 95% CI | $p$ |
|---|---|---|---|---|---|
| WikiText | CKA (linear) | 0.275 | 0.090 | [0.054, 0.125] | $5.87 \times 10^{-7}$ |
| | unbiased CKA (linear) | 0.275 | 0.089 | [0.054, 0.124] | $5.71 \times 10^{-7}$ |
| | CKA (RBF) | 0.281 | 0.083 | [0.049, 0.118] | $1.94 \times 10^{-6}$ |
| | unbiased CKA (RBF) | 0.282 | 0.082 | [0.048, 0.116] | $1.81 \times 10^{-6}$ |
| GSM8K | CKA (linear) | 0.318 | 0.047 | [0.026, 0.068] | $1.40 \times 10^{-5}$ |
| | unbiased CKA (linear) | 0.318 | 0.047 | [0.026, 0.068] | $1.36 \times 10^{-5}$ |
| | CKA (RBF) | 0.317 | 0.048 | [0.026, 0.069] | $1.99 \times 10^{-5}$ |
| | unbiased CKA (RBF) | 0.318 | 0.047 | [0.025, 0.068] | $1.89 \times 10^{-5}$ |
| MATH | CKA (linear) | 0.306 | 0.059 | [0.034, 0.084] | $3.48 \times 10^{-6}$ |
| | unbiased CKA (linear) | 0.307 | 0.059 | [0.034, 0.083] | $3.46 \times 10^{-6}$ |
| | CKA (RBF) | 0.302 | 0.063 | [0.036, 0.090] | $4.00 \times 10^{-6}$ |
| | unbiased CKA (RBF) | 0.303 | 0.062 | [0.035, 0.088] | $3.92 \times 10^{-6}$ |
| TruthfulQA | CKA (linear) | 0.307 | 0.058 | [0.034, 0.082] | $1.75 \times 10^{-6}$ |
| | unbiased CKA (linear) | 0.308 | 0.058 | [0.034, 0.081] | $1.43 \times 10^{-6}$ |
| | CKA (RBF) | 0.306 | 0.058 | [0.033, 0.082] | $3.66 \times 10^{-6}$ |
| | unbiased CKA (RBF) | 0.308 | 0.056 | [0.033, 0.080] | $2.63 \times 10^{-6}$ |

*Table 39.* Fictional biography generation task. We fit a mixed-effects regression between mutual information and representational similarity. Here, the summary similarity score is calculated using the average of maximum-aligned scores. $\beta$ indicates the coefficient of similarity, and the 95% CI represents the confidence interval of $\beta$. **The table shows that representational similarity has a significant positive effect on mutual information.**

| Dataset | Metric | Intercept | $\beta$ | 95% CI | $p$ |
|---|---|---|---|---|---|
| WikiText | CKA (linear) | 0.086 | 0.099 | [0.072, 0.126] | $3.39 \times 10^{-13}$ |
| | unbiased CKA (linear) | 0.087 | 0.098 | [0.072, 0.124] | $3.67 \times 10^{-13}$ |
| | CKA (RBF) | 0.089 | 0.097 | [0.071, 0.123] | $1.93 \times 10^{-13}$ |
| | unbiased CKA (RBF) | 0.091 | 0.095 | [0.070, 0.121] | $2.12 \times 10^{-13}$ |
| GSM8K | CKA (linear) | 0.124 | 0.070 | [0.055, 0.086] | $4.29 \times 10^{-19}$ |
| | unbiased CKA (linear) | 0.125 | 0.069 | [0.054, 0.085] | $5.20 \times 10^{-19}$ |
| | CKA (RBF) | 0.121 | 0.073 | [0.057, 0.089] | $1.94 \times 10^{-19}$ |
| | unbiased CKA (RBF) | 0.123 | 0.071 | [0.056, 0.087] | $2.50 \times 10^{-19}$ |
| MATH | CKA (linear) | 0.112 | 0.080 | [0.062, 0.098] | $1.85 \times 10^{-17}$ |
| | unbiased CKA (linear) | 0.113 | 0.079 | [0.061, 0.097] | $2.34 \times 10^{-17}$ |
| | CKA (RBF) | 0.106 | 0.086 | [0.066, 0.106] | $2.03 \times 10^{-17}$ |
| | unbiased CKA (RBF) | 0.108 | 0.084 | [0.064, 0.103] | $3.01 \times 10^{-17}$ |
| TruthfulQA | CKA (linear) | 0.116 | 0.073 | [0.055, 0.091] | $9.57 \times 10^{-16}$ |
| | unbiased CKA (linear) | 0.118 | 0.071 | [0.054, 0.089] | $9.84 \times 10^{-16}$ |
| | CKA (RBF) | 0.115 | 0.073 | [0.055, 0.091] | $4.75 \times 10^{-15}$ |
| | unbiased CKA (RBF) | 0.118 | 0.070 | [0.053, 0.088] | $4.28 \times 10^{-15}$ |

*Table 40.* Haiku composition task. We fit a mixed-effects regression between mutual information and representational similarity. Here, the summary similarity score is calculated using the average of maximum-aligned scores. $\beta$ indicates the coefficient of similarity, and the 95% CI represents the confidence interval of $\beta$. **The table shows that representational similarity has a significant positive effect on mutual information.**

| Dataset | Metric | Intercept | $\beta$ | 95% CI | $p$ |
|---|---|---|---|---|---|
| WikiText | CKA (linear) | 0.628 | 1.048 | [0.823, 1.274] | $8.91 \times 10^{-20}$ |
| | unbiased CKA (linear) | 0.637 | 1.040 | [0.816, 1.264] | $8.54 \times 10^{-20}$ |
| | CKA (RBF) | 0.702 | 0.970 | [0.750, 1.189] | $4.29 \times 10^{-18}$ |
| | unbiased CKA (RBF) | 0.719 | 0.953 | [0.738, 1.168] | $3.67 \times 10^{-18}$ |
| GSM8K | CKA (linear) | 1.104 | 0.602 | [0.472, 0.733] | $1.22 \times 10^{-19}$ |
| | unbiased CKA (linear) | 1.112 | 0.595 | [0.466, 0.723] | $1.24 \times 10^{-19}$ |
| | CKA (RBF) | 1.088 | 0.613 | [0.479, 0.747] | $3.39 \times 10^{-19}$ |
| | unbiased CKA (RBF) | 1.103 | 0.599 | [0.468, 0.730] | $3.27 \times 10^{-19}$ |
| MATH | CKA (linear) | 0.984 | 0.710 | [0.554, 0.865] | $3.73 \times 10^{-19}$ |
| | unbiased CKA (linear) | 0.992 | 0.701 | [0.547, 0.855] | $3.93 \times 10^{-19}$ |
| | CKA (RBF) | 0.934 | 0.756 | [0.589, 0.923] | $8.06 \times 10^{-19}$ |
| | unbiased CKA (RBF) | 0.951 | 0.739 | [0.575, 0.903] | $9.36 \times 10^{-19}$ |
| TruthfulQA | CKA (linear) | 0.990 | 0.701 | [0.551, 0.852] | $6.29 \times 10^{-20}$ |
| | unbiased CKA (linear) | 1.006 | 0.686 | [0.539, 0.833] | $6.31 \times 10^{-20}$ |
| | CKA (RBF) | 0.986 | 0.693 | [0.538, 0.847] | $1.48 \times 10^{-18}$ |
| | unbiased CKA (RBF) | 1.013 | 0.669 | [0.521, 0.817] | $9.74 \times 10^{-19}$ |

*Table 41.* Vacation brainstorming task. We fit a mixed-effects regression between mutual information and representational similarity. Here, the summary similarity score is calculated using the average of maximum-aligned scores. $\beta$ indicates the coefficient of similarity, and the 95% CI represents the confidence interval of $\beta$. **The table shows that representational similarity has a significant positive effect on mutual information.**

| Dataset | Metric | Intercept | $\beta$ | 95% CI | $p$ |
|---|---|---|---|---|---|
| WikiText | CKA (linear) | 0.582 | 0.161 | [0.086, 0.235] | $2.21 \times 10^{-5}$ |
| | unbiased CKA (linear) | 0.583 | 0.159 | [0.086, 0.233] | $2.22 \times 10^{-5}$ |
| | CKA (RBF) | 0.596 | 0.144 | [0.072, 0.216] | $9.66 \times 10^{-5}$ |
| | unbiased CKA (RBF) | 0.599 | 0.141 | [0.070, 0.212] | $9.68 \times 10^{-5}$ |
| GSM8K | CKA (linear) | 0.651 | 0.099 | [0.055, 0.142] | $9.04 \times 10^{-6}$ |
| | unbiased CKA (linear) | 0.653 | 0.097 | [0.054, 0.140] | $1.01 \times 10^{-5}$ |
| | CKA (RBF) | 0.648 | 0.102 | [0.057, 0.147] | $8.23 \times 10^{-6}$ |
| | unbiased CKA (RBF) | 0.650 | 0.099 | [0.055, 0.143] | $9.33 \times 10^{-6}$ |
| MATH | CKA (linear) | 0.624 | 0.129 | [0.077, 0.181] | $1.13 \times 10^{-6}$ |
| | unbiased CKA (linear) | 0.625 | 0.128 | [0.077, 0.180] | $9.97 \times 10^{-7}$ |
| | CKA (RBF) | 0.615 | 0.137 | [0.081, 0.193] | $1.34 \times 10^{-6}$ |
| | unbiased CKA (RBF) | 0.617 | 0.136 | [0.081, 0.190] | $9.86 \times 10^{-7}$ |
| TruthfulQA | CKA (linear) | 0.633 | 0.114 | [0.064, 0.164] | $7.39 \times 10^{-6}$ |
| | unbiased CKA (linear) | 0.635 | 0.112 | [0.063, 0.161] | $7.23 \times 10^{-6}$ |
| | CKA (RBF) | 0.632 | 0.113 | [0.062, 0.163] | $1.43 \times 10^{-5}$ |
| | unbiased CKA (RBF) | 0.637 | 0.109 | [0.060, 0.158] | $1.25 \times 10^{-5}$ |

### E.6. Mixed-effects regression results obtained by controlling for performance disparity

*Table 42.* Results of the mixed-effects regression controlling for performance disparity between the two models. Performance disparity is measured as the difference in their MMLU scores. The summary similarity score is computed as the global average representational similarity obtained using linear CKA on WikiText. Here, $\beta$ denotes the regression coefficient, and the 95% CI refers to the 95% confidence interval of $\beta$. **The table shows that representational similarity has a significant effect on cooperation and novelty even when controlling for performance disparity. This implies that the trend is not merely a byproduct of performance disparities.**

| Game | Predictor | $\beta$ | 95% CI | $p$ |
|---|---|---|---|---|
| Word Guessing | Representational Similarity | 6.957 | [5.160, 8.754] | $< 0.001$ |
| | Performance Disparity | -0.829 | [-1.710, 0.053] | 0.066 |
| Public Good | Representational Similarity | 48.846 | [27.680, 70.012] | $< 0.001$ |
| | Performance Disparity | -14.418 | [-30.133, 1.297] | 0.072 |
| Divide-a-Dollar | Representational Similarity | 0.452 | [0.221, 0.684] | $< 0.001$ |
| | Performance Disparity | 0.032 | [-0.110, 0.174] | 0.656 |
| KBC | Representational Similarity | 13.406 | [-0.762, 27.574] | 0.064 |
| | Performance Disparity | -22.757 | [-33.156, -12.357] | $< 0.001$ |
| Story (Uniqueness) | Representational Similarity | -0.268 | [-1.455, 0.919] | 0.658 |
| | Performance Disparity | 0.320 | [-0.635, 1.275] | 0.511 |
| Biography (Uniqueness) | Representational Similarity | -0.696 | [-1.814, 0.423] | 0.223 |
| | Performance Disparity | 1.580 | [0.868, 2.293] | $< 0.001$ |
| Haiku (Uniqueness) | Representational Similarity | -3.488 | [-4.810, -2.167] | $< 0.001$ |
| | Performance Disparity | -0.012 | [-1.052, 1.028] | 0.982 |
| Vacation (Uniqueness) | Representational Similarity | -1.046 | [-1.625, -0.468] | $< 0.001$ |
| | Performance Disparity | -0.414 | [-0.955, 0.128] | 0.134 |
| Story (MI) | Representational Similarity | 0.054 | [0.019, 0.088] | 0.002 |
| | Performance Disparity | 0.001 | [-0.025, 0.028] | 0.935 |
| Biography (MI) | Representational Similarity | 0.124 | [0.094, 0.154] | $< 0.001$ |
| | Performance Disparity | 0.009 | [-0.010, 0.027] | 0.363 |
| Haiku (MI) | Representational Similarity | 1.353 | [1.069, 1.637] | $< 0.001$ |
| | Performance Disparity | 0.107 | [-0.059, 0.273] | 0.208 |
| Vacation (MI) | Representational Similarity | 0.130 | [0.068, 0.192] | $< 0.001$ |
| | Performance Disparity | -0.014 | [-0.066, 0.038] | 0.600 |

## E.7. Mixed-effects regression results for each layer group

*Table 43.* Results of the mixed-effects regression for the early, middle, and late layer groups. The summary similarity score is computed as the global average representational similarity obtained using linear CKA on WikiText within each corresponding layer group. Here, $\beta$ denotes the regression coefficient, and the 95% CI refers to the 95% confidence interval of $\beta$. **The table shows that representational similarity within the early layer group exhibits the strongest effect on cooperation and novelty. This implies that shared basic lexical–semantic grounding is a central factor underlying increased cooperation and reduced novelty.**

| Game | Layer | $\beta$ | 95% CI | $p$ |
|---|---|---|---|---|
| Word Guessing | Early | 10.488 | [7.843, 13.133] | $< 0.001$ |
| | Middle | 4.515 | [3.400, 5.629] | $< 0.001$ |
| | Late | 3.490 | [2.592, 4.388] | $< 0.001$ |
| Public Good | Early | 73.298 | [44.482, 102.113] | $< 0.001$ |
| | Middle | 32.803 | [17.927, 47.679] | $< 0.001$ |
| | Late | 30.089 | [17.215, 42.962] | $< 0.001$ |
| Divide-a-Dollar | Early | 0.600 | [0.282, 0.917] | $< 0.001$ |
| | Middle | 0.287 | [0.139, 0.436] | $< 0.001$ |
| | Late | 0.142 | [0.020, 0.264] | 0.023 |
| KBC | Early | 21.823 | [3.930, 39.717] | 0.017 |
| | Middle | 12.136 | [2.929, 21.343] | 0.010 |
| | Late | 14.237 | [6.002, 22.472] | 0.001 |
| Story (Uniqueness) | Early | -0.694 | [-2.379, 0.991] | 0.419 |
| | Middle | -0.249 | [-1.134, 0.636] | 0.581 |
| | Late | -0.064 | [-0.850, 0.723] | 0.874 |
| Biography (Uniqueness) | Early | -0.760 | [-2.278, 0.758] | 0.326 |
| | Middle | -0.849 | [-1.612, -0.085] | 0.029 |
| | Late | -1.165 | [-1.792, -0.539] | $< 0.001$ |
| Haiku (Uniqueness) | Early | -4.371 | [-6.222, -2.519] | $< 0.001$ |
| | Middle | -2.230 | [-3.205, -1.255] | $< 0.001$ |
| | Late | -1.947 | [-2.809, -1.086] | $< 0.001$ |
| Vacation (Uniqueness) | Early | -1.444 | [-2.142, -0.746] | $< 0.001$ |
| | Middle | -0.774 | [-1.205, -0.343] | $< 0.001$ |
| | Late | -0.563 | [-0.999, -0.127] | 0.011 |
| Story (MI) | Early | 0.056 | [0.012, 0.101] | 0.013 |
| | Middle | 0.047 | [0.023, 0.072] | $< 0.001$ |
| | Late | 0.048 | [0.027, 0.069] | $< 0.001$ |
| Biography (MI) | Early | 0.164 | [0.122, 0.206] | $< 0.001$ |
| | Middle | 0.076 | [0.057, 0.095] | $< 0.001$ |
| | Late | 0.061 | [0.045, 0.077] | $< 0.001$ |
| Haiku (MI) | Early | 1.871 | [1.479, 2.263] | $< 0.001$ |
| | Middle | 0.811 | [0.634, 0.987] | $< 0.001$ |
| | Late | 0.639 | [0.496, 0.781] | $< 0.001$ |
| Vacation (MI) | Early | 0.171 | [0.088, 0.253] | $< 0.001$ |
| | Middle | 0.087 | [0.041, 0.134] | $< 0.001$ |
| | Late | 0.091 | [0.048, 0.134] | $< 0.001$ |

# F. When temperature is set to 0.3

## F.1. Mixed-effects regression results

*Table 44.* Results of the mixed-effects regression when the temperature is set to 0.3. The summary similarity score is computed as the global average representational similarity obtained using linear CKA on WikiText. Here, $\beta$ denotes the regression coefficient, and the 95% CI refers to the 95% confidence interval of $\beta$. **The table shows that representational similarity has a significant positive effect on cooperation and a significant negative effect on novelty.**

| Game | $\beta$ | 95% CI | $p$ |
|---|---|---|---|
| Word Guessing | 7.292 | [5.580, 9.005] | < 0.001 |
| Public Good | 41.963 | [23.146, 60.781] | < 0.001 |
| Divide-a-Dollar | 0.468 | [0.243, 0.692] | < 0.001 |
| KBC | 14.990 | [2.486, 27.493] | 0.019 |
| Story (Uniqueness) | -1.145 | [-2.182, -0.108] | 0.030 |
| Biography (Uniqueness) | -1.952 | [-3.588, -0.317] | 0.019 |
| Haiku (Uniqueness) | -3.095 | [-4.395, -1.795] | < 0.001 |
| Vacation (Uniqueness) | -0.507 | [-0.899, -0.114] | 0.011 |
| Story (MI) | 0.041 | [0.004, 0.077] | 0.028 |
| Biography (MI) | 0.172 | [0.138, 0.205] | < 0.001 |
| Haiku (MI) | 1.396 | [1.129, 1.664] | < 0.001 |
| Vacation (MI) | 0.197 | [0.128, 0.266] | < 0.001 |

## F.2. Mixed-effects regression results obtained by controlling for performance disparity

*Table 45.* Results of the mixed-effects regression controlling for performance disparity between the two models. Performance disparity is measured as the difference in their MMLU scores. The summary similarity score is computed as the global average representational similarity obtained using linear CKA on WikiText. Here, $\beta$ denotes the regression coefficient, and the 95% CI refers to the 95% confidence interval of $\beta$. **The table shows that representational similarity has a significant effect on cooperation and novelty even when controlling for performance disparity. This implies that the trend is not merely a byproduct of performance disparities.**

| Game | Predictor | $\beta$ | 95% CI | $p$ |
|---|---|---|---|---|
| Word Guessing | Representational Similarity | 7.215 | [5.480, 8.949] | < 0.001 |
| | Performance Disparity | -0.238 | [-1.093, 0.618] | 0.586 |
| Public Good | Representational Similarity | 39.995 | [21.262, 58.728] | < 0.001 |
| | Performance Disparity | -9.267 | [-23.770, 5.236] | 0.210 |
| Divide-a-Dollar | Representational Similarity | 0.508 | [0.278, 0.738] | < 0.001 |
| | Performance Disparity | 0.098 | [-0.036, 0.233] | 0.153 |
| KBC | Representational Similarity | 14.032 | [1.092, 26.973] | 0.034 |
| | Performance Disparity | -4.857 | [-15.397, 5.682] | 0.366 |
| Story (Uniqueness) | Representational Similarity | -1.038 | [-2.086, 0.010] | 0.052 |
| | Performance Disparity | 0.488 | [-0.310, 1.285] | 0.230 |
| Biography (Uniqueness) | Representational Similarity | -1.853 | [-3.720, 0.014] | 0.052 |
| | Performance Disparity | 1.312 | [0.239, 2.385] | 0.017 |
| Haiku (Uniqueness) | Representational Similarity | -3.022 | [-4.351, -1.693] | < 0.001 |
| | Performance Disparity | 0.274 | [-0.649, 1.198] | 0.560 |
| Vacation (Uniqueness) | Representational Similarity | -0.496 | [-0.890, -0.102] | 0.014 |
| | Performance Disparity | 0.145 | [-0.256, 0.546] | 0.478 |
| Story (MI) | Representational Similarity | 0.044 | [0.006, 0.082] | 0.023 |
| | Performance Disparity | 0.011 | [-0.016, 0.038] | 0.435 |
| Biography (MI) | Representational Similarity | 0.166 | [0.132, 0.200] | < 0.001 |
| | Performance Disparity | -0.017 | [-0.038, 0.004] | 0.120 |
| Haiku (MI) | Representational Similarity | 1.352 | [1.077, 1.628] | < 0.001 |
| | Performance Disparity | -0.105 | [-0.263, 0.054] | 0.196 |
| Vacation (MI) | Representational Similarity | 0.179 | [0.109, 0.248] | < 0.001 |
| | Performance Disparity | -0.086 | [-0.136, -0.036] | 0.001 |

## F.3. Mixed-effects regression results obtained by controlling for other design factors

*Table 46.* Results of the mixed-effects regression controlling for other design factors. The summary similarity score is computed as the global average representational similarity obtained using linear CKA on WikiText. The predictors are rescaled to $[0, 1]$ to enable comparison of effect sizes. The outcome variables are standardized to $z-$scores, and game type is included as a control variable to account for systematic differences across games. The reported value is the regression coefficients for each predictor, and the values in parentheses are the corresponding p-values. **The table shows that the effect of representational similarity remains robust even when controlling for other model-relevant factors. Moreover, the effect size of similarity is the strongest, compared to the other factors.**

|  | Cooperation | Uniqueness | Mutual information |
|---|---|---|---|
| Representational Similarity | 0.336 (0.011) | -0.443 (0.023) | 0.391 ($<$0.001) |
| Size difference | -0.213 ($<$0.001) | 0.114 (0.350) | -0.039 (0.332) |
| Model family | 0.014 (0.730) | 0.085 (0.322) | 0.100 ($<$0.001) |
| Tokenizer | 0.060 (0.041) | -0.066 (0.309) | 0.016 (0.435) |
| Same model | -0.033 (0.523) | -0.188 (0.079) | 0.016 (0.646) |

## F.4. Mixed-effects regression results for each layer group

*Table 47.* Results of the mixed-effects regression for the early, middle, and late layer groups. The summary similarity score is computed as the global average representational similarity obtained using linear CKA on WikiText within each corresponding layer group. Here, $\beta$ denotes the regression coefficient, and the 95% CI refers to the 95% confidence interval of $\beta$. **The table shows that representational similarity within the early layer group exhibits the strongest effect on cooperation and novelty. This implies that shared basic lexical–semantic grounding is a central factor underlying increased cooperation and reduced novelty.**

| Game | Layer | $\beta$ | 95% CI | $p$ |
|---|---|---|---|---|
| Word Guessing | Early | 10.376 | [7.831, 12.921] | < 0.001 |
| | Middle | 4.532 | [3.456, 5.609] | < 0.001 |
| | Late | 3.510 | [2.640, 4.379] | < 0.001 |
| Public Good | Early | 54.792 | [29.218, 80.366] | < 0.001 |
| | Middle | 26.578 | [13.220, 39.936] | < 0.001 |
| | Late | 23.928 | [11.980, 35.877] | < 0.001 |
| Divide-a-Dollar | Early | 0.603 | [0.281, 0.925] | < 0.001 |
| | Middle | 0.274 | [0.129, 0.419] | < 0.001 |
| | Late | 0.165 | [0.049, 0.281] | 0.005 |
| KBC | Early | 20.039 | [3.372, 36.705] | 0.018 |
| | Middle | 9.693 | [0.540, 18.847] | 0.038 |
| | Late | 10.711 | [2.369, 19.054] | 0.012 |
| Story (Uniqueness) | Early | -1.884 | [-3.274, -0.494] | 0.008 |
| | Middle | -0.719 | [-1.457, 0.018] | 0.056 |
| | Late | -0.574 | [-1.230, 0.083] | 0.087 |
| Biography (Uniqueness) | Early | -1.411 | [-3.622, 0.800] | 0.211 |
| | Middle | -1.149 | [-2.217, -0.080] | 0.035 |
| | Late | -1.750 | [-2.619, -0.882] | < 0.001 |
| Haiku (Uniqueness) | Early | -3.822 | [-5.613, -2.031] | < 0.001 |
| | Middle | -1.859 | [-2.788, -0.930] | < 0.001 |
| | Late | -1.258 | [-2.056, -0.460] | 0.002 |
| Vacation (Uniqueness) | Early | -0.963 | [-1.419, -0.507] | < 0.001 |
| | Middle | -0.411 | [-0.710, -0.112] | 0.007 |
| | Late | -0.319 | [-0.630, -0.007] | 0.045 |
| Story (MI) | Early | 0.037 | [-0.011, 0.085] | 0.130 |
| | Middle | 0.034 | [0.009, 0.059] | 0.008 |
| | Late | 0.032 | [0.010, 0.054] | 0.004 |
| Biography (MI) | Early | 0.244 | [0.196, 0.293] | < 0.001 |
| | Middle | 0.117 | [0.095, 0.138] | < 0.001 |
| | Late | 0.085 | [0.066, 0.103] | < 0.001 |
| Haiku (MI) | Early | 2.163 | [1.778, 2.547] | < 0.001 |
| | Middle | 0.872 | [0.701, 1.042] | < 0.001 |
| | Late | 0.667 | [0.531, 0.804] | < 0.001 |
| Vacation (MI) | Early | 0.246 | [0.150, 0.342] | < 0.001 |
| | Middle | 0.136 | [0.089, 0.182] | < 0.001 |
| | Late | 0.140 | [0.099, 0.181] | < 0.001 |

