# OpenReview forum: "Representational Similarity and Model Behavior in Multi-Agent Interaction"
_ICML.cc/2026/Conference — ICML 2026 regular_

### Official Review · Reviewer_iZeP · 2026-02-15

**Soundness:** 3
**Presentation:** 3
**Significance:** 3
**Originality:** 3
**Overall Recommendation:** 4
**Confidence:** 3

**Summary:**

This paper asks an intriguing question: is the representational similarity between models a predictor of their tendency to cooperate, and do interaction among models with similar representations lead to increased homogeneity of collective outputs? these are important questions, and this paper does a good job of tackling them. They use CKA to measure RS between 23 models that are sorted into 276 pairs to play four different cooperative games. The authors make a serious attempt at ruling out potentialy confounds (e.g. behavioural similarity, model family or size difference) and show that the RS effects even once these have been modelled as covariates.

**Compliance With Llm Reviewing Policy:**

Affirmed.

**Key Questions For Authors:**

1. If this is not due to behavioural similarity, what is causing it?
2. Which representations need to be similar for the effect to occur? The authors show that this is primarily driven by the early layers, which I found surprising - why might this be?
3. How does this result relate to generalised homophily (e.g. models preferring their own outputs or those of similar models). is that the mediating variable?

**Limitations:**

yes

**Strengths And Weaknesses:**

general comments
- the authors chose CKA to compute RS using a bunch of standardised tests. It's great that the RS computation is independent of the main task. CKA is probably a sensible choice, and it's great that the authors tried two different variants. I did wonder whether the results genealised to another way of computing RS however, or whether the authros could say anything about *what is was* about the similarity of representations (and which representations?) that drove this effect.
- the choices of task seemed a bit arbitrary to me. If I had been planning this experiment, I would deliberately have chosen both games in which cooperation involves similar responses (like stag hunt or matching pennies) and those where it involves different responses (like chicken or battle of the sexes). That would have allowed them to study head on the issue of whether the causal chain is similar RS -> behavioural similarity -> cooperation or not.
- I get that the authors regress out other variables (e.g. behavioural simliarity per se, similarity in model size, family etc) but this analysis assumes that the effect of these confounders on cooperation is linear; if they fully accounted for the results but showed more complex trends then this could allow for the "residual" effect of RS. It would be stronger to control for these factors by design, where possible.
- perhaps this is overly demanding but for such a big claim I would have loved to have seen a causal intervention. If you take two identical models (same size, family, etc CKA = 1 to start), and fine-tune them (on an independent task) under a diversity penalty, do they become less cooperative? this would be the killer finding for me, would significantly increase the impact of this paper I think

---

> ### Author Rebuttal · Authors · 2026-03-31
>
> We thank Reviewer iZeP for the constructive feedback!
>
> ### **Different RS metrics and which aspects of representational similarity drove the effect**
>
> We carefully chose CKA given existing literature. For robustness, we computed two variants and used four different probe datasets, yielding many combinations. That said, as the reviewer notes, different metrics capture different aspects of similarity; for instance, while CKA captures global structure, metrics like Mutual k-NN capture local structure. It is possible that the observed trend would not generalize to all similarity metrics. Understanding which metrics best predict interaction outcomes, and under what conditions, is an interesting direction we leave for future work.
>
> ### **Task choices**
>
> We chose these games because they are canonical tools for studying cooperation, as described in the related work (Appendix A; we will move this to the main body in the camera-ready version, as ICML allows an extra page). Regarding the reviewer's suggested alternatives, chicken and battle of the sexes are also symmetric games. In these games, when one player chooses one response and the other chooses the opposite response, they are in fact exhibiting the same underlying behavioral tendency. As a result, these games would not cleanly separate the causal chain of similar RS → behavioral similarity → cooperation in the way the reviewer envisions.
>
> ### **Linear relationships with other confounders**
>
> We agree this is a limitation. Our mixed-effects regression assumes linear relationships between covariates and outcomes, and more complex non-linear relationships may not be captured. We will add this to the limitations section.
>
> ### **A causal intervention**
>
> We did consider this. Manipulating representational similarity through fine-tuning or representation engineering remains technically challenging, and such interventions risk introducing additional confounds; for example, modifying representations in one domain can inadvertently affect behavior in seemingly unrelated domains (e.g., https://arxiv.org/abs/2509.13450). To avoid this, we adopted the large-scale observational approach using default models. We fully agree that a clean causal intervention would be a compelling follow-up, and we mentioned this in our future work section.
>
> ### **Behavioral similarity and the early layer finding**
>
> We also find the early layer result surprising, and it is consistent with our finding that behavioral similarity alone does not explain the pattern. We don't have a definitive mechanistic explanation, but one intuition is that shared early-layer representations reflect shared conceptual and lexical grounding. Think of human negotiators who share vs. don't share cultural intuitions about money: what a fair offer looks like, what aggressive bidding signals. Models with similar foundational representations may "read" each other's moves more accurately, enabling smoother coordination regardless of strategic sophistication. Understanding the precise mechanism warrants further mechanistic interpretability research.
>
> ### **Relation to homophily**
>
> We believe homophily is related to our findings. However, whether homophily is the mediating variable and what the underlying mechanism is remains an open question that we believe warrants dedicated future investigation.

---

> > ### Author Rebuttal · Reviewer_iZeP · 2026-04-01
> >
> > The authors acknowledge my concerns. They don't feel the need to modify the paper to address them; this is fine by me. It's a good paper, with some limitations. I don't see any reason to change my score, but I appreciate the authors' contribution.

---

> > > ### Author Response · Authors · 2026-04-08
> > >
> > > Thank you so much for your constructive feedback! Even though we couldn't include the fine-tuning experiments, we will make sure to discuss this in the future work section.

---

### Official Review · Reviewer_3axZ · 2026-03-04

**Soundness:** 3
**Presentation:** 4
**Significance:** 3
**Originality:** 3
**Overall Recommendation:** 4
**Confidence:** 4

**Summary:**

In this work, authors present an extensive empirical study of multi-model interactions in multiple games. They investigate how representational similarity, defined as CKA over a dataset, impacts cooperative behavior and innovation. The four games chosen to represent cooperation include a referrential communication task, and different game theory problems. The innovation tasks are two steps creative writing setups, where agents initially share their notes before writing their final output.

**Compliance With Llm Reviewing Policy:**

Affirmed.

**Final Justification:**

This is a strong paper. My concerns regarding clarity, scope, definitions, and relevance of results were discussed in a reasonable way within the alotted time. While they do not completely disappear, I believe that they should in no way block this contribution from being accepted.

**Key Questions For Authors:**

(1) relating to (b), in 3.1 you explain that the chosen games have "long been used to see how players coordinate and collaborate", or represent a socio-economic approach to "cooperative and coordination dinamics". Why are those games representative according to previous works? Can you clarify or perhaps cite the definition of collaboration (and innovation) you are using? This is partially done in Appendix A, though missing from the main text. Or should I consider these tasks as simply examples? in which case I would tone down the general statements that hint at all collaboration and innovation being impacted by representational similarity in intro and conclusion, as well as the repeated link to humans, which without related works in the main text is rather unexpected.

This is perhaps my main concern.

(2) Lexical and semantic grounding is mentioned in the abstract, and then very briefly in the last paragraph of 5. Similarly, Appendix F.7 only explains this with a similar single sentence in the table legend. The jump from "early layers are most impactful" to stating in the abstract that collaboration is shaped by shared lexical and semantic grounding is non-trivial, and requires at least a citation, and probably more specific experiments. I would recommend toning down this claim either in more technical terms "early detokenisation" citing "The Remarkable Robustness of LLMs: Stages of Inference?" or explicitly stating it is an unverified hypothesis.

(3) while you extensively test many CKA varients, and aknowledge the existence of other representational similarity metrics, Some analysis of the differences between the metrics could be informative, and perhaps strengthen the argument. Do the differences between linear/non linear CKA inform us on the geometric properties of similarity which might characterize cooperative/innovative behaviour? Or is the correlation between all those different metrics a way to say they're all sufficiently capturing linear relations, so anything more complex is not needed?

This is more of a a personnal curiosity question when faced with those huge CKA correlation tables, but is not a major issue at all.

**Limitations:**

I would recommend limiting conclusions on 1) innovation 2) cooperation 3) lexicon and semantic information more explicitely

**Strengths And Weaknesses:**

Soundness:

> a) Soundness: Methods are sound (multiple tools are considered at each step, and justified, alternatives are explained). Empirical approach is extensive over reasonable model range, from widely used to fully open models over different sizes. Size effects, and some family effects are mentioned.

> b) Soundness: The theoretical framing is where I have my main doubts - choosing the words "cooperation" or "innovative" allows for more interesting research questions than simply saying "better at 4 collaborative tasks" or "more variation in token output from shared input". I believe nonetheless that stronger justification is required - If there is evidence from previous litterature that models would scale from these specific chosen tasks to generally more cooperative/innovative behavior in ALL contexts, it should be stated strongly as it strengthens the paper greatly. Otherwise, working on constrained definitions of cooperation/innovation, either the claims need to be toned down, or stronger warnings and limitations added.

> c) Soundness: As is often the case with empirical works, we observe something, but it is hard to be sure why it happens. The authors make great effort at showing multiple possible causes that they tested for, and provide in my opinion sufficient detailed results to allow future works to investigate potential causal links between representational similarity and cooperation/innovation.

> d) Presentation is very good. This paper is easy to read, and very extensive efforts have been made to make it easy to reproduce, including prompts, datasets, and clear methods. I've set a maximum grade to it. NONETHELESS I find the choice of putting related works in the appendix harms the narrative, and makes the paper harder to understand. Part of (b) is adressed in this appendix.

> e) Significance: sure. multiagent frameworks are increasingly relevant.

> f) Originality: sure. This has not been done before, it has been done well, and combines recent insights from multiple fields.

---

> ### Author Rebuttal · Authors · 2026-03-31
>
> We thank Reviewer 3axZ for the constructive feedback!
>
> ### **Related works in the appendix**
>
> We apologize for this. We wanted to include related work in the main body but were constrained by the page limit. Since ICML allows one extra page for the camera-ready version, we will move the related work section into the main text.
>
> ### **Cooperation and innovation scope**
>
> We chose these games because they are canonical tools for studying cooperation (Hendriks & Koster, 2010; Hendriks, 2014; Mayol, 2018; Hendriks, 2016; Hauert et al., 2006; Lai et al., 2024; Wu et al., 2024; Li & Shirado, 2025; Piedrahita et al., 2025; https://en.wikipedia.org/wiki/Public_goods_game; https://en.wikipedia.org/wiki/Ultimatum_game). To strengthen generalizability, we tested our hypothesis across multiple games covering both asymmetric and symmetric cooperative settings. Moving the related work into the main text should also help address this concern by providing the missing context. We also want to clarify that we do not claim representational similarity predicts cooperation and innovation in “all contexts”. As discussed in Section 6, we expect the relationship to be context-dependent, and understanding when it emerges or disappears remains an important direction for future work.
>
> ### **Implication of the early layer finding**
>
> Thank you for the suggestion. We will tone down the abstract claim and frame it more carefully as a hypothesis grounded in prior work showing that early transformer layers primarily handle lexical and semantic processing. We will cite the following:
>
> - https://arxiv.org/pdf/2406.19384
> - https://arxiv.org/pdf/2506.02132v4
> - https://arxiv.org/pdf/2403.01509
>
> ### **Difference between linear and non-linear CKA**
>
> Thank you for the question. There is currently no perfect metric for representational similarity; different metrics capture different geometric properties of the representation space, each with its own limitations. We used multiple CKA variants to demonstrate that our findings are not an artifact of any single metric choice. The consistent results across linear and non-linear variants suggest that the observed effects are robust. That said, whether the linear/non-linear distinction matters may depend on the specific task, and we leave this as an open question for future work.

---

> > ### Author Rebuttal · Reviewer_3axZ · 2026-04-01
> >
> > Thank you for this rebuttal. The reframing you acknowledge seems important and will in my opinion make claims easier to understand. I see no reason to reduce the score given to this paper.will maintain my score, which is high but relevant for this paper

---

> > > ### Author Response · Authors · 2026-04-08
> > >
> > > Thank you so much for your constructive feedback!

---

### Official Review · Reviewer_8dL3 · 2026-03-13

**Soundness:** 4
**Presentation:** 4
**Significance:** 3
**Originality:** 3
**Overall Recommendation:** 5
**Confidence:** 3

**Summary:**

This paper investigates the relationship between the internal representational similarity of Large Language Models (LLMs) and their behavioral outcomes in multi-agent interactions. Drawing inspiration from human social neuroscience—where neural similarity often predicts cooperative success while diversity fosters innovation—the authors conduct a large-scale empirical study involving 276 model pairs (derived from 23 open-weight LLMs). They evaluate these pairs across eight distinct tasks categorized into "Cooperation" (e.g., Public Goods Game, Coordinate) and "Novelty/Innovation" (e.g., Collaborative Story Writing, Haiku). The central finding is a clear trade-off: higher representational similarity (measured via Centered Kernel Alignment, CKA) significantly improves performance in cooperative tasks but leads to a decrease in the novelty and creativity of joint outputs. The study further identifies that similarity in early transformer layers is the strongest predictor of these behaviors, suggesting that shared lexical and semantic grounding is fundamental to model interaction.

**Compliance With Llm Reviewing Policy:**

Affirmed.

**Final Justification:**

This is a solid empirical paper that links internal model states (CKA) to multi-agent behavioral outcomes. The core finding—a clear trade-off between cooperation and innovation based on representational similarity—is a significant and practical insight for the MAS community.

​The authors' rebuttal was convincing and directly addressed my concerns. Specifically, providing empirical data on how prompts shift CKA scores (from 0.5 to 0.7) demonstrated that their metrics are robust and capture meaningful dynamics. While the study is limited to open-weight models, the methodology is rigorous and the findings are well-supported.

This work moves the field toward a more scientific understanding of LLM interactions, and I recommend it for acceptance.

**Key Questions For Authors:**

1. **Training Data Overlap:** To what extent do you think the representational similarity is a proxy for training data overlap (e.g., models trained on the same Pile or Common Crawl subsets)? If two models were trained on entirely different languages but aligned to the same concepts, would your findings still hold?
2. **Prompt Sensitivity:** Did you experiment with "persona" prompting to see if it can artificially decrease representational similarity and thereby boost innovation in highly similar pairs?
3. **The "Sweet Spot":** Based on your data, is there a theoretical "optimal" similarity score that balances both cooperation and innovation, or are they strictly mutually exclusive?
4. **Layer Selection:** You found early layers to be most predictive. Does this imply that "social" AI behavior is more about shared vocabulary than high-level reasoning? How would this change if the tasks required deeper strategic planning?

**Limitations:**

Yes, the authors have adequately discussed limitations, including the focus on open-weight models and the static nature of the tasks. They also touch upon the ethical implications of "homophily" in AI systems potentially reinforcing biases.

**Strengths And Weaknesses:**

## Strength
* **Soundness**
The paper is technically very robust. The authors utilize a significant sample size (276 pairs) which provides high statistical power. The use of linear mixed-effects regression models is a major strength, as it allows the authors to control for potentially confounding variables such as model size, parameter count disparity, and model family. This ensures that the observed effects are truly attributable to representational similarity rather than just "larger models performing better."

* **Presentation**
The manuscript is exceptionally well-structured and clear. The distinction between "Cooperation" and "Novelty" tasks is logically sound, and the visualizations (specifically the regression plots showing the divergence of cooperation vs. novelty) are intuitive. The authors also provide a clear positioning of their work within the context of both AI multi-agent systems and human neurobiology.

* **Significance**
This work addresses a critical bottleneck in Multi-Agent Systems (MAS): how to choose the "right" models for a team. By providing a predictive metric (CKA) that can be calculated without running expensive interaction simulations, the paper offers practical utility for practitioners. It also highlights a fundamental trade-off that suggests there is no "one-size-fits-all" model alignment strategy for all types of collaborative tasks.

* **Originality**
While CKA and multi-agent gaming are established, the specific application of "neural similarity" concepts from human cognitive science to the internal states of LLMs during social interaction is highly original. It shifts the focus from "what" models say (output) to "how" they represent the world (internal state) as a driver of social behavior.

## Weaknesses

* **Correlation vs. Causation:** While the statistical correlation is strong, the paper does not definitively prove that representational similarity *causes* these behaviors. It remains possible that a third factor (e.g., specific training data overlap) drives both similarity and behavior.
* **Scope of Models:** The study is limited to open-weight models because CKA requires access to internal activations. Consequently, the findings might not immediately generalize to the most powerful closed-source models (like GPT-4 or Claude 3.5), which may exhibit different behavioral patterns due to RLHF or proprietary training techniques.
* **Measurement of Innovation:** Assessing "Novelty" and "Creativity" is notoriously difficult. The reliance on LLM-based auto-raters for these metrics, while common, introduces a layer of subjective bias that could be further validated by human experts.

---

> ### Author Rebuttal · Authors · 2026-03-31
>
> We thank Reviewer 8dL3 for the thoughtful and positive feedback!
>
> ### **Training data overlap**
>
> This is an interesting question. Training data overlap likely has a meaningful effect on both representational similarity and interaction outcomes, but post-training and RLHF could also play a significant role. Exploring which factors shaping representational similarity actually drive the observed effects remains an open question for future work. Based on our early-layer findings, we suspect that two models trained on different languages but aligned to the same concepts would still show better cooperation and reduced novelty — though this scenario is difficult to realize in practice, as language and culture are deeply intertwined.
>
> ### **Prompt sensitivity**
>
> This is another interesting question. We observed that prompts can substantially shift CKA scores; for instance, prepending math-style instructions to the probing prompt increased similarity markedly (e.g., 0.5 → 0.7). However, we believe this reflects the limitations of current similarity metrics rather than a genuine change in the models' underlying representations. As noted in Section 6, as in neuroscience, there is no perfect metric for representational similarity, and developing better measures remains an open problem.
>
> ### **The optimal similarity score**
>
> We believe this is highly context-dependent rather than a fixed value. Applications requiring high novelty would benefit from pairing dissimilar models, while those requiring reliable coordination and mutual understanding would benefit from more similar ones. Identifying when similarity helps versus hurts will require testing across a broader range of tasks, which we leave for future work.
>
> ### **Early layer finding**
>
> Our results suggest that shared basic vocabulary and conceptual grounding, captured in early layers, is the strongest predictor of cooperation and innovation. However, we suspect this finding is task-dependent: tasks requiring deeper strategic planning may place greater weight on mid-to-late layer alignment. Exploring how layer importance shifts across task types is an interesting direction for future work.

---

> > ### Author Rebuttal · Reviewer_8dL3 · 2026-04-02
> >
> > ​I thank the authors for their thoughtful and professional rebuttal.
> > My concerns have been fully addressed for the following reasons:
> >
> > ​* Empirical Evidence on Prompt Sensitivity: The authors provided a concrete example of how prompting (e.g., adding math-style instructions) can shift CKA similarity scores from 0.5 to 0.7. This specific data point is highly valuable as it clarifies the extent to which representational similarity metrics capture dynamic model states.
> >
> > * ​Clarification on Task-Dependent Trade-offs: The response regarding the "Sweet Spot" and the context-dependency of similarity (cooperation vs. innovation) reinforces the paper's framework. The authors successfully articulated that the goal of the paper is not to provide a single "optimal" number, but to reveal the fundamental tension between these two behavioral outcomes.
> >
> > ​* Layer-wise Insights: The authors' hypothesis regarding strategic planning and mid-to-late layer alignment provides a more nuanced interpretation of their results. This shows that the current "early-layer" findings are well-situated within the scope of the tested tasks while acknowledging the potential complexity of more advanced interactions.
> >
> > ​* Conceptual vs. Data-driven Similarity: The discussion on training data overlap and the cross-lingual hypothetical scenario shows a deep understanding of the underlying mechanisms.
> >
> > ​Overall, the rebuttal has strengthened my confidence in the robustness of the study and its utility for the Multi-Agent Systems community. I maintain my original recommendation of Accept.

---

> > > ### Author Response · Authors · 2026-04-08
> > >
> > > Thank you so much for your constructive feedback!

---

### Official Review · Reviewer_6u4m · 2026-03-13

**Soundness:** 3
**Presentation:** 4
**Significance:** 4
**Originality:** 4
**Overall Recommendation:** 5
**Confidence:** 4

**Summary:**

This paper explores the relationship between the internal representational alignment of LLMs and their performance in multi-agent environments, measuring both cooperation and novelty. By evaluating 276 model pairs across 23 open-weight LLMs, the authors use Centered Kernel Alignment (CKA) to quantify similarity independently of model interaction. The study finds a consistent trade-off: higher representational similarity significantly improves cooperative performance in strategic games (such as word guessing and public goods) but leads to a measurable decline in collective novelty and creativity during creative writing tasks.

**Compliance With Llm Reviewing Policy:**

Affirmed.

**Final Justification:**

After the rebuttal, my concern is fully resolved. I have increased the score to 5.

**Key Questions For Authors:**

1. L310, right column: Could the authors clarify the method used to measure and control for behavioral similarity? How did you ensure that the reported effects of representational similarity are not a byproduct of models happening to make identical numerical choices in games like Divide-a-Dollar?
2. Research in biological systems, such as mice, suggests that neural similarity (shared subspace) increases through interaction (e.g., Zhang et al., Nature 2025). Did the authors study whether CKA scores increased after multiple rounds of interaction? This may be hard to do experiments without finetuning LLMs. The authors can give me an intuitive guess.
3. What is the intuition for the claim that shared lexical-semantic grounding (found in early layers) correlates with success in economic games? While this link is intuitive for Word Guessing, it is less clear why it would drive performance in numerical games like the Public Goods game.
4. Could the authors provide a version of Figure 5 that breaks down the regression coefficients for each individual cooperation game? I suspect that the influence of CKA may be lower in economic games compared to Word Guessing task.
5. What is the authors’ opinion about Platonic Representation Hypothesis (Huh et al., ICML 2024), stating that all models (including vision models and language models) converge to a similar representation given enough training data? How would you link this hypothesis to your findings?

**Limitations:**

yes

**Strengths And Weaknesses:**

## Strengths:
### 1. Novelty and originality
The paper presents a highly creative and novel research direction by applying the concept of interbrain synchrony from neuroscience to the study of LLM interactions. This approach is a significant and original contribution to the field.

### 2. Technically Solid
The authors carefully design the experiments and evaluation metrics to conclude the results.
The results are comprehensive and convincing.

### 3. Relevance
This work addresses a highly relevant problem by offering a practical indicator to predict when a team of AI models will cooperate effectively or provide diverse, novel ideas. Furthermore, these findings suggest a convergence between human brains and artificial neural networks.

### 4. Clarity
It is very well-written, and I enjoy reading the entire paper in detail.

## Weaknesses:
I would like to highlight that this is a technically solid and interesting paper. I would give a score of 5 (accept) if the authors resolve the major flaws elaborated below.

### 1. Correlation is interpreted as causation
The manuscript conflates correlation with causation. For instance, the authors use words like "The effect of representational similarity..." (Figure 1 caption), note that it "shows a strong effect", and investigate what drives this trend.

Actually, the study just measures the correlation between representational similarity (RS) and interaction outcomes across pre-trained models. Because the methodology lacks controlled manipulation (such as activation patching) to eliminate latent confounders, it cannot prove causation. Although the authors strengthen their correlational findings by isolating factors like performance disparity and model size via mixed-effects regression, this statistical control does not establish causality. Observational data, even when controlled, remain vulnerable to unmeasured confounders.

Following the principles of causal inference (e.g., Pearl, 2009), a causal effect can only be constructed through interventional experiments, such as activation patching or representation engineering, that directly manipulate the internal states.

The authors should revise their phrasing throughout the manuscript to reflect associative or predictive relationships rather than causal effects. For example, the following lines have to be rephrased.

**Suggestion:** Changing the claims about causality to correlation would be enough to resolve this weakness.
For example,
L076: effect → correlation
L093: representational similarity shows a strong effect on cooperation → is strongly correlated.


### 2. Large models trained on similar datasets.
While the authors attempt to isolate the role of representational similarity by controlling for model size and performance benchmarks (MMLU), the Capacity Gap remains a plausible confounding factor. As the authors state, CKA scores are a proxy for unmeasured latent variables like training data overlap. The authors have failed to block the backdoor path between training history and model behavior.

The claim that representational similarity is the driver of cooperation (L423, left), rather than a consequence of shared data exposure, requires further validation.

In my humble opinion, representational similarity is the driver of cooperation. I hypothesize that there is a hidden neural mechanism during training that contributes to both representational similarity and cooperative performance.

**Suggestion:** The authors should tone down their causal claims. You can frame representational similarity as a predictive indicator. This would make the paper more scientifically accurate.


### 3. Minor: Most of the games have well-defined solutions
Many of the games used in this study are well-known in economics, and their Nash equilibria are all over the internet. Since these AI models were trained on the internet, they probably just memorized the optimal way to play. Even if the solutions are not in the training data, some of the large models may be good enough to come up with Nash equilibria of those games.

 **Suggestion:** I would recommend the authors test their hypothesis using a standard cooperative MARL benchmark like Overcooked, where there is no single best solution, and requires real-time coordination. It could be some other similar games, but more friendly to LLMs (I am also not sure if most LLMs are good at Overcooked because it’s not a text-based game). Alternatively, you could put this in a limitation and future works.

### 4. Minor: Role Asymmetry in Word Guessing
Symmetric metrics like CKA ignore the asymmetric communication between a guesser and a hint maker.

**Suggestion:** This could be easily solved by averaging cooperative scores under this setting: (G,H) = (model A, model B) and (G,H) = (model B, model A), where G is a guesser, and H is a hint maker.

## Reference
- Pearl, J., 2009. Causality. Cambridge University Press.
- Huh, M., Cheung, B., Wang, T. and Isola, P., 2024. The Platonic Representation Hypothesis. ICML 2024
- Zhang, X., Phi, N., Li, Q., Gorzek, R., Zwingenberger, N., Huang, S., Zhou, J.L., Kingsbury, L., Raam, T., Wu, Y.E. and Wei, D., 2025. Inter-brain neural dynamics in biological and artificial intelligence systems. Nature

---

> ### Author Rebuttal · Authors · 2026-03-31
>
> We thank Reviewer 6u4m for the thoughtful and constructive feedback!
>
> ### **Correlation vs. causation**
>
> Thanks for flagging this. You're right that our experiments establish prediction, not causation. Our use of "effect" follows the standard terminology of mixed-effects linear regression, where the coefficient of a predictor is called its "effect" on the dependent variable, not a causal claim (https://en.wikipedia.org/wiki/Mixed_model). One of the neuroscience papers motivating our work (https://www.nature.com/articles/s41467-017-02722-7) also uses identical phrasing: "a significant effect of neural similarity on social distance." Similarly, "which aspects drive the trend?" referred to what drives the observed statistical pattern, not cooperation itself. That said, we're happy to revise any phrasing that reads as causal.
>
> ### **Games with well-defined solutions**
>
> We respectfully disagree that our games (except KBC) have well-defined solutions, since they admit multiple Nash equilibria. This is precisely why they've been canonical tools for studying cooperation in multiple domains. There's no universal strategy guaranteeing the social optimum. KBC is the exception because it has a unique equilibrium, and as we note in the paper, it shows the weakest similarity effect, consistent with this reasoning. We agree that testing across more varied settings is important and have flagged this in our future work section.
>
> ### **Role asymmetry in word guessing**
>
> Our mixed-effects regression actually does capture role asymmetry: in the equation on page 4, the random effect, $u_i$, corresponds to a hint giving model and the random effect, $v_j$, corresponds to a guessing model. Therefore, their distinct contributions are modeled separately. We included the asymmetric word guessing game specifically to test whether our hypothesis extends beyond symmetric settings.
>
> ### **Controlling for behavioral similarity**
>
> For the public good, divide-a-dollar, and KBC games, we collected each model's numerical choices during play (e.g., contribution amounts), computed the absolute difference between the two models' choices, and included this as an additional predictor in the mixed-effects regression. This lets us isolate the CKA effect from any confound due to models simply making similar choices.
>
> ### **Whether CKA scores increase after interaction**
>
> Great question. We actually ran a preliminary experiment on this. We observed CKA increases after interaction, but found that CKA is quite sensitive to prompt context: even inserting math-style instructions before the probing prompt (without any interaction) caused large CKA shifts (e.g., 0.5 → 0.7). This makes it hard to attribute post-interaction CKA increases specifically to internal synchrony, prompt sensitivity, or similar context window effects. We think this is a fascinating direction for future work.
>
> ### **Early layers and economic games**
>
> We don't have a definitive explanation, but one intuition is that shared early-layer representations reflect shared conceptual grounding. Think of human negotiators who share vs. don't share cultural intuitions about money: what a fair offer looks like, what aggressive bidding signals. Models with similar foundational representations may "read" each other's moves more accurately, enabling smoother coordination regardless of strategic sophistication. This also helps explain why behavioral similarity alone doesn't account for the cooperation gains: the effect seems to operate at the level of mutual interpretation, not just numerical convergence.
>
> ### **Regression coefficients by game**
>
> The table below presents the standardized similarity coefficient for each game, controlling for other factors. As you can see, the strongest CKA effects appear in public good and divide-a-dollar, which aligns with the intuition above; success in those games depends heavily on reading the other agent's intentions, not just converging on a fixed equilibrium.
>
> | Word guessing | Public good | Divide-a-dollar | KBC |
> |---|---|---|---|
> | 0.162 | 0.623 | 0.769 | 0.223 |
>
> ### **Platonic Representation Hypothesis**
>
> We think the PRH is partially right: representational similarity likely increases as models scale, but perfect convergence seems unlikely given differences in training recipes including RLHF. Our findings add an interesting angle: if models do converge toward a shared representation, our results predict they'd become better cooperators but less collectively creative. Tracking this tradeoff as models evolve seems like a worthwhile future direction.

---

> > ### Author Rebuttal · Reviewer_6u4m · 2026-04-01
> >
> > I thank the author for the thoughtful discussion, and I really enjoy reading it.
> > Therefore, **I increased the score to 5.**
> > This is an excellent work!
> >
> > One **subjective concern** that I have to reiterate, but I will leave it to the authors if they would like to change it or not.
> > While I appreciate the explanation, I still believe the word **effect is very misleading**.
> > This term is a leftover habit from 100 years ago. Back then, researchers used the word **effect** for experiments where they controlled everything. In those cases, the math results actually showed a cause.
> > Over time, people started using **effect** as a simple name for any math result, even when they were just observing data without controlling it. I guess because standard statistics textbooks use this term in the regression analysis.
> > However, in modern Machine Learning (as Causal Inference is also its subfield), we do not call something an effect unless we can prove it is a cause. To keep this paper accurate for a modern ML audience, I suggest that the authors replace **effect** with **association** or **relationship.**

---

> > > ### Author Response · Authors · 2026-04-08
> > >
> > > Thank you so much for your constructive feedback! We will adjust the expression accordingly. We appreciate it!

---

### Decision · Program_Chairs · 2026-04-30

**Decision:**

Accept (regular)

**Comment:**

This paper studies how representational similarity (via CKA) relates to multi-agent LLM interaction. The main result shows a trade-off where higher similarity improves cooperation but reduces novelty. This pattern consistently exists across a large set of models and tasks. Reviewers find the idea novel, the experiments solid, and the paper clearly written.

The main concern was overclaiming causality. Reviewers (6u4m, 8dL3) pointed out that the results are correlational, and the wording should reflect that. The authors acknowledged this and will revise the phrasing. Other issues (e.g., training data confounds, task scope, interpretation of early-layer effects) were discussed and mostly addressed by clarifying limitations and toning down claims.

The rebuttal was strong and resolved most concerns; all reviewers are positive after the discussion. While the work is empirical and doesn’t fully explain mechanisms, it provides a useful and practical insight for multi-agent LLM design.